# PATH PLANNING FOR DIFFUSION LANGUAGE MODEL SAMPLING

## ABSTRACT

Any order generation of discrete data using masked diffusion language models (MDMs) offers a compelling alternative to traditional autoregressive models, especially in domains that lack a natural causal ordering of data. However, current popular MDMs depart from their successful continuous diffusion model counterparts with simplified masked inference wherein unmasked tokens cannot be iteratively refined—even if there is a mistake. In this paper, we extract the full power of MDMs by introducing a novel inference sampling strategy termed *Path Planning (P2)* that decomposes each generation step into two sub-stages: planning and denoising. Under P2, the planner at every step selects appropriate tokens that are marked to be updated, which can then be sampled using the denoiser. We demonstrate that P2 generalizes all existing sampling strategies for MDMs and critically enhances generative quality through the new capability of refining and updating existing unmasked tokens. We theoretically prove that P2 establishes a (new) expanded evidence lower bound (ELBO) on the log marginal likelihood of data. We instantiate P2 with a family of planners including: 1.) Self-Planning, 2.) BERT-Planning, and 3.) Trained-Planning with a learned planner leading to SOTA generative performance for MDMs on a suite of domains. Specifically, solely using P2 inference, we observe relative improvements of $22\%$ in protein sequence foldability, $8\%$ in RNA sequence pLDDT, $4\%$ in math reasoning, $68\%$ in story generation (ROUGE score), and $33\%$ in code generation for the challenging pass@1 metric.

## 1 INTRODUCTION

Diffusion models in continuous domains are currently the most popular generative modeling family, with state-of-the-art sample quality across the entire AI spectrum of applications (Watson et al., 2023; Rombach et al., 2022). The success of the diffusion framework in continuous spaces, comparatively, raises the possibility of having similarly expressive models that can also operate on discrete data domains. Despite the appeal of discrete diffusion models, which are arguably a more natural for certain discrete domains—e.g., biological sequences—that do not have a causal ordering, the most successful discrete generative models are autoregressive models (Achiam et al., 2023). One key reason that drives this gap is that, despite the generality of accommodating a multitude of noising processes, most successful discrete diffusion approaches have converged to absorbing state diffusion (Austin et al., 2021; Lou et al., 2023) (MDMs). Moreover, while considerable effort has focused on improving training for MDMs (Sahoo et al., 2024; Shi et al., 2024; Gat et al., 2024; Shi et al., 2024), resulting in new, simple, and scalable training recipes, considerably less attention has been devoted to unlocking their full potential at inference—which is limited to simple uniform denoising. This raises a question: *Can we design new inference strategies for MDMs to improve generative quality?*

**Current work**. In this paper, we answer the above research question affirmatively by investigating how the order in which tokens are unmasked during MDM inference affects generative quality. We motivate our investigation by making the critical observation that, while the MDM reverse process requires that each token is uniformly likely to be unmasked at a given step, this correctly reconstructs the true data distribution only under a perfect denoiser. However, since any trained MDM is inherently imperfect due to the nature of training and convergence in non-convex optimization, it has been empirically observed that a uniformly random unmasking order is suboptimal in many settings (Ou et al., 2024; Shih et al., 2022; Li et al., 2021). Moreover, in current MDM inference it is not possible to course-correct incorrectly denoised tokens at future steps during inference, which leads to error propagation and overall suboptimal generative quality.

We begin our study by reexamining the typical MDM ELBO and show that, for a fixed denoiser, we can expand the ELBO to include two additional terms, both involving a "planner" whose role is to select which tokens should be unmasked at a given inference step *as well as* optionally choosing already unmasked tokens to be resampled (see Figure 1). Our ELBO shows that while the optimal planner for the optimal denoiser is indeed uniform unmasking, the strategy prescribed by the reverse process, one can obtain better generative quality for an imperfect denoiser through the use of a non-uniform planner.

**Main contributions**. These observations lead to our proposed method, *Path Planning (P2)*, which makes use of the expanded ELBO to introduce a family of planners for use at inference time. Crucially, by noting the similarity between the planner ELBO terms and the typical MLM objective, we show that in practice we can obtain effective planners by employing either pre-trained BERT-type models, training a light-weight planner offline, or simply using the already trained denoiser. Moreover, we show that P2 generalizes all known existing sampling strategies in the MDM literature (see Table 1). We validate our P2 framework across a diverse set of experimental settings, showing that by using P2, a 1B parameter MDM model can outperform a 7B Llama model in math reasoning while far outpacing state-of-the-art ARMs for code generation on the same-sized models. At the same time, for biological sequence design, we show that the combination of

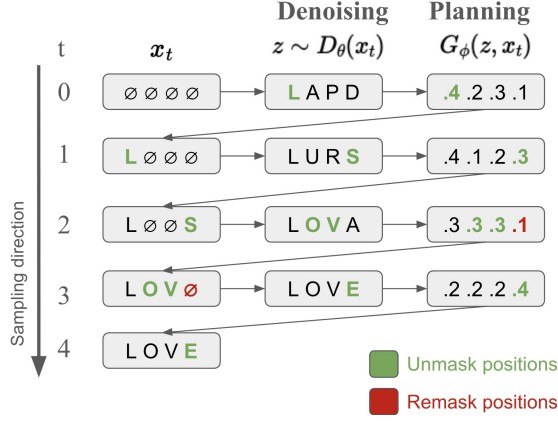

Figure 1: Illustration of P2 sampling (Algorithm 1). At each step, the denoiser $D_\theta$ predicts $z$, and the planner $G_\phi$ selects positions to unmask (green) and remask (red).

P2 and DPLM (Wang et al., 2024) leads to state-of-the-art generation quality for proteins. Finally, for RNA design, we outperform all prior models and observe that our sequences lead to higher structural plausibility than even true, naturally occurring sequences.

## 2 BACKGROUND AND PRELIMINARIES

**Notation**. Let $\mathcal{V} = \{1, \ldots, d\}$ be a finite vocabulary set. We designate the final element of this set to a specialized mask token $d = \mathbf{m}$, whereas the remaining $d - 1$ elements in $\mathcal{V}$ form the categories found in a typical vocabulary set. We are interested in generating sequences of length $L$ from $\mathcal{V}$. A discrete data sample $\mathbf{x}$ is then a realization of a category in $\mathcal{V}^L$. Let $\Delta^d := \{v \in \mathbb{R}^d : v^i \geq 0, i = 1, \ldots, d, \sum_{i=1}^d v^i = 1\}$ represent the d-dimensional probability simplex. Each point on $u \in \Delta^d$ corresponds to a categorical distribution $\mathrm{Cat}(j; u) = u^j$ for $j \in \mathcal{V}$. We write a discrete sequence of length $L$ as $\mathbf{x} = (x^1, \ldots, x^L) \in \mathcal{V}^L$. The data distribution $\mathbf{p}_{\mathrm{data}}$ is provided as an empirical distribution on $n$ sequences in the form of a training set $\mathcal{D} = \{\mathbf{x}\}^n \subset \mathcal{V}^L$. We further use boldface $\mathbf{x}$ to denote the entire sequence and normal script to indicate an individual token. We denote for $x \in \mathcal{V}$, $\delta(x) \in \Delta^d$ given by $\mathrm{Cat}(j; \delta(x)) = 1$ if $j = x$ and 0 otherwise. Finally, we reserve superscripts for set indexing purposes, e.g. $x^i, i \in [d]$, while subscripts are used to represent positions in time of a discrete sample $x_t, t \in [0, 1]$.

### 2.1 MASKED DISCRETE DIFFUSION MODELS

We can define diffusion models on discrete spaces by constructing a forward noising process that progressively converts the data distribution $\mathbf{p}_{\mathrm{data}}$ to a structureless prior. Without loss of generality, let $\mathbf{p}_0(\mathbf{x}) := \mathbf{p}_{\mathrm{data}}(\mathbf{x})$ be the data distribution at time $t = 0$ and let $\mathbf{p}_1 := [\delta(\mathbf{m})]^n$ the prior which consists of a fully masked sequence. For simplicity of exposition, we consider a discretization of time into $T$ sub-intervals, i.e. $t(i) = i/T$. This enables the specification of the forward corruption process using a noising kernel $\mathbf{p}_t(\mathbf{x}_t | \mathbf{x}_0)$. One of the most popular forward-noising processes (Sahoo et al., 2024; Gat et al., 2024; Shi et al., 2024; Zhao et al., 2024a) is the so-called "simplified masked" process, which corrupts each unmasked token $x_t^i \neq \mathbf{m}$ in a sequence independently:

$$\mathbf{p}_t(\mathbf{x}_t | \mathbf{x}_0) = \prod_{i=1}^L p_t(x_t^i | x_0^i) = \prod_{i=1}^L \mathrm{Cat}(x_t^i; \alpha_t \delta(x_0^i) + (1 - \alpha_t)\delta(\mathbf{m})). \tag{1}$$

Here, $\alpha_t$ plays the role of a noise schedule and is an decreasing reparametrization of time such that $\alpha_0 = 1$ and $\alpha_1 = 0$. A key detail of the simplified masking process is that once a token is masked, it remains masked for the remainder of the process. Similar to conventional diffusion models in continuous space, the specification of the forward process also allows us to write a time-reversed process that iteratively denoises a sample from $t \to t-1$ until a clean, fully unmasked sample is procured at time $t = 0$. For the simplified masking process, the time reversal also factorizes across tokens within the sequence. More precisely, the reverse transition kernel for a token $x_t^i$ conditioned on $x_0^i$ is given by:

$$q_t(x_{t-1}^i|x_t^i, x_0^i) = \begin{cases} \mathrm{Cat}(x_{t-1}^i; \delta(x_t^i)) & x_t^i \neq \mathbf{m} \\ \mathrm{Cat}\left(x_{t-1}^i; \frac{(1-\alpha_{t-1})\delta(\mathbf{m}) + (\alpha_{t-1}-\alpha_t)\delta(x_0^i)}{1-\alpha_t}\right) & x_t^i = \mathbf{m}. \end{cases} \quad (2)$$

It is important to highlight that once a token is unmasked and realized as one of the remaining $d-1$ categories, it remains fixed for the rest of the denoising steps. The form of Equation (2) suggests a natural parameterization to learn the reverse process using a time-independent denoiser network $D_\theta : \mathcal{V}^L \to (\Delta^d)^L$ that predicts the probabilities of a clean sample $\mathbf{z} \sim D_\theta(x_t)$ at $t = 0$:

$$q_{t,\theta}(x_{t-1}^i|x_t^i, D_\theta^i(\mathbf{x}_t)) = \begin{cases} \mathrm{Cat}(x_{t-1}^i; \delta(x_t^i)) & x_t^i \neq \mathbf{m} \\ \mathrm{Cat}\left(x_{t-1}^i; \frac{(1-\alpha_{t-1})\delta(\mathbf{m}) + (\alpha_{t-1}-\alpha_t)D_\theta^i(\mathbf{x}_t)}{1-\alpha_t}\right) & x_t^i = \mathbf{m}. \end{cases} \quad (3)$$

where $D_\theta^i$ refers to selecting the $i$-th index of the output of the denoiser $D_\theta(\mathbf{x}_t)$—i.e. the approximate distribution of $x_0^i$ given the conditional information from $\mathbf{x}_t$. Using the reverse parametrization and taking an infinitesimal time discretization $T \to \infty$, it is possible to construct an evidence lower bound (ELBO) to the log marginal likelihood on the data distribution of the approximate data distribution from iteratively sampling via Eq. 3, $\mathbf{p}_\theta(\mathbf{x}_0)$, which also yields a natural optimization objective for learning the denoiser $D_\theta$,

$$\log \mathbf{p}_\theta(\mathbf{x}_0) \geq -\int_0^1 \frac{d\alpha_t}{dt} \cdot \frac{1}{1-\alpha_t} \mathbb{E}_{\mathbf{x}_t \sim \mathbf{p}_t(\cdot|\mathbf{x}_0)} \left[ \sum_{i=1, x_t^i=\mathbf{m}}^L \delta(x_0^i)^T \log D_\theta^i(\mathbf{x}_t) \right] dt. \quad (4)$$

This effectively renders training a masked discrete diffusion model as optimizing a weighted cross-entropy loss (Eijkelboom et al., 2024).

A major limitation of vanilla MDMs is that, in the continuous-time limit $T \to \infty$, the probability of denoising multiple tokens simultaneously vanishes due to independent updates via Eq. 3. In this regime, an analytic Gillespie-style sampler (Gillespie, 1977; 1976) reveals that denoising proceeds by uniformly sampling a masked position (see §D.3), offering no control over the generation order. Next, we consider a new, more complex inference scheme that principally allows for changing unmasked tokens to any other token in $\mathcal{V}$, allowing for the index of the next token to be resampled.

Table 1: Generalization of existing sampling Methods within the P2 Framework. **Masked Planner** ($G_M^j$) gives the probability that a mask token should be unmasked. **Unmasked Planner** ($G_U^j$) gives the probability that an unmasked token should be kept. $D_\theta^j$ gives the prediction probability of the denoiser at position $j$. TopKMargin($D_\theta^j(\mathbf{x}_t)$) denotes selection based on the probability margin between the top-2 predictions.

| Method | Remasking | Planning | Stochasticity Control | Mask Planner ($G_M^j(\mathbf{z}, \mathbf{x}_t)$) | Unmask Planner ($G_U^j(\mathbf{z}, \mathbf{x}_t)$) |
|---|---|---|---|---|---|
| Ancestral (Shi et al., 2024; Sahoo et al., 2024) | ✗ | ✗ | ✗ | $\mathcal{U}(0,1)$ | 1 |
| MaskGIT (Chang et al., 2022b) | ✗ | ✓ | ✗ | $\mathrm{Cat}(z^j; D_\theta^j(\mathbf{x}_t))$ | 1 |
| Greedy Ancestral (Gong et al., 2025) | ✗ | ✓ | ✗ | $\mathrm{Cat}(z^j; D_\theta^j(\mathbf{x}_t))$ | 1 |
| TopK-Marginal (Kim et al., 2025) | ✗ | ✓ | ✗ | TopKMargin($D_\theta^j(\mathbf{x}_t)$) | 1 |
| DFM Sampling (Campbell et al., 2024) | ✗ | ✗ | ✓ | $\mathcal{U}(0,1)$ | $\mathcal{U}(0,1)$ |
| RDM Sampling (Zheng et al., 2023) | ✓ | ✓ | ✗ | $\mathrm{Cat}(z^j; D_\theta^j(\mathbf{x}_t))$ | $\mathrm{Cat}(z^j; D_\theta^j(\mathbf{x}_t))$ |
| DDPD (Liu et al., 2024) | ✓ | ✓ | ✗ | $G_\phi^j(\mathbf{z})$ | $G_\phi^j(\mathbf{z})$ |
| **P2 (Self-Planning)** | ✓ | ✓ | ✓ | $\mathrm{Cat}(z^j; D_\theta^j(\mathbf{x}_t))$ | $\mathrm{Cat}(z^j; D_\theta^j(\mathbf{x}_t))$ |
| **P2 (BERT Planner)** | ✓ | ✓ | ✓ | $\mathrm{Cat}(z^j; D_\theta^j(\mathbf{x}_t))$ | $\mathrm{Cat}(z^j; B_\phi^j(\mathbf{z}))$ |
| **P2 (Trained Planner)** | ✓ | ✓ | ✓ | $\mathrm{Cat}(z^j; D_\theta^j(\mathbf{x}_t))$ | $T_\phi^j(\mathbf{x}_t, \mathbf{z})$ |

## 3 DISCRETE DIFFUSION WITH PATH PLANNING

We now aim to improve the generation capability of MDMs by modifying the reverse denoising process by introducing a planning component in a novel inference strategy termed P2. In Table 1 we contrast P2 with the extensive existing literature in planning for MDMs. In particular, P2 is the only model with remasking, planning, and stochasticity control. In what follows, we further explore novel forms of planners as we find that the optimal planner depends on the application.

### 3.1 THE P2 SAMPLING STRATEGY

In order to formulate P2, we begin by modifying the approximate backwards process (Eq. 3), introducing a new function $G_\phi : \mathcal{V}^L \times \mathcal{V}^L \to [0,1]^L$, with parameters $\phi$, which we refer to as the planner. Intuitively, $G_\phi^j(\mathbf{z}, \mathbf{x}_t)$ approximates the probability that the $j$'th token in a partially denoised sequence should be (re)sampled conditioned on the rest of the sequence $\mathbf{x}_t \in \mathcal{V}^L$ and predicted clean data $\mathbf{z}$.

P2 departs from the vanilla MDM inference procedure, where the backward transition $q_{t,\theta}(x_{t-1}^i | x_t^i, D_\theta^i(\mathbf{x}_t))$ in Equation (3) is denoised independently for each coordinate in the sequence by instead assigning the likelihood of denoising at $x_t^i$ as a *function of the planner* $G_\phi$. Succinctly, the P2 strategy is used to update a partially noised sequence $\mathbf{x}_t$ by first sampling a denoised sequence given a partially noised sequence $x_t$, i.e., $\mathbf{z} \sim D_\theta(\mathbf{x}_t)$, after which we can leverage our planner $G_\phi(\mathbf{z}, \mathbf{x}_t)$ to determine which positions in the sequence to update. If $x_t^i = \mathbf{m}$, we unmask to the sample $z^i$ with probability $G_\phi^i(\mathbf{z}, \mathbf{x}_t)$. Conversely, if $x_t^i \neq \mathbf{m}$, with probability $G_\phi^i(\mathbf{z}, \mathbf{x}_t)$, we construct $\bar{\mathbf{x}}_t$ from $\mathbf{x}_t$ via setting $x_t^i$ to $\mathbf{m}$ (remasking), and then we resample $x_{t-1}^i \sim D_\theta^i(\bar{\mathbf{x}}_t)$ so that $x_{t-1}^i \neq x_t^i$. The conditionally independent coordinate-wise reverse transitions are then, for $x_{t-1}^i \neq x_t^i$:

$$q_{t,\theta}(x_{t-1}^i | \mathbf{x}_t, \mathbf{z}) = \begin{cases} \mathrm{Cat}\left(x_{t-1}^i; \frac{\alpha_{t-1}-\alpha_t}{1-\alpha_t} G_\phi^i(\mathbf{z}, \mathbf{x}_t)\delta(z^i)\right) & x_t^i = \mathbf{m} \\ \mathrm{Cat}\left(x_{t-1}^i; \frac{(\alpha_{t-1}-\alpha_t)G_\phi^i(\mathbf{z},\mathbf{x}_t)}{(1-\alpha_t)(1-\mathrm{Cat}(x_t^i, D_\theta^i(\bar{\mathbf{x}}_t)))} D_\theta^i(\bar{\mathbf{x}}_t)\right) & x_t^i \neq \mathbf{m}, \end{cases} \quad (5)$$

and the case $x_{t-1}^i = x_t^i$ is obtained by ensuring these sum to 1.

We highlight the masked case in Equation (5) proceeds in the same manner as the classical MDM inference setup outside of the key difference that the index to be denoised is selected by the planner $G_\phi$. Furthermore, P2 updates a masked token by an intermediate step of remasking and then denoising to a different token by resampling from $D_\theta(\bar{\mathbf{x}}_t)$, using the newly constructed $\bar{\mathbf{x}}_t$. Critically, we see that P2 allows for the planner $G_\phi$ to guide the denoising process towards a more optimal path of denoising orders using the information from both the partially noised sequence $\mathbf{x}_t$ and the predicted clean sequence $\mathbf{z}$ from the denoiser—including resampling incorrect denoised tokens. We outline the full top-k instantiation of the P2 algorithm in pseudocode in Algorithm 1 and include a computationally viable Gillespie sampler method (Gillespie, 1977; 1976) for P2 in Algorithm 5.

### 3.2 DESIGNING THE PLANNER

The P2 sampling strategy requires the design of a planner $G_\phi$ whose role is to select tokens to update by exploiting information about the current $\mathbf{x}_t$ and $\mathbf{z}$. To construct the planner, such that we can guarantee convergence to a fully unmasked sequence at $t = 1$ we first decompose $G_\phi$ into two components:

$$G_\phi^j(\mathbf{z}, \mathbf{x}_t) = \begin{cases} G_M^j(\mathbf{z}, \mathbf{x}_t) & x_t^j = \mathbf{m} \\ 1 - G_U^j(\mathbf{z}, \mathbf{x}_t) & x_t^j \neq \mathbf{m}. \end{cases} \quad (6)$$

where $G_M^j(\mathbf{z}, \mathbf{x}_t)$ is the *masked token planner* that predicts the likelihood that a masked token at the $j$'th position should be unmasked, and an *unmasked token planner* $G_U^j(\mathbf{z}, \mathbf{x}_t)$ which predicts the probability that an unmasked token should be kept. We then employ a modified "top k" sampling strategy, which introduces the possibility of changing multiple tokens per iteration and better exploits the information provided by a monotone non-decreasing scheduler function $\kappa : \{1, \dots, L\} \to \{1, \dots, L\}$, with $\kappa(L) = L$. The purpose of the scheduler is to determine the number of tokens, $\kappa(t)$, that are guaranteed to be unmasked at the reverse step $t$.

The final component of P2 is a stochasticity parameter $\eta$, which controls the frequency of remasking as in DFM (Campbell et al., 2024). This parameter allows a practitioner to control the trade-off between efficiency and additional self-correction and is standard in continuous diffusion models. This defines a family of probability path measures for our planner:

$$\tilde{G}_\eta^j(\mathbf{z}, \mathbf{x}) \propto \eta\mathrm{Cat}(x^j; \delta(\mathbf{m}))G_M^j(\mathbf{z}, \mathbf{x}) + (1 - \mathrm{Cat}(x^j; \delta(\mathbf{m})))G_U^j(\mathbf{z}, \mathbf{x}), \quad \eta \geq 0. \quad (7)$$

### 3.3 A FAMILY OF PLANNERS: INSTANTIATIONS OF P2

We next propose three practical instantiations of the planner $G_\phi$ employed in our P2 framework.

**Self-Planning.** We propose a self-planning mechanism by leveraging the denoiser's own predicted probabilities to guide updating decisions. Concretely, we set $G_U^j(\mathbf{z}, \mathbf{x}) = G_M^j(\mathbf{z}, \mathbf{x}) = \mathrm{Cat}(z^j; D_\theta^j(\mathbf{x}))$, and as a result the denoiser itself serves as the planner. For masked positions, the

denoiser is trained to predict tokens given the surrounding context, and the predicted probabilities serve as confidence estimates for the correctness of token predictions. This methodology aligns with established practices in the literature (Gong et al., 2025; Chang et al., 2022a; Zheng et al., 2023; Wang et al., 2024; 2025b) as outlined in Table 1. We further highlight that instantiations of the self-planner recover the methodology of established results. For instance, both MaskGIT (Chang et al., 2022b) and Greedy Ancestral (Gong et al., 2025) are special cases of self-planning *without stochasticity control* and when the unmask planner $G_U(\mathbf{z}, \mathbf{x}) = 1$—disabling the remasking technique from self-planning (see e.g. Table 1). Surprisingly, for unmasked tokens probabilities, the denoiser—despite only being trained solely on masked positions—still has access to robust representations of unmasked positions, and as a result is still informative for resampling, and thus sequence generation.

---

**Algorithm 1** P2 Sampling (simplified)

1: **Input:** $\mathbf{x}_0 \leftarrow (\mathbf{m}, \dots, \mathbf{m}), G_\phi, D_\theta$, Schedule $\kappa$
2: **for** $t = 1 : L$ **do**
3:     **Plan:**
4:     $\mathbf{z} \sim D_\theta(\mathbf{x}_t)$
5:     UpdatePos $\leftarrow \text{Top}_{\kappa(t)}\left(\tilde{G}_\eta(\mathbf{z}, \mathbf{x}_t)\right)$
6:     **Denoise:**
7:     $x_t^j \leftarrow \begin{cases} z^j & \text{if } j \in \text{UpdatePos} \wedge x_t^j = \mathbf{m} \\ \mathbf{m} & \text{if } j \notin \text{UpdatePos} \end{cases}$
8: **end for**
9: **return** $\mathbf{x}_L$

---

**Algorithm 2** P2 Planner Training (Frozen $D_\theta$)

1: **Input:** $\mathbf{x}_0 \sim \mathbf{p}_0, D_\theta, G_\phi$
2: Sample $t \sim \mathcal{U}(0, 1)$
3: Sample $\mathbf{x}_t \sim p_t(\cdot \mid \mathbf{x}_0)$
4: $\mathbf{z} \sim D_\theta(\mathbf{x}_t)$
5: $\text{logits}^j \leftarrow G_M^j(\mathbf{z}, \mathbf{x}_t)$ for $j$ such that $x_t^j = \mathbf{m}$ and $G_U^j(\mathbf{z}, \mathbf{x}_t)$ otherwise
6: $\text{label}^j \leftarrow \mathbb{1}[z^j = x_0^j]$
7: $\mathcal{L}(\phi) \leftarrow \frac{d\alpha_t}{dt} \cdot \frac{1}{1 - \alpha_t} \cdot \text{CE}(\text{label}, \text{logits})$
8: **Update:** $\phi \leftarrow \phi - \nabla_\phi \mathcal{L}(\phi)$

---

**BERT-planning**. In BERT-planning, we introduce a class of planners based on a pre-trained BERT model (Devlin et al., 2019), which is trained to denoise from a 12% masking rate at training and 1.5% random flipping rate. Despite such a simple training objective, BERT learns to estimates the naturalness of an unmasked token with the predicted probabilities which demonstrates wide application in zero-shot mutation prediction (Hie et al., 2024), suggesting that BERT may serve as an effective choice for $G_U$. Compared to training a dedicated planner that is equal-size to denoiser as in DDPD (Liu et al., 2024), BERT is more versatile, flexible in sizes, and often available in common tasks such as text (Devlin et al., 2019; Liu et al., 2019; Lan et al., 2020), protein (Lin et al., 2023; Hayes et al., 2025; Wang et al., 2024; 2025b) and RNA (Penić et al., 2024). Mathematically, we formulate BERT planning using a BERT model $B_\phi : \mathcal{V}^L \to (\Delta^d)^L$, such that $\text{Cat}(z^j; B_\phi^j(\mathbf{z}))$ assigns the probability that the $j$-th token in the sequence $\mathbf{z}$ is clean. In BERT planning we set the unmask planner to be the BERT $G_U^j(\mathbf{z}, \mathbf{x}) = \text{Cat}(z^j; B_\phi^j(\mathbf{z}))$ and mask planner to be the denoiser $G_M^j(\mathbf{z}, \mathbf{x}) = \text{Cat}(z^j; D_\theta^j(\mathbf{x}))$.

**Trained-Planner**. We can also employ a trained planner that operates on the denoiser's prediction and the current masked input. Specifically, we freeze the denoiser during training and fine-tune the BERT planner by taking $G_M^j(\mathbf{z}, \mathbf{x}) = G_U^j(\mathbf{z}, \mathbf{x}) = \text{Cat}(z^j; B_\phi^j(\mathbf{z}))$ using a cross-entropy loss derived from the planner ELBO objective. In this case, the planner learns to predict whether each token should be selected based on whether the denoiser's output matches the ground-truth token. As detailed in Algorithm 2, the planner is supervised to match the optimal decoding trajectory—i.e., one that prioritizes correct positions.

During sampling for experiments using P2 Train - see Table 5 - we use the same parameterization as with P2 BERT for constructing $\tilde{G}_\eta^j$ of Eq. 7 in Algorithm 1. That is, we set the unmask planner to be the fine-tuned BERT $G_U^j(\mathbf{z}, \mathbf{x}) = \text{Cat}(z^j; B_\phi^j(\mathbf{z}))$ and mask planner to be the denoiser $G_M^j(\mathbf{z}, \mathbf{x}) = \text{Cat}(z^j; D_\theta^j(\mathbf{x}))$.

We emphasize that, although we only use the fine-tuned BERT model as $G_U$ for sampling, it is trained on both masked and unmasked positions in Algorithm 2. This allows for the model to have a meaningful training signal in that it gets to see both 0 and 1 as the label - without training on masked positions this would not be the case, since the label is always 1 in unmasked positions.

The training of the planner in Algorithm 2 is theoretically supported by the following propositions. Note that indeed in Proposition 2 we make the assumption that the same network backbone is used as both $G_M$ and $G_U$ in training, and this should always be done in practice, even if one intends to use $T$ only as $G_U$ or $G_M$ during sampling.

**Proposition 1.** *Define $P_0^{\theta,\phi} \in \Delta^{d^L}$ by $P_0^{\theta,\phi}(\mathbf{x}) = \mathbb{P}(X_0^{\theta,\phi} = \mathbf{x})$, where $X^{\theta,\phi}$ is the continuous time Markov chain resulting from sending $T \to \infty$ in the discrete-time P2 formulation Eq. 5. Then we have an "Evidence Based Lower Bound" $\mathcal{E}(\mathbf{x}_0) \leq \log(P_0^{\theta,\phi}(\mathbf{x}_0))$ for each fixed $\mathbf{x}_0 \in \mathcal{V}^L$ given by $\mathcal{E}(\mathbf{x}_0) = \mathcal{E}_{MP}(\mathbf{x}_0) + \mathcal{E}_{UP}(\mathbf{x}_0) + \mathcal{E}_D(\mathbf{x}_0)$, where:*

$$\mathcal{E}_{MP}(\mathbf{x}_0) = -\int_0^1 \frac{d\alpha_t}{dt} \cdot \frac{1}{1-\alpha_t} \mathbb{E}_{\mathbf{x}_t \sim \mathbf{p}_t(\cdot;\mathbf{x}_0)} \left[ \sum_{i=1, \mathbf{x}_t^i = \mathbf{m}}^{L} \mathbb{E}_{\mathbf{z} \sim D_\theta(\mathbf{x}_t)} \left[ CE\left( Cat(z^i; \delta(x_0^i)), G_M^i(\mathbf{z}, \mathbf{x}_t) \right) \right] \right] dt$$

$$\mathcal{E}_{UP}(\mathbf{x}_0) = -\int_0^1 \frac{d\alpha_t}{dt} \cdot \frac{1}{1-\alpha_t} \mathbb{E}_{\mathbf{x}_t \sim \mathbf{p}_t(\cdot;\mathbf{x}_0)} \left[ \sum_{i=1, \mathbf{x}_t^i \neq \mathbf{m}}^{L} \mathbb{E}_{\mathbf{z} \sim D_\theta(\mathbf{x}_t)} \left[ CE\left( Cat(z^i; \delta(x_0^i)), G_U^i(\mathbf{z}, \mathbf{x}_t) \right) \right] \right] dt$$

$$\mathcal{E}_D(\mathbf{x}_0) = -\int_0^1 \frac{d\alpha_t}{dt} \cdot \frac{1}{1-\alpha_t} \mathbb{E}_{\mathbf{x}_t \sim \mathbf{p}_t(\cdot;\mathbf{x}_0)} \left[ \sum_{i=1, \mathbf{x}_t^i = \mathbf{m}}^{L} \delta(\mathbf{x}_0^i)^\top \log(D_\theta^i(\mathbf{x}_t)) \right] dt.$$

*Here $\mathbf{p}_t$ is defined per Eq. 1 and $CE(a, b) = a \log(b) + (1-a)\log(1-b)$ for $a, b \in [0, 1]$, with $0 \log 0 = 0$.*

Here $\mathcal{E}_D$ is the ELBO of the denoiser in a standard MDM, $\mathcal{E}_{UP}$ is the ELBO of the unmasking planner, and $\mathcal{E}_{MP}$ is the ELBO of the masking planner. See §B for a proof via taking limits of the time-discretized ELBO for the dynamics Eq. 5 and §D for a proof and more fine-grained theoretical analysis from a continuous-time point of view. This proposition establishes the validity of using P2 and presents a novel, finer-grained ELBO for planner-based masked language models. The explicit inclusion of a non-trivial $G_\phi$ in our ELBO allows both for training a planner and for evaluating the efficacy of an "off-the-shelf" planner. Table S6, we show that planners ranging from 8M to 3B parameters have similar ELBOs and thus have similar generation performance (Figure S3), which corroborates the effectiveness of training on this bound. We also remark that, while we use the default stochasticity parameter of $\eta = 1$ to evaluate the ELBO and for the loss in P2 train and find this to be indicative of planner performance, one could also use the ELBO of Proposition 1 to evaluate varied stochasticity levels. We include the form of the ELBO for general $\eta$ in §B.3 for reference.

We note that the loss $\mathcal{L}(\phi)$ used in Algorithm 2 operates on a frozen denoiser $D_\theta$, and $\mathcal{L}(\phi) = -\mathbb{E}_{\mathbf{x}_0 \sim \mathbf{p}_0}[\mathcal{E}_{MP}(\mathbf{x}_0) + \mathcal{E}_{UP}(\mathbf{x}_0)]$. Moreover, $\mathcal{E}_{MP}$ optimizes the role of the masked planner as a mechanism for selecting the viable masked position to insert a "clean" token as suggested by $D_\theta$. $\mathcal{E}_{UP}$ acts as a mechanism for selecting an unmasked token to resample via remasking and inserting back into $D_\theta$. Indeed, these roles are verified via finding the optimal form of the trained planner as per Proposition 2.

**Proposition 2.** *Let $T_\phi : \mathcal{V}^L \times \mathcal{V}^L \to [0, 1]$ be trained via $\mathcal{L}(\phi) = -\mathbb{E}_{\mathbf{x}_0 \sim \mathbf{p}_0}[\mathcal{E}_{MP}(\mathbf{x}_0) + \mathcal{E}_{UP}(\mathbf{x}_0)]$, taking $G_U = G_M = T_\phi$ in $\mathcal{E}_{MP}$ and $\mathcal{E}_{UP}$ from Proposition 1. Define, for $\mathbf{z}, \mathbf{x_t} \in \mathcal{V}^L$ with $\mathbf{z}$ a sequence of unmasked tokens satisfying $\mathbf{z}^i = \mathbf{x}_t^i$ for all $i$ such that $\mathbf{x}_t^i \neq \mathbf{m}$:*

$$\bar{T}^i(\mathbf{z}, \mathbf{x}_t) = \begin{cases} \mathbf{p}_0\left( x_0^i = z^i | x_0^j = x_t^j, \forall j \neq i \text{ such that } x_t^j \neq \mathbf{m} \right), & x_t^i \neq \mathbf{m} \\ \mathbf{p}_0\left( x_0^i = z^i | x_0^j = x_t^j, \forall j \text{ such that } x_t^j \neq \mathbf{m} \right), & x_t^i = \mathbf{m} \end{cases}.$$

*Then, for any $D_\theta$, $\mathcal{L}(\phi)$ is uniquely minimized over $T_\phi^i$ when $T_\phi^i(\mathbf{z}, \mathbf{x}_t) = \bar{T}^i(\mathbf{z}, \mathbf{x}_t)$.*

Observe that this means, for any denoiser, the optimal $T_\phi^i$ is aiming to steer towards planned paths which are representative of the data distribution in both its roles as $G_U^i$ and $G_M^i$. In unmasked positions, a token in position $i$ is kept with probability proportional to to the probability the token is in the data distribution conditionally upon the information from the partially denoised sequence $x_t$ in positions other than $i$. In masked positions, a suggested token $z^i$ for position $i$ from the denoiser is selected with probability proportional to the probability $z^i$ is in position $i$ under the data distribution conditionally upon the information from the current $x_t$. For further discussion of the form of the optimal planner and a proof of Proposition 2, see §B.2.

## 4 EXPERIMENTS

We empirically evaluate our Path Planning (P2) inference framework for MDMs across three distinct discrete generative modeling tasks: protein sequence generation, natural language generation, and RNA sequence generation. Our main experiments Section 4.1-Section 4.3 aim to investigate the em-

pirical benefit P2 by evaluating the generated sequences for their functional quality, sample diversity, and task completion at various model scales. We also conduct comprehensive ablations to investigate the impact of planner choice in §4.4 and finally turn to inference-time scaling experiments in §4.5.

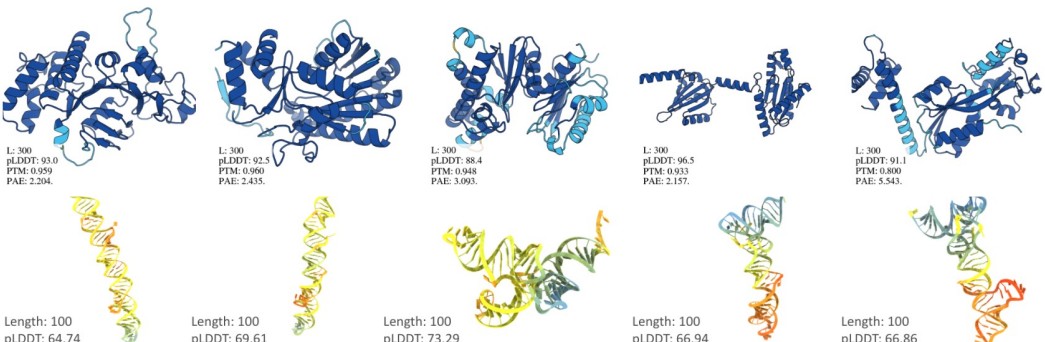

Figure 2: Visualizing the predicted structures of generated protein (top) and RNA (bottom) sequences. Additional structures depicted in Figure S7.

Table 2: Protein sequence generation benchmark. We evaluate structure quality via pLDDT, pTM, and pAE, and diversity via token entropy and sequence uniqueness. Foldability is the percentage of sequences satisfying pLDDT > 80, pTM > 0.7, and pAE < 10. See Appendix F.1 for setup and Table S4 for model size ablations.

| Model | pLDDT↑ | pTM↑ | pAE↓ | Foldability (%)↑ | Entropy↑ | Diversity (%)↑ |
|---|---|---|---|---|---|---|
| EvoDiff | 31.84 | 0.21 | 24.76 | 0.43 | **4.05** | 93.19 |
| ESM3 | 34.13 | 0.23 | 24.65 | 1.50 | 3.99 | **93.44** |
| ProGen2 | 49.38 | 0.28 | 23.38 | 4.48 | 2.55 | 89.31 |
| DPLM | 80.23 | 0.65 | 12.07 | 48.14 | 3.14 | 92.80 |
| **DPLM + P2-train** (ours) | **83.45** | **0.72** | **10.15** | **58.86** | 3.35 | 92.69 |

## 4.1 PROTEIN SEQUENCE GENERATION

We consider the task of protein sequence generation and measure the foldability, structural quality (pLDDT, pTM, pAE), and diversity (diversity & entropy) of generated proteins and benchmark against state-of-the-art autoregressive and MDMs. Through this experiment we assess whether P2 improves structural metrics while preserving entropy and diversity of generated sequences. For each model we generate 100 sequences at lengths in $\{200, 300, \ldots, 800\}$ using its default decoding strategy. Structural quality is assessed using ESMFold (Lin et al., 2023). We define a sequence as *foldable* if it satisfies: pLDDT > 80, pTM > 0.7, and pAE < 10. Entropy and diversity metrics are also computed to assess mode collapse see Appendix F.1 for further details.

**Results**. As shown in Table 2, applying P2 to DPLM significantly improves all folding metrics. Compared to DPLM with RDM sampling, P2 boosts pLDDT from 80.23 to 83.45 and foldability from 48.14% to 58.86%, while maintaining comparable entropy and diversity. These results confirm that P2 enhances generation quality without sacrificing diversity. Notably, DPLM + P2 outperforms all baselines—including EvoDiff, ESM3, and the 2.7B parameter ProGen2—with fewer parameters and better foldability. In Figure 2 we visualize the predicted 3D structures of generated proteins, showing visually coherent and plausible folds. Additional length-wise breakdowns and scaling ablations are included the appendix, and in particular in Table S4 and Figure S1.

## 4.2 LANGUAGE GENERATION

We next investigate the ability of P2 inference in language modeling tasks and evaluate on a suite of diverse including reading comprehension (TriviaQA (Joshi et al., 2017)), paragraph completion (LAMBADA (Paperno et al., 2016)), math reasoning (GSM8K (Cobbe et al., 2021)), story infilling (ROCStories (Mostafazadeh et al., 2016)), and code generation (HumanEval (Bavarian et al., 2022))—adopted from SMDM (Gong et al., 2025) and DiffuLLaMA (Nie et al., 2025). Additional experiments on modeling bidirectional relations, i.e. reverse curse behavior, are included in Appendix G.1.1. We apply P2 to two strong diffusion models: 1.) MDM (1.1B) and 2.) DiffuLLaMA (7B), and compare them to ancestral sampling. For P2, we sweep the stochasticity strength $\eta \in [0, 2.0]$ with step size 0.2, and report the best result per task in Table 3, with full experimental setup provided in Appendix F.2.

**Results**. We observe that P2 consistently improves generation quality across all five benchmarks, indicating improved global reasoning, fewer intermediate errors, and more coherent generations.

Table 3: Language generation benchmarks. Accuracy (%) is reported for TriviaQA, LAMBADA, and GSM8K; ROUGE-1/2/L for ROCStories; and pass@1 for HumanEval.

| Model | TriviaQA | LAMBADA | GSM8K | ROUGE-1/2/L | Code |
|---|---|---|---|---|---|
| GPT2-S (127M) | 4.0 | 25.9 | 44.8 | 7.8 / 0.8 / 7.4 | 1.6 |
| DiffuGPT-S (127M) | 2.0 | 45.0 | 50.2 | 13.7 / 1.4 / 12.6 | 0.3 |
| SEDD-S (170M) | 1.5 | 12.4 | 45.3 | 11.9 / 0.7 / 10.9 | 0.7 |
| GPT2-M (355M) | 6.7 | 37.7 | 50.7 | 8.6 / 0.9 / 8.2 | 2.6 |
| DiffuGPT-M (355M) | 3.8 | 60.5 | 52.6 | 18.7 / 2.7 / 17.0 | 2.9 |
| SEDD-M (424M) | 1.8 | 23.1 | 53.5 | 13.1 / 1.4 / 12.2 | 0.5 |
| Plaid1B (1.3B) | 1.2 | 8.6 | 32.6 | 12.1 / 1.1 / 11.2 | 0.1 |
| TinyLlama (1.1B) | – | 43.2 | – | – | – |
| GPT-2 (1.5B) | – | 44.6 | – | – | – |
| MDM (1.1B) | – | 52.7 | 58.5 | – | – |
| MDM + P2-self (ours) | **–** | **52.9** | **60.9** | **–** | **–** |
| LLaMA2 (7B) | 45.4 | 68.8 | 58.6 | 11.6 / 2.1 / 10.5 | 1.7 |
| DiffuLLaMA (7B) | 18.5 | 53.7 | – | 20.3 / 2.8 / 18.2 | 13.2 |
| DiffuLLaMA + P2-self (ours) | **18.8** | **54.8** | **–** | **25.4 / 7.1 / 23.4** | **17.6** |

On GSM8K, P2 lifts MDM performance from $58.5\%$ to $60.9\%$, surpassing the 7B autoregressive baseline LLaMA2 ($58.6\%$). Finally, on code generation, DiffuLLaMA with P2 achieves a $17.6\%$ pass@1, significantly outperforming both ancestral sampling ($13.2\%$) and LLaMA2 ($1.7\%$). On ROCStories, P2 boosts ROUGE-1/2/L scores by more than 5 absolute points.

Table 4: RNA sequence generation results. pLDDT and MFE measure structural quality, Entropy measures diversity, and GC content reflects biophysical realism.

| Source | pLDDT (↑) | MFE (↓) | Entropy (↑) | GC% (↑) |
|---|---|---|---|---|
| Native | 48.26 | -35.83 | **1.96** | 49.64 |
| RiNALMo-150M | 59.01 | -30.12 | 1.29 | 29.50 |
| RiNALMo-650M | 46.99 | -31.90 | 1.33 | 28.06 |
| MDM | 68.12 | -48.46 | 1.93 | 60.84 |
| MDM + P2-BERT (ours) | **73.28** | **-51.91** | 1.86 | **65.47** |

## 4.3 RNA SEQUENCE GENERATION

We evaluate P2 in the context of RNA generation, where biophysical plausibility is critical (see Appendix F.4 for training and evaluation details). A 150M-parameter MDM is trained on 27M sequences from RNACentral (Petrov, 2021). For evaluation, we follow the protein protocol and predict RNA structures using an external folding model (Shen et al., 2024), measuring pLDDT, minimum free energy (MFE), sequence entropy, and GC content. We generate 100 sequences of 100 base pairs each. As shown in Table 4, the MDM already surpasses RiNALMo baselines in structural quality and energy. Applying P2 with BERT-Planning (from RiNALMo-150M) further improves pLDDT ($68.1 \rightarrow 73.3$), lowers MFE ($-48.5 \rightarrow -51.9$), and increases GC content ($60.8\% \rightarrow 65.5\%$)—key indicators of biophysically plausible RNA. These gains come with only a small reduction in entropy.

## 4.4 ABLATION STUDIES

We conduct ablation studies to evaluate whether P2 improves performance across different domains and to understand how its variants compare to existing sampling strategies. We focus each ablation experiment on a specific domain and seek to answer the following key experimental questions:

**Q1: Does P2 outperform prior sampling strategies for protein sequence generation?** We compare P2 against common decoding strategies using a 150M MDM on protein generation ( Table 5). P2-Train (ours) achieves the highest pLDDT (83.45) and foldability (58.86%), outperforming RDM (Zheng et al., 2023) and Greedy Ancestral, MaskGIT (Chang et al., 2022b) and Top-K Marginal (Kim et al., 2025) by large margins. The performance gap further highlights that the design choices made in P2 which differentiate it from the related baselines, such as MaskGIT (Chang et al., 2022b) and Top-K Marginal (Kim et al., 2025), play a crucial role in real-world applications. P2-Self and P2-Bert also yield consistent gains, while P2-Train with an additionally post-trained planner exhibits the best performance, validating that planner-based sampling significantly enhances structural quality.

**Q2: Can P2 improve generative fluency and accuracy in code and story infilling tasks?**

Table 5: Protein sequence generation: comparison of sampling strategies.

| Method | pLDDT (↑) | pTM (↑) | pAE (↓) | Foldability (%) (↑) | Entropy (↑) | Diversity (%) (↑) |
|---|---|---|---|---|---|---|
| Vanilla Ancestral | 54.11 | 0.43 | 19.96 | 6.29 | **3.90** | **93.28** |
| Greedy Ancestral | 63.69 | 0.51 | 17.50 | 13.00 | 3.83 | 93.03 |
| DFM Sampling | 63.20 | 0.41 | 19.90 | 17.00 | 2.85 | 91.36 |
| RDM Sampling | 78.79 | 0.65 | 12.13 | 48.57 | 3.11 | 92.70 |
| TopK-Marginal | 55.46 | 0.32 | 22.03 | 10.86 | 2.10 | 92.45 |
| P2-Self (ours) | 80.98 | 0.68 | 11.43 | 49.86 | 3.25 | 92.63 |
| P2-Bert (ours) | 70.80 | 0.51 | 16.09 | 35.43 | 2.36 | 90.66 |
| P2-Train (ours) | **83.45** | **0.72** | **10.15** | **58.86** | 3.35 | 92.69 |

Using a 7B DiffuLLaMA model, we assess generation quality in HumanEval and ROCStories benchmarks (Table 6). We find our P2-Self model achieves the highest pass@1 and ROUGE scores, outperforming both ancestral decoding and RDM.

Table 6: Language generation ablation: code generation (HumanEval) and story infilling (ROCStories).

| Method | pass@1↑ | ROUGE-1↑ | ROUGE-2↑ | ROUGE-L↑ |
|---|---|---|---|---|
| Vanilla Ancestral | 0.121 | 17.18 | 2.72 | 15.57 |
| Greedy Ancestral | 0.161 | 24.68 | 7.12 | 22.85 |
| DFM Sampling | 0.116 | 16.62 | 2.42 | 15.23 |
| RDM Sampling | 0.132 | 20.31 | 2.83 | 18.16 |
| **P2-Self (ours)** | **0.180** | **25.27** | **7.36** | **23.25** |

**Q3: Does P2 improve structural quality in RNA generation while maintaining diversity?**

Table 7 shows that P2-Bert (ours) improves pLDDT and MFE while preserving GC content and entropy. This indicates that P2 remains effective across biomolecular domains, even when transferring planners pretrained on different modalities.

**Summary.** P2 generalizes and improves upon all major masked diffusion sampling strategies. With its flexible decoding design, P2 can subsume Vanilla, Greedy, RDM, and DFM via appropriate planner configurations. Its variants—P2-Self, P2-Bert, and P2-Train—not only retain diversity but also unlock substantial gains in structural and functional accuracy across domains.

Table 7: RNA sequence generation ablation.

| Method | pLDDT↑ | MFE↓ | Entropy↑ | GC (%)↑ |
|---|---|---|---|---|
| Vanilla Ancestral | 68.12 | −48.46 | **1.93** | 60.84 |
| Greedy Ancestral | 37.41 | −32.32 | 1.66 | 49.27 |
| DFM Sampling | 33.17 | −26.32 | 1.93 | 49.23 |
| RDM Sampling | 67.35 | −47.54 | 1.89 | 59.42 |
| P2-Self (ours) | 69.41 | −48.21 | 1.89 | 59.84 |
| **P2-Bert (ours)** | **73.28** | **−51.91** | 1.86 | **65.47** |

Additional ablations, including the effects of stochasticity $\eta$ (Figure S2, Figure S5) and planner scale (Figure S3, Figure S6, Table S5), are provided in §G.2.3. In Table S6, we compare ELBO values between $G_\phi$ and show that self-planning often outperforms BERT-based planning due to a better fit with the underlying denoiser. Further appendix results include analysis on short protein sequences ( Table S8), comparisons with baseline ESM2 ( Table S9), a comparison with Top-K Marginal (Kim et al., 2025) ( Table S10), and a robustness study reporting variance across runs ( Table S11).

## 4.5 INFERENCE-TIME SCALING AND COMPUTATIONAL COMPLEXITY

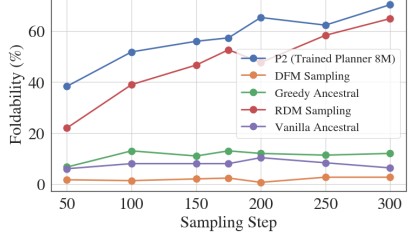

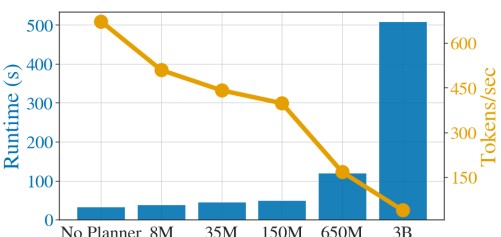

Figure 3: Inference-time scaling: Foldability vs. Sampling steps.

Figure 4: Runtime (bar) and throughput (line) for different planner sizes (150M denoiser on an A100).

A key strength of Path Planning (P2) is its resampling-based decoding mechanism, which allows flexible control over generation fidelity by varying the number of sampling steps. We evaluate P2 (Trained Planner, 8M) on protein sequence generation with varying sampling steps: $\{50, 100, 150, 200, 250, 300\}$, generating 300 sequences of length 200 for each setting. As shown

in Figure 3, P2 consistently improves foldability with increased sampling steps and maintains its advantage beyond 200 steps, where other methods plateau.

**Computational complexity**. P2 offers a tunable tradeoff between sampling quality and runtime, depending on planner size. In Figure 4, we compare sampling speed across various planner models using a 150M denoiser on a single NVIDIA A100 GPU. All baseline strategies—including Vanilla, Greedy, DFM, RDM, and P2-Self—share a common "No Planner" runtime profile, yielding the highest throughput of 673.16 tokens/sec. Introducing an external planner naturally incurs additional cost. However, the 8M P2 planner—used in all protein experiments—maintains a strong balance, achieving 509.55 tokens/sec with only a $24\%$ overhead.

## 5 RELATED WORK

Masked diffusion language models (MDMs) have emerged as promising alternatives to autoregressive models for discrete generation (Sahoo et al., 2024; Shi et al., 2024; Nie et al., 2025; Gong et al., 2025). To improve sampling, several heuristic methods—greedy unmasking (Gong et al., 2025), remasking (Zheng et al., 2023; Wang et al., 2024), and informed correctors (Zhao et al., 2024b)—have been proposed, though they lack structured guidance. Order-based strategies from Any-Order Autoregressive Models (AOARMs) (Li et al., 2021; Shih et al., 2022) enable greater flexibility but often require costly planners or fixed schedules. DDPD (Liu et al., 2024) separates planning and denoising, but operates on uniform diffusion without mask-awareness. In contrast, our P2 sampler introduces a lightweight, modular mechanism for dynamic, mask-aware planning compatible with frozen denoisers. A detailed comparison with DDPD is provided in §D.4.

Recent work has sought to improve generation order in MDMs. ReMDM-conf (Wang et al., 2025a) schedules the temperature of the Gibbs distribution used in confidence-based informed correctors, while Kim et al. (2025) propose a Top-K heuristic based on local confidence gaps. Our path planning framework generalizes such strategies within a unified, optimizable, and principled formulation. Closest to our approach is LO-ARM (Wang et al., 2025c), which treats generation order as a latent variable and learns it via REINFORCE. However, its reliance on high-variance policy gradients limits scalability. Our method instead offers a simple, differentiable ELBO objective, enabling efficient and scalable learning of generation policies.

## 6 CONCLUSION

We demonstrate that unmasking order significantly impacts the generative performance of masked diffusion language models (MDMs). By expanding the ELBO formulation, we introduce a *planner* that optimizes token selection during inference. We propose *Path Planning (P2)*, a sampling framework that generalizes all existing MDM sampling strategies. P2 delivers state-of-the-art improvements across diverse tasks, including language generation and biological sequence design, enabling MDMs to outperform larger autoregressive models. Our findings highlight the importance of inference strategies in discrete diffusion models, paving the way for more efficient and effective sequence generation.

## ETHICS STATEMENT

This work investigates improvements to discrete diffusion models for generative modeling across text, code, and biological sequences. While our method, Path Planning (P2), demonstrates significant gains in generative quality, we recognize the potential for both positive and negative downstream impacts.

On the positive side, more effective discrete generative models can advance research in reasoning, programming, and biomolecular design. In particular, applications in protein and RNA sequence modeling may accelerate scientific discovery and therapeutic design. However, these same capabilities could also be misused, for instance in generating harmful or dual-use biological sequences. To mitigate this, all biological experiments in this work are purely computational and evaluated against standard, publicly available benchmarks; no wet-lab synthesis or functional validation was performed. We explicitly discourage and do not support the malicious application of our methods.

All datasets used are publicly available and widely adopted in the community. We have not introduced new data that could expose private or sensitive information. Our models are trained and released in accordance with open-science practices, but with careful documentation of intended use and limitations to discourage misuse.

## 7 REPRODUCIBILITY STATEMENT

We provide the PyTorch implementation in Appendix E. For the experiments, we integrate our approach into the SMDM (Gong et al., 2025) GitHub codebase[1] to obtain the results for "MDM (1.1B) + P2" reported in Table 3. Similarly, the results for "DiffuLLaMA (7B) + P2" in Table 3 are derived using the DiffuLLaMA (Nie et al., 2025) GitHub codebase[2]. For the protein sequence generation experiments, we utilize the DPLM (Wang et al., 2024) open-source codebase[3]. The RNA sequence generation results are obtained by adapting the DPLM codebase for MDM training, combined with the RiNALMo (Penić et al., 2024) language model architecture.

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

APPENDIX

# A  RELATED WORKS: EXTENDED DISCUSSION

Masked diffusion language models (MDMs) represent a promising alternative to autoregressive models for discrete data generation, particularly in language modeling. Recent advancements have focused on simplifying and generalizing the MDM framework to improve performance and training efficiency (Shi et al., 2024; Sahoo et al., 2024). These studies introduced a continuous-time variational objective for MDMs, expressed as a weighted integral of cross-entropy losses, facilitating the training of models with state-dependent masking schedules. At the GPT-2 scale, these MDMs outperformed prior diffusion-based language models and demonstrated superior capabilities in zero-shot language modeling tasks (Nie et al., 2025; Gong et al., 2025).

MDMs generate sequences starting from a fully masked input and progressively unmasking positions until a clean sequence is reached. Once a token is unmasked, it will stay unchanged. However, there is not guarantee that the state is correct, considering the approximation errors arise from the imperfect fit to real-world data distributions. Additionally, time discretization (Zhao et al., 2024b) and numerical errors (Zheng et al., 2025) may further the error incurred during sampling processes.

To address these challenges, several solutions have been proposed. These include methods allowing models to revise prior predictions and guiding sampling trajectories using internal or external knowledge. Examples include informed correctors (Zhao et al., 2024b), greedy ancestral methods (Gong et al., 2025), and RDM sampling techniques (Zheng et al., 2023; Wang et al., 2024), which leverage model scores to replace random masking with targeted corrections. None of these works, however, allow for the use of an external planner, and (Zheng et al., 2023; Wang et al., 2024) are simply using a top-k sampling strategy without any concern for the theoretical underpinnings of the sampling strategies viability.

In terms of theoretically-backed methods for selecting the denoising order during a generative model's sampling process, the current literature is quite sparse. Shih et al. (2022); Li et al. (2021) discuss this task from the perspective of Any-Order Autoregressive models, with Li et al. (2021) requiring a specially-trained external planner model using a specially designed architecture and Shih et al. (2022) taking the perspective that a fixed family of possible generation orders should be chosen a priori to eliminate redundancy.

The most closely related work to ours is likely the recent DDPD (Liu et al., 2024) introduced a generative process divided into a planner, which identifies corrupted positions, and a denoiser, which refines these positions. Though they discuss the ability to employ a MDM denoiser within their framework, their analysis and sampling is through the lens of uniform discrete diffusion models. In particular, as with Li et al. (2021), the success of their strategy is contingent upon training a large specialized planner model of comparable size to the denoiser itself. Moreover, in their framework, since they are based on uniform diffusion models, the partially de-noised sequence never contains any masked states, and there is no way for the planner to be separated into masked and unmasked components to design a sampling strategy with guaranteed finite-time along the lines of our Algorithm 1. Given the possible perceived similarity of this work with ours, we provide a thorough comparison of DDPD with P2 in Algorithm 4, highlighting the greater flexibility and difference in role of P2s' planners.

# B  PROOFS OF PROPOSITIONS 1 AND 2

## B.1  PROOF OF PROPOSITION 1: TIME DISCRETIZATION APPROACH

In this section we provide a self-contained proof of Proposition 1, using directly a lower bound for the time-discretized, coordinate-wise conditionally independent dynamics Eq. 5. We refer the reader interested in a direct and more concise proof of Proposition 1 using the theory of continuous time Markov chains (see Section C for the definition and basic theory thereof) to Section D.

**Proposition 1.** *Define $P_0^{\theta,\phi} \in \Delta^{d^L}$ by $P_0^{\theta,\phi}(\mathbf{x}) = \mathbb{P}(X_0^{\theta,\phi} = \mathbf{x})$, where $X^{\theta,\phi}$ is the continuous time Markov chain resulting from sending $T \to \infty$ in the discrete-time P2 formulation Eq. 5. Then we have an "Evidence Based Lower Bound" $\mathcal{E}(\mathbf{x}_0) \leq \log(P_0^{\theta,\phi}(\mathbf{x}_0))$ for each fixed $\mathbf{x}_0 \in \mathcal{V}^L$ given by*

$$\mathcal{E}(\mathbf{x}_0) = \mathcal{E}_{MP}(\mathbf{x}_0) + \mathcal{E}_{UP}(\mathbf{x}_0) + \mathcal{E}_D(\mathbf{x}_0), \text{ where:}$$

$$\mathcal{E}_{MP}(\mathbf{x}_0) = -\int_0^1 \frac{d\alpha_t}{dt} \cdot \frac{1}{1-\alpha_t} \mathbb{E}_{\mathbf{x}_t \sim \mathbf{P}_t(\cdot;\mathbf{x}_0)} \left[ \sum_{i=1,\mathbf{x}_t^i=\mathbf{m}}^{L} \mathbb{E}_{\mathbf{z} \sim D_\theta(\mathbf{x}_t)} \left[ CE\left( Cat(z^i; \delta(x_0^i)), G_M^i(\mathbf{z}, \mathbf{x}_t) \right) \right] \right] dt$$

$$\mathcal{E}_{UP}(\mathbf{x}_0) = -\int_0^1 \frac{d\alpha_t}{dt} \cdot \frac{1}{1-\alpha_t} \mathbb{E}_{\mathbf{x}_t \sim \mathbf{P}_t(\cdot;\mathbf{x}_0)} \left[ \sum_{i=1,\mathbf{x}_t^i \neq \mathbf{m}}^{L} \mathbb{E}_{\mathbf{z} \sim D_\theta(\mathbf{x}_t)} \left[ CE\left( Cat(z^i; \delta(x_0^i)), G_U^i(\mathbf{z}, \mathbf{x}_t) \right) \right] \right] dt$$

$$\mathcal{E}_D(\mathbf{x}_0) = -\int_0^1 \frac{d\alpha_t}{dt} \cdot \frac{1}{1-\alpha_t} \mathbb{E}_{\mathbf{x}_t \sim \mathbf{P}_t(\cdot;\mathbf{x}_0)} \left[ \sum_{i=1,\mathbf{x}_t^i=\mathbf{m}}^{L} \delta(\mathbf{x}_0^i)^\top \log(D_\theta^i(\mathbf{x}_t)) \right] dt.$$

*Here $\mathbf{p}_t$ is defined per Eq. 1 and $CE(a,b) = a\log(b) + (1-a)\log(1-b)$ for $a,b \in [0,1]$, with $0\log 0 = 0$.*

Consider $\mathbf{q}_\theta^T$ the distribution on $\mathcal{V}^L$ resulting from iteratively sampling independently in each coordinate according to Eq. 5, with initial data $(\mathbf{m}, \dots, \mathbf{m})$. Our starting point is the standard standard ELBO in discrete time used for diffusion models Sohl-Dickstein et al. (2015). That is, fixing $\mathbf{x}_T = (\mathbf{m}, \dots, \mathbf{m})$, we let $\mathbf{q}_{t,\theta}^T(\mathbf{x}_{t-1}|\mathbf{x}_t)$ be the one-step transition probabilities describing our time-discretized sampling scheme on $\mathcal{V}^L$:

$$\mathbf{q}_{t,\theta}^T(\mathbf{x}_{t-1}|\mathbf{x}_t) = \mathbb{E}_{\mathbf{z} \sim D_\theta(\mathbf{x}_t)} \left[ \mathbf{q}_{t,\theta}^T(\mathbf{x}_{t-1}|\mathbf{x}_t, \mathbf{z}) \right] \tag{8}$$

$$\mathbf{q}_{t,\theta}^T(\mathbf{x}_{t-1}|\mathbf{x}_t, \mathbf{z}) = \prod_{i=1}^{L} q_{t,\theta}(x_{t-1}^i|\mathbf{x}_t, \mathbf{z}),$$

with $q_{t,\theta}(x_{t-1}^i|\mathbf{x}_t, \mathbf{z})$ as in Eq. 5. Note that this follows immediately from the assumed conditional independence and marginalizing over the independent samples $\mathbf{z}$. We also let $\mathbf{q}_t^T(\mathbf{x}_{t-1}|\mathbf{x}_t, \mathbf{x}_0)$ be the one-step transitions for the reference reverse process on $\mathcal{V}^L$ given by

$$\mathbf{q}_t^T(\mathbf{x}_{t-1}|\mathbf{x}_t, \mathbf{x}_0) = \prod_{i=1}^{L} q_t(x_{t-1}^i|x_t^i, x_0^i). \tag{9}$$

for $q(x_{t-1}^i|x_t^i, x_0^i)$ as in Eq. 2.

Then the discrete time ELBO reads:

$$\log(\mathbf{q}_\theta^T(\mathbf{x}_0) \geq -\sum_{t=T}^{2} \sum_{\mathbf{x}_t \in \mathcal{V}^L} \hat{\mathbf{q}}_t^T(\mathbf{x}_t|\mathbf{x}_0) \sum_{\mathbf{x}_{t-1} \in \mathcal{V}^L} \mathbf{q}_t^T(\mathbf{x}_{t-1}|\mathbf{x}_t, \mathbf{x}_0) \log\left( \frac{\mathbf{q}_t^T(\mathbf{x}_{t-1}|\mathbf{x}_t, \mathbf{x}_0)}{\mathbf{q}_{t,\theta}^T(\mathbf{x}_{t-1}|\mathbf{x}_t)} \right) \tag{10}$$

$$+ \sum_{\mathbf{x}_1 \in \mathcal{V}^L} \hat{\mathbf{q}}_1^T(\mathbf{x}_1|\mathbf{x}_0) \log(\mathbf{q}_{1,\theta}^T(\mathbf{x}_0|\mathbf{x}_1)),$$

where $\hat{\mathbf{q}}_t^T(\mathbf{x}_t|\mathbf{x}_0)$ is the distribution at timestep $t$ of the backward dynamics with transition probabilities Eq. 9 with initial data $\mathbf{x}_T = (\mathbf{m}, \dots, \mathbf{m})$. For reference, we include a derivation of this result in the following subsection B.5.

We will use Eq. 10 to show $\lim_{T \to \infty} \log(\mathbf{q}_\theta^T(\mathbf{x}_0) \geq \mathcal{E}(\mathbf{x}_0)$ from Proposition 1.

In the proof, we will make use of the following Lemma, the proof of which is delayed to subsection B.4.

**Lemma 1.** *For $\mathbf{x} \neq \mathbf{y} \in \mathcal{V}^L$, $\mathbf{x}_0 \in \mathcal{V}^L$, and setting $t = \lfloor sT \rfloor$ for fixed $s \in (0,1)$, we have the following limits:*

$$\lim_{T \to \infty} \hat{\mathbf{q}}_t^T(\mathbf{x}|\mathbf{x}_0) = \mathbf{p}_{1-s}(\mathbf{x}|\mathbf{x}_0), \tag{11}$$

$$\lim_{T \to \infty} \hat{\mathbf{q}}_1^T(\mathbf{x}_0|\mathbf{x}_0) = 1 \tag{12}$$

$$\lim_{T \to \infty} T\mathbf{q}_t^T(\mathbf{y}|\mathbf{x}, \mathbf{x}_0) \tag{13}$$

$$= \begin{cases} -\frac{\alpha'_{1-s}}{1-\alpha_{1-s}} Cat\left(y^i; \delta(x_0^i)\right) Cat\left(x^i; \delta(\mathbf{m})\right) & , d_{HAM}(\mathbf{x}, \mathbf{y}) = 1, x^i \neq y^i \\ 0 & , \text{otherwise} \end{cases}$$

$$\lim_{T \to \infty} T\mathbf{q}_{t,\theta}^T(\mathbf{y}|\mathbf{x}) \tag{14}$$

$$= -\frac{\alpha'_{1-s}}{1-\alpha_{1-s}} \begin{cases} Cat(y^i; D_\theta^i(\mathbf{x}))\mathbb{E}_{\mathbf{z} \sim D_\theta(\mathbf{x})}\left[G_\phi^i(\mathbf{z}^{-i,y^i}, \mathbf{x})\right] & , d_{HAM}(\mathbf{x}, \mathbf{y}) = 1, x^i \neq y^i, x^i = \mathbf{m} \\ \frac{\mathbb{E}_{\mathbf{z} \sim D_\theta(\mathbf{x})}\left[G_\phi^i(\mathbf{z}, \mathbf{x})\right]}{1 - Cat(x^i; D_\theta^i(\bar{\mathbf{x}}))} Cat(y^i; D_\theta^i(\bar{\mathbf{x}})) & , d_{HAM}(\mathbf{x}, \mathbf{y}) = 1, x^i \neq y^i, x^i \neq \mathbf{m} \\ 0, & \text{otherwise} \end{cases},$$

*where here we recall $d_{HAM}$ refers to the Hamming distance, and the notation $\mathbf{z}^{-i,y^i}$ means replacing the $i$'th coordinate of $\mathbf{z}$ with $y^i$.*

We now proceed with the proof of Proposition 1.

*Proof.* First we observe that, from Eq. 12, in the limit as $T \to \infty$, the reconstruction loss - i.e. the second term in Eq. 10 - vanishes. We thus turn our attention to bounding the first term.

We observe first that we are seeking to find

$$\lim_{T \to \infty} \sum_{t=T}^{2} f^T(t) = \lim_{T \to \infty} \frac{1}{T} \sum_{t=T}^{2} Tf^T(t)$$

for a sequence of functions $f^T : \{2, \dots, T\} \to \mathbb{R}$, with $f^T(t) = g^T(t/T)$ for $g^T : [0,1] \to \mathbb{R}$. With the uniform integrability of $Tg^T$, this converges to the Riemann integral

$$\int_0^1 \lim_{T \to \infty} Tf^T(\lfloor Ts \rfloor)ds. \tag{15}$$

We identified the limit of $\hat{\mathbf{q}}$ for the outermost sum in Eq. 10 in Eq. 11. Now we turn to finding:

$$\lim_{T \to \infty} -T \sum_{\mathbf{x}_{t-1} \in \mathcal{V}^L} \mathbf{q}_t^T(\mathbf{x}_{t-1}|\mathbf{x}_t, \mathbf{x}_0) \log\left(\frac{\mathbf{q}_t^T(\mathbf{x}_{t-1}|\mathbf{x}_t, \mathbf{x}_0)}{\mathbf{q}_{t,\theta}^T(\mathbf{x}_{t-1}|\mathbf{x}_t)}\right), t = \lfloor sT \rfloor,$$

after which we will use Eq. 15 and Eq. 11 to replace the outermost sum over times in Eq. 9 with an integral, the sum over $\mathbf{x}_t$ with the expected value against $\mathbf{p}_{1-s}$, and the terms in the integrand with the above limit.

We treat this as two terms:

$$E_1^\infty(\mathbf{x}_0, \mathbf{x}, s) = \sum_{\mathbf{y} \in \mathcal{V}^L, \mathbf{y} \neq \mathbf{x}} \lim_{T \to \infty} -T\mathbf{q}_t^T(\mathbf{y}|\mathbf{x}, \mathbf{x}_0) \log\left(\frac{\mathbf{q}_t^T(\mathbf{y}|\mathbf{x}, \mathbf{x}_0)}{\mathbf{q}_{t,\theta}^T(\mathbf{y}|\mathbf{x})}\right), \quad t = \lfloor sT \rfloor \tag{16}$$

and

$$E_2^\infty(\mathbf{x}_0, \mathbf{x}, s) = \lim_{T \to \infty} -T\mathbf{q}_t^T(\mathbf{x}|\mathbf{x}, \mathbf{x}_0) \log\left(\frac{\mathbf{q}_t^T(\mathbf{x}|\mathbf{x}, \mathbf{x}_0)}{\mathbf{q}_{t,\theta}^T(\mathbf{x}|\mathbf{x})}\right), \quad t = \lfloor sT \rfloor. \tag{17}$$

We begin with Eq. 16. Using for $\mathbf{x} \neq \mathbf{y} \in \mathcal{V}^L, t = \lfloor sT \rfloor$

$$\lim_{T \to \infty} -T\mathbf{q}_t^T(\mathbf{y}|\mathbf{x}, \mathbf{x}_0) \log\left(\frac{\mathbf{q}_t^T(\mathbf{y}|\mathbf{x}, \mathbf{x}_0)}{\mathbf{q}_{t,\theta}^T(\mathbf{y}|\mathbf{x})}\right)$$

$$= \lim_{T \to \infty} -T\mathbf{q}_t^T(\mathbf{y}|\mathbf{x}, \mathbf{x}_0) \log\left(\frac{\lim_{T \to \infty} T\mathbf{q}_t^T(\mathbf{y}|\mathbf{x}, \mathbf{x}_0)}{\lim_{T \to \infty} T\mathbf{q}_{t,\theta}^T(\mathbf{y}|\mathbf{x})}\right),$$

where we interpret $0 \log 0 = 0$, and Eq. 13, Eq. 14, every term in the sum becomes 0 when $d_{\mathrm{HAM}}(\mathbf{x}, \mathbf{y}) > 1$, $y^i \neq x_0^i$, or $x^i \neq \mathbf{m}$, and we arrive at:

$$E_1^\infty(\mathbf{x}_0, \mathbf{x}, s) = \frac{\alpha'_{1-s}}{1 - \alpha_{1-s}} \sum_{i=1, x^i=\mathbf{m}}^{L} \log \left( \frac{1}{\mathrm{Cat}(x_0^i; D_\theta^i(\mathbf{x})) \mathbb{E}_{\mathbf{z} \sim D_\theta(\mathbf{x})} \left[ G_\phi^i(\mathbf{z}^{-i, x_0^i}, \mathbf{x}) \right]} \right) \qquad (18)$$

$$= -\frac{\alpha'_{1-s}}{1 - \alpha_{1-s}} \sum_{i=1, x^i=\mathbf{m}}^{L} \log \left( \mathrm{Cat}(x_0^i; D_\theta^i(\mathbf{x})) \right)$$

$$- \frac{\alpha'_{1-s}}{1 - \alpha_{1-s}} \sum_{i=1, x^i=\mathbf{m}}^{L} \log \left( \mathbb{E}_{\mathbf{z} \sim D_\theta(\mathbf{x})} \left[ G_\phi^i(\mathbf{z}^{-i, x_0^i}, \mathbf{x}) \right] \right).$$

We now turn our attention to the limit Eq. 17. We first observe that, from Eq. 2, for $x^i \in \mathcal{V}$, $t = \lfloor Ts \rfloor$, $\lim_{T \to \infty} q_t(x^i | x^i, x_0^i) = 1$, so by definition of $\mathbf{q}_t^T$ from Eq. 9, $\lim_{T \to \infty} \mathbf{q}_t^T(\mathbf{x} | \mathbf{x}, \mathbf{x}_0) = 1$. Then:

$$E_2^\infty(\mathbf{x}_0, \mathbf{x}, s) = \lim_{T \to \infty} -T \log \left( \frac{\mathbf{q}_t^T(\mathbf{x} | \mathbf{x}, \mathbf{x}_0)}{\mathbf{q}_{t,\theta}^T(\mathbf{x} | \mathbf{x})} \right)$$

$$= \lim_{T \to \infty} -T \log \left( 1 - \sum_{\mathbf{y} \neq \mathbf{x}} \mathbf{q}_t^T(\mathbf{y} | \mathbf{x}, \mathbf{x}_0) \right) + \lim_{T \to \infty} T \log \left( 1 - \sum_{\mathbf{y} \neq \mathbf{x}} \mathbf{q}_{t,\theta}^T(\mathbf{y} | \mathbf{x}) \right)$$

$$= \sum_{\mathbf{y} \neq \mathbf{x}} \lim_{T \to \infty} T \mathbf{q}_t^T(\mathbf{y} | \mathbf{x}, \mathbf{x}_0) - \sum_{\mathbf{y} \neq \mathbf{x}} \lim_{T \to \infty} T \mathbf{q}_{t,\theta}^T(\mathbf{y} | \mathbf{x}), \quad t = \lfloor Ts \rfloor$$

where we used standard log asymptotics.

Inserting now Eq. 13 and Eq. 14, once again any terms in the sum so that $d_{\mathrm{HAM}}(\mathbf{x}, \mathbf{y}) > 1$ vanish, and we arrive at

$$E_2^\infty(\mathbf{x}_0, \mathbf{x}, s) = -\frac{\alpha'_{1-s}}{1 - \alpha_{1-s}} \sum_{i=1, x^i=m}^{L} \sum_{y^i \in \mathcal{V}, y^i \neq \mathbf{m}} \mathrm{Cat}(y^i; \delta(x_0^i) \qquad (19)$$

$$+ \frac{\alpha'_{1-s}}{1 - \alpha_{1-s}} \sum_{i=1, x^i=m}^{L} \sum_{y^i \in \mathcal{V}, y^i \neq \mathbf{m}} \mathrm{Cat}(y^i; D_\theta^i(\mathbf{x})) \mathbb{E}_{\mathbf{z} \sim D_\theta(\mathbf{x})} \left[ G_\phi^i(\mathbf{z}^{-i, y^i}, \mathbf{x}) \right]$$

$$+ \frac{\alpha'_{1-s}}{1 - \alpha_{1-s}} \sum_{i=1, x^i \neq m}^{L} \sum_{y^i \in \mathcal{V}, y^i \neq x^i} \frac{\mathbb{E}_{\mathbf{z} \sim D_\theta(\mathbf{x})} \left[ G_\phi^i(\mathbf{z}, \mathbf{x}) \right]}{1 - \mathrm{Cat}(x^i; D_\theta^i(\bar{\mathbf{x}})} \mathrm{Cat}(y^i; D_\theta^i(\bar{\mathbf{x}}))$$

$$= -\frac{\alpha'_{1-s}}{1 - \alpha_{1-s}} \sum_{i=1, x^i=m}^{L} \left( 1 - \mathbb{E}_{\mathbf{z} \sim D_\theta(\mathbf{x})} \left[ G_\phi^i(\mathbf{z}, \mathbf{x}) \right] \right)$$

$$- \frac{\alpha'_{1-s}}{1 - \alpha_{1-s}} \sum_{i=1, x^i \neq m}^{L} -\mathbb{E}_{\mathbf{z} \sim D_\theta(\mathbf{x})} \left[ G_\phi^i(\mathbf{z}, \mathbf{x}) \right],$$

where in the second step we recall $\mathbf{z}^{-i, y^i}$ is denoting replacing the $i$'th coordinate of $\mathbf{z}$ with $y^i$, so the sum in the second term is just taking the expected value in the missing coordinate, and the law of total probability applied to $D_\theta^i(\mathbf{x})$ cancels the denominator in the third term.

Recalling we are finding the limit as $T \to \infty$ of the right hand side of Eq. 10, using the limits Eq. 11, Eq. 18, Eq. 19 with the observation Eq. 15, we arrive at:

$$\lim_{T \to \infty} \log(\mathbf{q}_\theta^T(\mathbf{x}_0) \geq \int_0^1 \mathbb{E}_{\mathbf{x}_s \sim \mathbf{p}_{1-s}(\cdot|\mathbf{x}_0)} \left[ E_1^\infty(\mathbf{x}_0, \mathbf{x}_s, s) + E_2^\infty(\mathbf{x}_0, \mathbf{x}_s, s) \right] ds \qquad (20)$$

$$= - \int_0^1 \frac{\alpha'_{1-s}}{1 - \alpha_{1-s}} \mathbb{E}_{\mathbf{x}_s \sim \mathbf{p}_{1-s}(\cdot|\mathbf{x}_0)} \left[ \sum_{i=1,x_s^i=\mathbf{m}}^L \log \left( \mathrm{Cat}(x_0^i; D_\theta^i(\mathbf{x}_s)) \right) \right.$$

$$+ \sum_{i=1,x_s^i=\mathbf{m}}^L \log \left( \mathbb{E}_{\mathbf{z} \sim D_\theta(\mathbf{x}_s)} \left[ G_\phi^i(\mathbf{z}^{-i,x_0^i}, \mathbf{x}_s) \right] \right)$$

$$+ \sum_{i=1,x_s^i=\mathbf{m}}^L \left( 1 - \mathbb{E}_{\mathbf{z} \sim D_\theta(\mathbf{x})} \left[ G_\phi^i(\mathbf{z}, \mathbf{x}_s) \right] \right)$$

$$\left. + \sum_{i=1,x_s^i \neq m}^L -\mathbb{E}_{\mathbf{z} \sim D_\theta(\mathbf{x}_s)} \left[ G_\phi^i(\mathbf{z}, \mathbf{x}_s) \right] \right] ds.$$

We handle the 4 terms in Eq. 20 separately. For the first, we observe that making the time change $t = 1 - s$, this is $\mathcal{E}_D(\mathbf{x}_0)$ from Proposition 1.

For the second, we recall $\alpha$ is decreasing, so the time-dependent term in front is positive. Thus, by Jensen's inequality:

$$- \int_0^1 \frac{\alpha'_{1-s}}{1 - \alpha_{1-s}} \mathbb{E}_{\mathbf{x}_s \sim \mathbf{p}_{1-s}(\cdot|\mathbf{x}_0)} \left[ \sum_{i=1,x_s^i=\mathbf{m}}^L \log \left( \mathbb{E}_{\mathbf{z} \sim D_\theta(\mathbf{x}_s)} \left[ G_\phi^i(\mathbf{z}^{-i,x_0^i}, \mathbf{x}_s) \right] \right) \right] ds$$

$$\geq - \int_0^1 \frac{\alpha'_{1-s}}{1 - \alpha_{1-s}} \mathbb{E}_{\mathbf{x}_s \sim \mathbf{p}_{1-s}(\cdot|\mathbf{x}_0)} \left[ \sum_{i=1,x_s^i=\mathbf{m}}^L \mathbb{E}_{\mathbf{z} \sim D_\theta(\mathbf{x}_s)} \left[ \log \left( G_\phi^i(\mathbf{z}^{-i,x_0^i}, \mathbf{x}_s) \right) \right] \right] ds$$

$$= - \int_0^1 \frac{\alpha'_{1-s}}{1 - \alpha_{1-s}} \mathbb{E}_{\mathbf{x}_s \sim \mathbf{p}_{1-s}(\cdot|\mathbf{x}_0)} \left[ \sum_{i=1,x_s^i=\mathbf{m}}^L \mathbb{E}_{\mathbf{z} \sim D_\theta(\mathbf{x}_s)} \left[ \mathrm{Cat}(z^i; \delta(x_0^i)) \log \left( G_\phi^i(\mathbf{z}, \mathbf{x}_s) \right) \right] \right] ds$$

$$\geq - \int_0^1 \frac{\alpha'_{1-s}}{1 - \alpha_{1-s}} \mathbb{E}_{\mathbf{x}_s \sim \mathbf{p}_{1-s}(\cdot|\mathbf{x}_0)} \left[ \sum_{i=1,x_s^i=\mathbf{m}}^L \mathbb{E}_{\mathbf{z} \sim D_\theta(\mathbf{x}_s)} \left[ \mathrm{CE} \left( \mathrm{Cat}(z^i; \delta(x_0^i)), G_\phi^i(\mathbf{z}, \mathbf{x}_s) \right) \right] \right] ds$$

$$= \mathcal{E}_{MP}(\mathbf{x}_0)$$

where $\mathcal{E}_{MP}(\mathbf{x}_0)$ is as in Proposition 1. The second inequality comes from the fact that $a \log(b) \geq \mathrm{CE}(a, b) = a \log(b) + (1 - a) \log(1 - b)$ for $a, b \in [0, 1]$. To see this final equality, we again make the time change $t = 1 - s$ and recall that we define $G_M^i(\mathbf{z}, \mathbf{x}) = G_\phi^i(\mathbf{z}, \mathbf{x})$ when $x^i = \mathbf{m}$.

For the third term, the second term is already training the planner in masked positions, so we simply use that $G_\phi^i \in [0, 1]$ to bound this below by 0.

Finally, for the last term, we use that $-a \geq \log(1 - a)$ for any $a \in [0, 1)$, so:

$$- \int_0^1 \frac{\alpha'_{1-s}}{1 - \alpha_{1-s}} \mathbb{E}_{\mathbf{x}_s \sim \mathbf{p}_{1-s}(\cdot|\mathbf{x}_0)} \left[ \sum_{i=1,x_s^i \neq \mathbf{m}}^L -\mathbb{E}_{\mathbf{z} \sim D_\theta(\mathbf{x}_s)} \left[ G_\phi^i(\mathbf{z}, \mathbf{x}_s) \right] \right] ds$$

$$\geq - \int_0^1 \frac{\alpha'_{1-s}}{1 - \alpha_{1-s}} \mathbb{E}_{\mathbf{x}_s \sim \mathbf{p}_{1-s}(\cdot|\mathbf{x}_0)} \left[ \sum_{i=1,x_s^i \neq \mathbf{m}}^L \log \left( \mathbb{E}_{\mathbf{z} \sim D_\theta(\mathbf{x}_s)} \left[ 1 - G_\phi^i(\mathbf{z}, \mathbf{x}_s) \right] \right) \right] ds$$

$$\geq - \int_0^1 \frac{\alpha'_{1-s}}{1 - \alpha_{1-s}} \mathbb{E}_{\mathbf{x}_s \sim \mathbf{p}_{1-s}(\cdot|\mathbf{x}_0)} \left[ \sum_{i=1,x_s^i \neq \mathbf{m}}^L \mathbb{E}_{\mathbf{z} \sim D_\theta(\mathbf{x}_s)} \left[ \log \left( 1 - G_\phi^i(\mathbf{z}, \mathbf{x}_s) \right) \right] \right] ds$$

$$= -\int_0^1 \frac{\alpha'_{1-s}}{1-\alpha_{1-s}} \mathbb{E}_{\mathbf{x}_s \sim \mathbf{p}_{1-s}(\cdot|\mathbf{x}_0)} \left[ \sum_{i=1, x_s^i \neq \mathbf{m}}^L \mathbb{E}_{\mathbf{z} \sim D_\theta(\mathbf{x}_s)} \left[ \mathrm{CE}\left(\mathrm{Cat}(z^i; \delta(x_0^i)), 1 - G_\phi^i(\mathbf{z}, \mathbf{x}_s))\right)\right]\right] ds$$

$$= \mathcal{E}_{UP}(\mathbf{x}_0),$$

where for the second inequality we applied Jensen's, for the second-to-last equality we use $\mathrm{Cat}(z^i; \delta(x_0^i)) = 1$ for $z^i \sim D_\theta^i(x_s)$ with $x_s^i \neq \mathbf{m}$ by assumption, and for the last equality yet again we make the time change $t = 1 - s$, and observe that we defined $G_U^i(\mathbf{z}, \mathbf{x}) = 1 - G_\phi^i(\mathbf{z}, \mathbf{x})$ when $x^i \neq \mathbf{m}$.

The proof of the proposition is now complete. $\square$

## B.2 FINDING THE OPTIMAL PLANNER UNDER THE ELBO-INFORMED LOSS: PROPOSITION 2

Here we derive the form of the optimal $G_U$ and $G_M$ using the training loss associated to the ELBO proved in Proposition 1 for a fixed MDM denoiser $D_\theta$. Recall, as discussed in Subsection 3.2 that we train via $\mathcal{L}(\phi) = -\mathbb{E}_{\mathbf{x}_0 \sim \mathbf{p}_0} \left[ \mathcal{E}_{MP}(\mathbf{x}_0) + \mathcal{E}_{UP}(\mathbf{x}_0)\right]$. In practice, we train a single network $T_\phi(\mathbf{z}, \mathbf{x})$ to play the role of both $G_U$ and $G_M$ in $\mathcal{E}_{UP}$ and $\mathcal{E}_M$ respectively. Making $\mathcal{L}(\phi)$ with this insertion explicit for reference:

$$\mathcal{L}(\phi) = \int_0^1 \frac{d\alpha_t}{dt} \cdot \frac{1}{1-\alpha_t} \mathbb{E}_{\mathbf{x}_0 \sim \mathbf{p}_0} \left[ \mathbb{E}_{\mathbf{x}_t \sim \mathbf{p}_t(\cdot; \mathbf{x}_0)} \left[ \sum_{i=1}^L \mathbb{E}_{\mathbf{z} \sim D_\theta(\mathbf{x}_t)} \left[ \mathrm{CE}\left(\mathrm{Cat}(z^i; \delta(x_0^i)), T_\phi^i(\mathbf{z}, \mathbf{x}_t))\right)\right]\right]\right] dt \tag{21}$$

where $\mathbf{p}_t$ is defined per Eq. 1 and $\mathrm{CE}(a, b) = a \log(b) + (1-a)\log(1-b)$.

We have the following proposition:

**Proposition 2.** *Let $T_\phi : \mathcal{V}^L \times \mathcal{V}^L \to [0, 1]$ be trained via $\mathcal{L}(\phi) = -\mathbb{E}_{\mathbf{x}_0 \sim \mathbf{p}_0} \left[ \mathcal{E}_{MP}(\mathbf{x}_0) + \mathcal{E}_{UP}(\mathbf{x}_0)\right]$, taking $G_U = G_M = T_\phi$ in $\mathcal{E}_{MP}$ and $\mathcal{E}_{UP}$ from Proposition 1. Define, for $\mathbf{z}, \mathbf{x_t} \in \mathcal{V}^\mathbf{L}$ with $\mathbf{z}$ a sequence of unmasked tokens satisfying $\mathbf{z}^i = \mathbf{x}_t^i$ for all $i$ such that $\mathbf{x}_t^i \neq \mathbf{m}$:*

$$\bar{T}^i(\mathbf{z}, \mathbf{x}_t) = \begin{cases} \mathbf{p}_0\left(x_0^i = z^i | x_0^j = x_t^j, \forall j \neq i \text{ such that } x_t^j \neq \mathbf{m}\right), & x_t^i \neq \mathbf{m} \\ \mathbf{p}_0\left(x_0^i = z^i | x_0^j = x_t^j, \forall j \text{ such that } x_t^j \neq \mathbf{m}\right), & x_t^i = \mathbf{m} \end{cases}.$$

*Then, for any $D_\theta$, $\mathcal{L}(\phi)$ is uniquely minimized over $T_\phi^i$ when $T_\phi^i(\mathbf{z}, \mathbf{x}_t) = \bar{T}^i(\mathbf{z}, \mathbf{x}_t)$.*

Note that in practice $\mathbf{z}$ which is inserted into $G_M$ and $G_U$ is always sampled from $D_\theta^i(\mathbf{x}_t)$, which is taken to be $\delta(x_t^i)$ in positions where $x_t^i \neq \mathbf{m}$. Thus the proposition considers exactly the form of sequences that $T_\phi$ will see during sampling. Also observe that this minimizer is doing exactly what we would desire from $\bar{T}^i$ in both its roles as $G_U^i$ and $G_M^i$:

- For $\bar{T}^i(\mathbf{z}, \mathbf{x}_t)$ as $G_U^i(\mathbf{z}, \mathbf{x}_t)$, we keep a previously unmasked position $z^i = x_t^i$ with probability proportional to the probability that, conditionally upon the information about the currently unmasked positions of $x_t$ other than $i$, $z^i = x_t^i$ is found in the $i$'th position of a sequence under the data distribution $\mathbf{p}_0$.

- Similarly, for $\bar{T}^i(\mathbf{z}, \mathbf{x}_t)$ as $G_M^i(\mathbf{z}, \mathbf{x}_t)$, we unmask a token in position $i$ to $z^i$ suggested by the denoiser with probability proportional to the conditional probability that $z^i$ is found in the $i$'th position of a sequence under the data distribution $\mathbf{p}_0$.

Thus, for any denoiser, the optimal $T_\phi^i$ is aiming to steer towards planned paths which are representative of the data distribution in both its roles as $G_U^i$ and $G_M^i$. We emphasize once again that even if one wishes to train only for the role of $G_U$ or $G_M$ respectively, to gain a meaningful training signal for both correctly and incorrectly denoised $\mathbf{z}$, one should use $G_U = G_M = T_\phi$ in Algorithm 2.

We now proceed with the proof of Proposition 2.

*Proof.* We begin by defining

$$\mathbf{r}_t(\mathbf{x}_0; \mathbf{x}_t) := \frac{\mathbf{p}_0(\mathbf{x}_0)\mathbf{p}_t(\mathbf{x}_t; \mathbf{x}_0)}{\mathbf{p}_t(\mathbf{x}_t; \mathbf{x}_0 \sim \mathbf{p}_0)},$$

where by $\mathbf{p}_t(\mathbf{x}_t; \mathbf{x}_0 \sim \mathbf{p}_0)$ we mean, as usual, $\mathbf{p}_t(\mathbf{x}_t; \mathbf{x}_0 \sim \mathbf{p}_0) = \sum_{\mathbf{x}_0 \in \mathcal{V}^L} \mathbf{p}_t(\mathbf{x}_t; \mathbf{x}_0) \mathbf{p}_0(\mathbf{x}_0)$, and where $\mathbf{p}_t(\mathbf{x}_t; \mathbf{x}_0)$ is as in Eq. 1. We then observe:

$$\mathcal{L}(\phi) = \int_0^1 \beta_t \mathbb{E}_{\mathbf{x}_t \sim \mathbf{p}_t(\cdot; \mathbf{x}_0 \sim \mathbf{p}_0)} \left[ \mathbb{E}_{\mathbf{z} \sim D_\theta(\mathbf{x}_t)} \left[ \mathbb{E}_{\mathbf{x}_0 \sim \mathbf{r}_t(\cdot; \mathbf{x}_t)} \left[ \sum_{i=1}^L \mathrm{CE} \left( \mathrm{Cat}(z^i; \delta(x_0^i)), T_\phi^i(\mathbf{z}, \mathbf{x}_t) \right) \right] \right] \right] dt,$$

where $\beta_t = \frac{d\alpha_t}{dt} \cdot \frac{1}{1-\alpha_t}$. Next, we observe that there is no relationship enforced between $T_\phi^i(\mathbf{z}, \mathbf{x}_t)$ and $T_\phi^i(\bar{\mathbf{z}}, \bar{\mathbf{x}}_t)$ for $(\mathbf{z}, \mathbf{x}_t) \neq (\bar{\mathbf{z}}, \bar{\mathbf{x}}_t) \in \mathcal{V}^L$. So, as $\beta_t$ is negative, minimizing $\mathcal{L}_{\mathrm{UP}}(\phi)$ amounts to maximizing

$$L(\mathbf{z}, \mathbf{x}_t) = \mathbb{E}_{\mathbf{x}_0 \sim \mathbf{r}_t(\cdot; \mathbf{x}_t)} \left[ \sum_{i=1}^L \mathrm{CE} \left( \mathrm{Cat}(z^i; \delta(x_0^i)), T_\phi^i(\mathbf{z}, \mathbf{x}_t) \right) \right]$$

for each $\mathbf{z}$ and $\mathbf{x}_t$. Using that each $i$'th term in the sum only depends on $\mathbf{x}_0$ through $x_0^i$, we have:

$$L(\mathbf{z}, \mathbf{x}_t) = \sum_{i=1}^L \mathrm{CE} \left( r_t^i(z^i; \mathbf{x}_t), G_U^i(\mathbf{z}, \mathbf{x}_t) \right),$$

where

$$r_t^i(z^i; \mathbf{x}_t) = \sum_{\mathbf{x}_0 \in \mathcal{V}^L} \mathbf{r}_t(\mathbf{x}_0^{-i, z^i}; \mathbf{x}_t),$$

where $\mathbf{x}_0^{-i, z^i}$ denotes that we remove the $i$'th coordinate of $\mathbf{x}_0$ and replace it with $z^i$. It now follows from the fact that for fixed $a$, $\mathrm{CE}(a, b) = a \log(b) + (1-a) \log(1-b)$ is uniquely maximized at $b = a$ that the optimal $G_U^i(\mathbf{z}, \mathbf{x}_t)$ is $r_t^i(z^i; \mathbf{x}_t)$. It remains to show, that thanks to the simple form of $\mathbf{p}_t$ from Eq. 1, indeed $r_t^i(z^i; \mathbf{x}_t)$ does not depend on time and is equal to $\bar{T}^i(\mathbf{z}, \mathbf{x}_t)$.

First, we note that, as $\mathbf{p}_0$ does not contain any sequences with the token $\mathbf{m}$ in its support by definition:

$$\mathbf{p}_t(\mathbf{x}_t; \mathbf{x}_0 \sim \mathbf{p}_0) = \sum_{\mathbf{x}_0 \in \mathcal{V}^L} \mathbf{p}_0(\mathbf{x}_0) \prod_{i=1}^L \mathrm{Cat}(x_t^i; \alpha_t \delta(x_0^i) + (1-\alpha_t)\delta(\mathbf{m})) \tag{22}$$

$$= \alpha_t^{L - N_M(\mathbf{x}_t)}(1-\alpha_t)^{N_M(\mathbf{x}_t)} \mathbf{p}_0(x_0^j = x_t^j, \forall j \text{ such that } x_t^j \neq \mathbf{m}),$$

where $N_M(\mathbf{x}_t)$ is the number of positions of $\mathbf{x}_t$ which are equal to $\mathbf{m}$. Indeed, this computation is the result of Ou et al. (2024) Proposition 1.

Then we observe that, similarly, for fixed $i$:

$$\sum_{\mathbf{x}_0 \in \mathcal{V}^L} \mathbf{p}_0(\mathbf{x}_0^{-i, z^i}) \mathbf{p}_t(\mathbf{x}_t; \mathbf{x}_0^{-i, z^i})$$

$$= \mathrm{Cat}(x_t^i; \alpha_t \delta(z^i) + (1-\alpha_t)\delta(\mathbf{m})) \sum_{\mathbf{x}_0 \in \mathcal{V}^L} \mathbf{p}_0(\mathbf{x}_0^{-i, z^i}) \prod_{j=1, j \neq i}^L \mathrm{Cat}(x_t^j; \alpha_t \delta(x_0^j) + (1-\alpha_t)\delta(\mathbf{m}))$$

$$= \mathrm{Cat}(x_t^i; \alpha_t \delta(z^i) + (1-\alpha_t)\delta(\mathbf{m})) \alpha_t^{L-1-N_M(\mathbf{x}_t^{-i})}(1-\alpha_t)^{N_M(\mathbf{x}_t^{-i})}$$

$$\times \mathbf{p}_0(x_0^i = z^i \text{ and } x_0^j = x_t^j, \forall j \text{ such that } x_t^j \neq \mathbf{m}),$$

where $\mathbf{x}_t^{-i} \in \mathcal{V}^{L-1}$ is obtained from $\mathbf{x}_t$ by removing its $i$'th coordinate. $r_t^i(z^i; \mathbf{x}_t)$ is precisely this term divided by $\mathbf{p}_t(\mathbf{x}_t; \mathbf{x}_0 \sim \mathbf{p}_0)$.

There are two cases to consider. The first is when $x_t^i = \mathbf{m}$. Then $N_M(\mathbf{x}_t^{-i}) = N_M(\mathbf{x}_t) - 1$ and $\mathrm{Cat}(x_t^i; \alpha_t \delta(z^i) + (1-\alpha_t)\delta(\mathbf{m})) = 1 - \alpha_t$, so

$$\sum_{\mathbf{x}_0 \in \mathcal{V}^L} \mathbf{p}_0(\mathbf{x}_0^{-i, z^i}) \mathbf{p}_t(\mathbf{x}_t; \mathbf{x}_0^{-i, z^i})$$

$$= \alpha_t^{L - N_M(\mathbf{x}_t)}(1-\alpha_t)^{N_M(\mathbf{x}_t)} \mathbf{p}_0(x_0^i = z^i \text{ and } x_0^j = x_t^j, \forall j \text{ such that } x_t^j \neq \mathbf{m}),$$

and dividing by Eq. 22, the time-dependent terms cancel, yielding the desired result. The second is when $x_t^i \neq \mathbf{m}$. Then $N_M(\mathbf{x}_t^{-i}) = N_M(\mathbf{x}_t)$ and $\mathrm{Cat}(x_t^i; \alpha_t \delta(z^i) + (1-\alpha_t)\delta(\mathbf{m})) = \alpha_t \mathrm{Cat}(z^i; \delta(x_t^i))$,

so

$$\sum_{\mathbf{x}_0 \in \mathcal{V}^L} \mathbf{p}_0(\mathbf{x}_0^{-i,z^i}) \mathbf{p}_t(\mathbf{x}_t; \mathbf{x}_0^{-i,z^i})$$

$$= \alpha_t^{L-N_M(\mathbf{x}_t)}(1-\alpha_t)^{N_M(\mathbf{x}_t)} \mathrm{Cat}(z^i; \delta(x_t^i)) \mathbf{p}_0(x_0^i = z^i \text{ and } x_0^j = x_t^j, \forall j \text{ such that } x_t^j \neq \mathbf{m}).$$

Using that $\mathrm{Cat}(z^i; \delta(x_t^i)) = 1$ by assumption and again dividing by Eq. 22, the time-dependent terms cancel, yielding the desired result. $\qquad\square$

### B.3    FORM OF THE ELBO FOR VARYING $\eta$

Here we show how to find the form of the ELBO from Proposition 1 for arbitrary stochasticity $\eta \geq 0$ in the definition of $\tilde{G}_\eta$ from Eq. 7.

We observe that taking $\eta \neq 1$ corresponds to modifying $G_M$ and $G_U$ to $G_{\eta,M}$ and $G_{\eta,U}$ respectively, where:

$$G_{\eta,M}^j(\mathbf{z},\mathbf{x}) = \frac{\eta G_M^j(\mathbf{z},\mathbf{x})}{C_\eta(\mathbf{z},\mathbf{x})}$$

$$G_{\eta,U}^j(\mathbf{z},\mathbf{x}) = \frac{G_U^j(\mathbf{z},\mathbf{x})}{C_\eta(\mathbf{z},\mathbf{x})}$$

$$C_\eta(\mathbf{z},\mathbf{x}) = \sum_{i=1,\mathbf{x}^i \neq \mathbf{m}}^L G_U^j(\mathbf{z},\mathbf{x}) + \eta \sum_{i=1,\mathbf{x}^i = \mathbf{m}}^L G_M^j(\mathbf{z},\mathbf{x})$$

(see Alg. 5 for reference). Inserting this choice into $\mathcal{E}$ from Proposition 1 yields $\mathcal{E}_\eta(\mathbf{x}_0) \leq \log(P_0^{\theta,\phi,\eta}(\mathbf{x}_0))$ for each fixed $\mathbf{x}_0 \in \mathcal{V}^L$ given by $\mathcal{E}_\eta(\mathbf{x}_0) = \mathcal{E}_{\eta,\mathrm{MP}}(\mathbf{x}_0) + \mathcal{E}_{\eta,\mathrm{UP}}(\mathbf{x}_0) + \mathcal{E}_\mathrm{D}(\mathbf{x}_0)$, where:

$$\mathcal{E}_{\eta,\mathrm{MP}}(\mathbf{x}_0) = -\int_0^1 \beta_t \mathbb{E}_{\mathbf{x}_t \sim \mathbf{P}_t(\cdot;\mathbf{x}_0)} \left[ \sum_{i=1,\mathbf{x}_t^i = \mathbf{m}}^L \mathbb{E}_{\mathbf{z} \sim D_\theta(\mathbf{x}_t)} \left[ \mathrm{CE}\left( \mathrm{Cat}(z^i; \delta(x_0^i)), \eta G_M^i(\mathbf{z},\mathbf{x}_t)/C_\eta(\mathbf{z},\mathbf{x}_t) \right) \right] \right] dt$$

$$\mathcal{E}_{\eta,\mathrm{UP}}(\mathbf{x}_0) = -\int_0^1 \beta_t \mathbb{E}_{\mathbf{x}_t \sim \mathbf{P}_t(\cdot;\mathbf{x}_0)} \left[ \sum_{i=1,\mathbf{x}_t^i \neq \mathbf{m}}^L \mathbb{E}_{\mathbf{z} \sim D_\theta(\mathbf{x}_t)} \left[ \mathrm{CE}\left( \mathrm{Cat}(z^i; \delta(x_0^i)), G_U^i(\mathbf{z},\mathbf{x}_t)/C_\eta(\mathbf{z},\mathbf{x}_t) \right) \right] \right] dt$$

$$\mathcal{E}_\mathrm{D}(\mathbf{x}_0) = -\int_0^1 \beta_t \mathbb{E}_{\mathbf{x}_t \sim \mathbf{P}_t(\cdot;\mathbf{x}_0)} \left[ \sum_{i=1,\mathbf{x}_t^i = \mathbf{m}}^L \delta(x_0^i)^\top \log(D_\theta^i(\mathbf{x}_t)) \right] dt,$$

where, as before, $\mathbf{p}_t$ is defined per Eq. 1, $\beta_t = \frac{d\alpha_t}{dt} \cdot \frac{1}{1-\alpha_t}$, and $\mathrm{CE}(a,b) = a\log(b)+(1-a)\log(1-b)$ for $a,b \in [0,1]$, with $0\log 0 = 0$. Note that the effect of increasing $\eta$ will be to place more weight on the role of the masked planner, since $\frac{\partial}{\partial \eta} G_{\eta,M}^j(\mathbf{z},\mathbf{x}) = \frac{G_M^j(\mathbf{z},\mathbf{x}) \sum_{i=1,x^i \neq \mathbf{m}} G_U^i(\mathbf{z},\mathbf{x})}{C_\eta^2(\mathbf{z},\mathbf{x})} \geq 0$, and hence $\mathrm{CE}(1, G_{\eta,M}^j(\mathbf{z},\mathbf{x}))$ is increasing in $\eta$ and $\mathrm{CE}(0, G_{\eta,M}^j(\mathbf{z},\mathbf{x}))$ is decreasing in $\eta$. Conversely, $\frac{\partial}{\partial \eta} G_{\eta,U}^j(\mathbf{z},\mathbf{x}) = -\frac{G_U^j(\mathbf{z},\mathbf{x}) \sum_{i=1,x^i = \mathbf{m}} G_M^i(\mathbf{z},\mathbf{x})}{C_\eta^2(\mathbf{z},\mathbf{x})} \leq 0$, so $\mathrm{CE}(1, G_{\eta,U}^j(\mathbf{z},\mathbf{x}))$ is decreasing in $\eta$ and $\mathrm{CE}(0, G_{\eta,U}^j(\mathbf{z},\mathbf{x}))$ is increasing in $\eta$. Recalling the loss for the planner is given by $\mathcal{L}_\eta(\phi) = -\mathbb{E}_{\mathbf{x}_0 \sim \mathbf{p}_0}[\mathcal{E}_{\eta,\mathrm{UP}}(\mathbf{x}_0) + \mathcal{E}_{\eta,\mathrm{MP}}(\mathbf{x}_0)]$ and that $\beta_t \leq 0$, indeed we see that increasing $\eta$ puts more weight on $G_M^i$ and less on $G_U^i$ matching the label $\mathrm{Cat}(z^i; \delta(x_0^i))$.

### B.4    PROOF OF LEMMA 1

We consider each limit one at a time.

For Eq. 11, we have:

$$\hat{\mathbf{q}}_{\lfloor sT \rfloor}^T(\mathbf{x}|\mathbf{x}_0) = \prod_{i=1}^L \hat{q}_{\lfloor sT \rfloor}^T(x^i|x_0^i)$$

where $\hat{q}^T_{\lfloor sT \rfloor}(\cdot|x^i_0)$ is the distribution after $\lfloor sT \rfloor$ jumps of a single independent coordinate evolving according to Eq. 2 and

$$\hat{q}^T_{\lfloor sT \rfloor}(x^i|x^i_0) = 0, x^i \notin \{x^i_0, \mathbf{m}\},$$

$$\hat{q}^T_{\lfloor sT \rfloor}(\mathbf{m}|x^i_0) = \prod_{t=T-1}^{T-1-\lfloor sT \rfloor} \frac{1 - \alpha_{(t-1)/T}}{1 - \alpha_{t/T}} = \frac{1 - \alpha_{1-1/T-\lfloor sT \rfloor/T}}{1 - \alpha_{1-1/T}} \to 1 - \alpha_{1-s} \text{ as } T \to \infty.$$

Similarly for Eq. 12, $\hat{q}^T_1(x^i_0|\mathbf{x}_0) = 1 - \hat{q}^T_1(\mathbf{m}|x^i_0) = 1 - \frac{1-\alpha_{1/T}}{1-\alpha_{1-1/T}} \to 1$ as $T \to \infty$.

For Eq. 13, observe from Eq. 2 that, for $y^i \neq x^i$, $t = \lfloor Ts \rfloor$:

$$\lim_{T \to \infty} q_t(y^i|x^i, x^i_0) = 0$$

$$\lim_{T \to \infty} Tq_t(y^i|x^i, x^i_0) = -\frac{\alpha'_{1-s}}{1 - \alpha_{1-s}} \text{Cat}\left(y^i; \delta(x^i_0)\right) \text{Cat}\left(x^i; \delta(\mathbf{m})\right),$$

where the second limit follows from the definition of the derivative. Thus we have, recalling the definition of $\mathbf{q}^T_t$ from Eq. 9, if $\mathbf{y}$ and $\mathbf{x}$ differ in two or more coordinates, the entire product vanishes, and if they differ in exactly 1, the limit is given by the above. This yields precisely Eq. 13.

Finally, for Eq. 14, observe from Eq. 5 that for $\mathbf{x}, \mathbf{z} \in \mathcal{V}^L$ and $y^i \neq x^i \in \mathcal{V}$, $t = \lfloor Ts \rfloor$:

$$\lim_{T \to \infty} q_{t,\theta}(y^i|\mathbf{x}, \mathbf{z}) = 0$$

$$\lim_{T \to \infty} Tq_{t,\theta}(y^i|\mathbf{x}, \mathbf{z}) = -\frac{\alpha'_{1-s}}{1 - \alpha_{1-s}} G^i_\phi(\mathbf{z}, \mathbf{x}) \text{Cat}(y^i; \delta(z^i)), x^i = \mathbf{m}$$

$$\lim_{T \to \infty} Tq_{t,\theta}(y^i|\mathbf{x}, \mathbf{z}) = -\frac{\alpha'_{1-s}}{1 - \alpha_{1-s}} \frac{G^i_\phi(\mathbf{z}, \mathbf{x})}{1 - \text{Cat}(x^i; D^i_\theta(\bar{\mathbf{x}}))} \text{Cat}(y^i; D^i_\theta(\bar{\mathbf{x}})), x^i \neq \mathbf{m},$$

where here, as before, $\bar{\mathbf{x}}$ is shorthand for removing the i'th coordinate of $\mathbf{x}$ and replacing with $\mathbf{m}$.

Then, recalling the definition of $\mathbf{q}^T_{t,\theta}$ from Eq. 8, we pass the limit inside the expected value to obtain, for $\mathbf{y} \neq \mathbf{x} \in \mathcal{V}^L$, $t = \lfloor sT \rfloor$:

$$\lim_{T \to \infty} T\mathbf{q}^T_{t,\theta}(\mathbf{y}|\mathbf{x})$$

$$= -\frac{\alpha'_{1-s}}{1 - \alpha_{1-s}} \begin{cases} \mathbb{E}_{\mathbf{z} \sim D_\theta(\mathbf{x})}\left[G^i_\phi(\mathbf{z}, \mathbf{x})\text{Cat}(y^i; \delta(z^i))\right] & , d_{\text{HAM}}(\mathbf{x}, \mathbf{y}) = 1, y^i \neq y^i, x^i = \mathbf{m} \\ \frac{\mathbb{E}_{\mathbf{z} \sim D_\theta(\mathbf{x})}\left[G^i_\phi(\mathbf{z}, \mathbf{x})\right]}{1 - \text{Cat}(x^i; D^i_\theta(\bar{\mathbf{x}}))}\text{Cat}(y^i; D^i_\theta(\bar{\mathbf{x}})) & , d_{\text{HAM}}(\mathbf{x}, \mathbf{y}) = 1, y^i \neq y^i, x^i \neq \mathbf{m} \\ 0, & \text{otherwise} \end{cases}$$

For the first term above, we recall that $\mathbf{z} \sim D_\theta(\mathbf{x})$ means each coordinate $z^i$ is sampled independently from $D^i_\theta(\mathbf{x})$. Thus:

$$\mathbb{E}_{\mathbf{z} \sim D_\theta(\mathbf{x})}\left[G^i_\phi(\mathbf{z}, \mathbf{x})\text{Cat}(y^i; \delta(z^i))\right] = \sum_{\mathbf{z} \in \mathcal{V}^L, z^i = y^i} \prod_{j=1}^L \text{Cat}(z^j; D^j_\theta(\mathbf{x})) G^i_\phi(\mathbf{z}, \mathbf{x})$$

$$= \text{Cat}(y^i; D^i_\theta(\mathbf{x})) \mathbb{E}_{\mathbf{z} \sim D_\theta(\mathbf{x})}\left[G^i_\phi(\mathbf{z}^{-i, y^i}, \mathbf{x})\right],$$

where $\mathbf{z}^{-i, y^i}$ denotes replacing the i'th coordinate of $\mathbf{z}$ with $y^i$. This yields precisely Eq. 14.

## B.5 DERIVING THE DISCRETE TIME ELBO EQ. 10

This computation is standard, but we include it here for the sake of completeness. We begin by observing:

$$\mathbf{q}^T_\theta(\mathbf{x}_0) = \sum_{\mathbf{x}_{T-1}, \dots, \mathbf{x}_1 \in \mathcal{V}^L} \prod_{t=T}^1 \mathbf{q}^T_{t,\theta}(\mathbf{x}_{t-1}|\mathbf{x}_t),$$

where $\mathbf{q}_{t,\theta}^T(\mathbf{x}_{t-1}|\mathbf{x}_t)$ are as in Eq. 8, and we recall here that we fix $\mathbf{x}_T = (\mathbf{m}, \ldots, \mathbf{m})$. So, letting $\bar{q}^T$ be the distribution of the reference path up to time 1:

$$\bar{\mathbf{q}}^T(\mathbf{x}_1, \mathbf{x}_2, \ldots, \mathbf{x}_{T-1}|\mathbf{x}_0) = \prod_{t=T}^{2} \mathbf{q}_t^T(\mathbf{x}_{t-1}|\mathbf{x}_t, \mathbf{x}_0),$$

where $\mathbf{q}_t^T$ are as in Eq. 9 we have:

$$\mathbf{q}_\theta^T(\mathbf{x}_0) = \sum_{\mathbf{x}_{T-1},\ldots,\mathbf{x}_1 \in \mathcal{V}^L} \bar{\mathbf{q}}_1^T(\mathbf{x}_1, \ldots, \mathbf{x}_{T-1}|\mathbf{x}_0) \frac{\prod_{t=T}^{1} \mathbf{q}_{t,\theta}^T(\mathbf{x}_{t-1}|\mathbf{x}_t)}{\bar{\mathbf{q}}_1^T(\mathbf{x}_1, \ldots, \mathbf{x}_{T-1}|\mathbf{x}_0)}.$$

Then, by Jensen's inequality:

$$\log(\mathbf{q}_\theta^T(\mathbf{x}_0) \geq \sum_{\mathbf{x}_{T-1},\ldots,\mathbf{x}_1 \in \mathcal{V}^L} \bar{\mathbf{q}}_1^T(\mathbf{x}_1, \ldots, \mathbf{x}_{T-1}|\mathbf{x}_0) \sum_{t=T}^{1} \log\left(\frac{\mathbf{q}_{t,\theta}^T(\mathbf{x}_{t-1}|\mathbf{x}_t)}{\bar{\mathbf{q}}_1^T(\mathbf{x}_1, \ldots, \mathbf{x}_{T-1}|\mathbf{x}_0)}\right),$$

and inserting the definition of $\bar{q}_1^T$ inside the log:

$$\log(\mathbf{q}_\theta^T(\mathbf{x}_0)$$

$$\geq \sum_{\mathbf{x}_{T-1},\ldots,\mathbf{x}_1 \in \mathcal{V}^L} \bar{\mathbf{q}}_1^T(\mathbf{x}_1, \ldots, \mathbf{x}_{T-1}|\mathbf{x}_0) \left[\sum_{t=T}^{2} \log\left(\frac{\mathbf{q}_{t,\theta}^T(\mathbf{x}_{t-1}|\mathbf{x}_t)}{\mathbf{q}_t^T(\mathbf{x}_{t-1}|\mathbf{x}_t, \mathbf{x}_0)}\right) + \log(q_{1,\theta}^T(\mathbf{x}_0|\mathbf{x}_1))\right]$$

$$= \sum_{t=T}^{2} \sum_{\mathbf{x}_{T-1},\ldots,\mathbf{x}_1 \in \mathcal{V}^L} \bar{\mathbf{q}}_1^T(\mathbf{x}_1, \ldots, \mathbf{x}_{T-1}|\mathbf{x}_0) \log\left(\frac{\mathbf{q}_{t,\theta}^T(\mathbf{x}_{t-1}|\mathbf{x}_t)}{\mathbf{q}_t^T(\mathbf{x}_{t-1}|\mathbf{x}_t, \mathbf{x}_0)}\right)$$

$$+ \sum_{\mathbf{x}_{T-1},\ldots,\mathbf{x}_1 \in \mathcal{V}^L} \bar{\mathbf{q}}_1^T(\mathbf{x}_1, \ldots, \mathbf{x}_{T-1}|\mathbf{x}_0) \log(\mathbf{q}_{1,\theta}^T(\mathbf{x}_0|\mathbf{x}_1)).$$

Marginalizing out the variables not appearing in each term in the sum, we have, denoting by $\hat{\mathbf{q}}_t^T(\cdot|\mathbf{x}_0)$ the marginal distribution at time $t$ of the chain with one step transitions $\mathbf{q}_t^T$:

$$\log(\mathbf{q}_\theta^T(\mathbf{x}_0) \geq \sum_{t=T}^{2} \sum_{\mathbf{x}_t \in \mathcal{V}^L} \hat{\mathbf{q}}_t^T(\mathbf{x}_t|\mathbf{x}_0) \sum_{\mathbf{x}_{t-1} \in \mathcal{V}^L} \mathbf{q}_t^T(\mathbf{x}_{t-1}|\mathbf{x}_t, \mathbf{x}_0) \log\left(\frac{\mathbf{q}_{t,\theta}^T(\mathbf{x}_{t-1}|\mathbf{x}_t)}{\mathbf{q}_t^T(\mathbf{x}_{t-1}|\mathbf{x}_t, \mathbf{x}_0)}\right)$$

$$+ \sum_{\mathbf{x}_1 \in \mathcal{V}^L} \hat{\mathbf{q}}_1^T(\mathbf{x}_1|\mathbf{x}_0) \log(\mathbf{q}_{1,\theta}^T(\mathbf{x}_0|\mathbf{x}_1)),$$

which is precisely Eq. 10.

## C  ADDITIONAL BACKGROUND: CONTINUOUS TIME PERSPECTIVE

This section contains additional background information for contextualizing P2 in the greater "discrete diffusion model" landscape, and providing necessary mathematical background to understand the continuous time Markov chain setup and proof in Section Eq. D. A reader already familiar with the general discrete diffusion framework and theory of continuous time Markov chains may skip this section.

### C.1  DISCRETE DIFFUSION/FLOW MODELS: CONTINUOUS TIME PROBLEM SETUP

Here we discuss the general formulation of the problem setup and motivation behind discrete diffusion Austin et al. (2021); Lou et al. (2023); Sun et al. (2023); Campbell et al. (2022) and discrete flow models Campbell et al. (2024); Gat et al. (2024). This helps contextualize this manuscript in the broader landscape of the generative modeling framework, as well as introduce some additional notation that will be useful for the mathematical derivations in Appendix D.

Suppose we have a set of $N$ tokens, $S = \{1, \ldots, N\}$, and samples of sequences of length $L$ comprised of elements of $S$ from some distribution $\mathbf{p}_{data} \in \Delta^{N^L}$. We seek to generate new samples from $\mathbf{p}_{data}$ via learning a "denoising" function $D_\theta$ which allows one to sample from $\mathbf{p}_\theta \approx \mathbf{p}_{data}$.

To find such a function, we choose a family of probability measures $\{P_t(\cdot;\mu)\}_{t\in[0,1],\mu\in\Delta^{N^L}}$ such that $P_0(\cdot;\mu) = \mu$ and $P_1 = \pi$, where $\pi \in \Delta^{N^L}$ is some easily-sampled from reference distribution. Then we find $\{\overleftarrow{X}_t\}_{t\in[0,1]}$ a continuous-time Markov chain with $\mathbb{P}(\overleftarrow{X}_t = \mathbf{x}) = \overleftarrow{P}_t(\mathbf{x}; \mathbf{p}_{data}) := P_{1-t}(\mathbf{x}; \mathbf{p}_{data})$, and seek to use the "denoising function" $D_\theta$ to simulate a continuous time Markov chain $\{X_t^\theta\}_{t\in[0,1]}$ which is close in distribution to $\overleftarrow{X}$. In the end, we will have that taking $X_0^\theta \sim \pi$ and simulating the chain to time 1, $X_1^\theta \overset{d}{\approx} \overleftarrow{X}_1 \sim \mathbf{p}_{data}$. To understand what this process $X^\theta$ is and why the use of this intermediary Markov chain is useful for finding a choice of $D_\theta$, we first briefly review the theory of continuous time Markov chains in Appendix C.2.

## C.2 TIME-INHOMOGENEOUS CONTINUOUS TIME MARKOV CHAINS (CTMC)

A (time-inhomogenous) continuous-time Markov chain $\{X_t\}_{t\geq 0}$ on a finite set $\mathcal{X}$ is a stochastic process satisfying the Markov property, which can be formally summarized $\mathbb{P}(X_t = y|X_{s_1} = x_1, \ldots, X_{s_k} = x_k, X_s = x) = \mathbb{P}(X_t = y|X_s = x), \forall y, x_1, \ldots, x_k, x \in \mathcal{X}, 0 \leq s_1 < s_2 < \ldots < s_k < s < t \leq 1$. One can construct such a process by specifying a "rate matrix" $Q_t \in \mathbb{R}^{|\mathcal{X}|\times|\mathcal{X}|}$ with $Q_t(y,x) > 0$ and $Q_t(x,x) = -\sum_{y\neq x} Q_t(y,x)$ for all $x \neq y \in \mathcal{X}$ and $t \geq 0$. Along with an initial distribution $\mu \in \Delta^{|X|}$, $Q$ determines the 1-dimensional time marginals $\mathbb{P}(X_t = \cdot) \in \Delta^{|X|}$ via the Kolmogorov equation:

$$\frac{d}{dt}\mathbb{P}(X_t = \cdot) = Q_t\mathbb{P}(X_t = \cdot), \qquad t \geq 0 \tag{23}$$
$$\mathbb{P}(X_0 = x) = \mu(x), \qquad x \in \mathcal{X}.$$

When the above holds, we will say $Q$ "generates" $X$. Note that one can see necessarily that if $Q$ generates $X$,

$$Q_t(y,x) := \lim_{s\downarrow t}\frac{d}{ds}\mathbb{P}(X_s = y|X_t = x), \quad x \neq y \in \mathcal{X} \tag{24}$$

Knowing the entries of $Q$ also provides a means of generating samples from $X_t$ at any given time, since paths of $\{X_t\}_{t\geq 0}$ can be realized via a sequence of jump times $\{\tau_n\}_{n\in\mathbb{N}}$, with $\tau_i = \inf\{t > \tau_{i-1} : X_t \neq X_{\tau_{i-1}}\}$ and the effective discrete-time jump process $\{X_{\tau_i}\}_{i\in\mathbb{N}}$. Then

$$\mathbb{P}(X_{\tau_{i+1}} = y|X_{\tau_i} = x, \tau_{i+1} = t) = -\frac{Q_t(y,x)}{Q_t(x,x)}, \tag{25}$$

and

$$\log(\mathbb{P}(\tau_{i+1} > t|X_{\tau_i} = x, \tau_i = s)) = \int_s^t Q_p(x,x)dp.$$

For more background on time-inhomogenous continuous-time Markov chains, see e.g. Chapter 2 of Yin & Zhang (2013) or the appendix of Ren et al. (2024).

## C.3 THE ROLE OF THE DENOISER AND THE APPROXIMATE BACKWARDS PROCESS

In the "discrete diffusion model" framework, one in fact starts with specifying a rate matrix $Q_t$ generating some Markov chain $\{X_t\}_{t\in[0,1]}$ with $X_0 \sim \mathbf{p}_{data}$ and $X_1 \sim \pi$ and defines $P_t(\mathbf{x}; \mathbf{p}_{data}) := \mathbb{P}(X_t = \mathbf{x})$. $\overleftarrow{X}_t$ is then simply defined as $X_{1-t}$, and a rate matrix $\overleftarrow{Q}_t$ which generates $\overleftarrow{X}$ can be found from $Q_t$ via an application of Bayes' rule (see Prop. 3.2 in Sun et al. (2023)). In the "Discrete Flow Model" framework, one instead starts with a desired interpolation (often linear) $P_t(\cdot;\mu)$ between arbitrary $\mu \in \Delta^{N^L}$ and $\pi$, and constructs a rate matrix $\overleftarrow{Q}_t$ generating a $\overleftarrow{X}_t$ with one-dimensional time marginals $\overleftarrow{P}_t(\cdot; \mathbf{p}_{data})$ a posteriori.

As explained above, in order to generate samples of $\overleftarrow{X}_t$ at a given time (and in particular of $\overleftarrow{X}_1 \sim \mathbf{p}_{data}$), it is sufficient to have access to the entries of $\overleftarrow{Q}_t$. In both settings, however, the entries of $\overleftarrow{Q}_t$ will naturally depend on the unknown distribution $\mathbf{p}_{data}$, and hence, using the form of this dependence, a denoiser function $D_\theta$ is constructed in an attempt to approximate these unknown quantities. This results in a rate matrix $Q_t^\theta \approx \overleftarrow{Q}_t$, which generates the approximate backwards Markov chain $\{X_t^\theta\}_{t\in[0,1]}$. The distribution of the output of the resulting sampling scheme is then

$$\mathbf{p}_\theta = P_1^\theta = \mathbb{P}(X_1^\theta = \cdot).$$

The form of the denoiser, as well as the choice of $P_t, \overleftarrow{Q}$, and $Q^\theta$ in our particular setup are introduced in Sections 2.1, C.5, and D.1.

### C.4 THE CONDITIONAL BACKWARDS PROCESS

A pervasive assumption made in the literature is that for any fixed $\mathbf{x}_0 \in S^L$,

$$P_t(y; \delta(\mathbf{x}_0)) = \prod_{i=1}^{L} p_t(y^i | x_0^i) \tag{26}$$

for a family of probability measures $\{p_t(\cdot|x_0^i)\}_{t\in[0,1]} \subset \Delta^N$. We denote by $\overleftarrow{X}^{\mathbf{x}_0}$ the "conditional backwards process," on the point $\mathbf{x}_0$, defined as the Markov chain with distribution $\mathbb{P}(\overleftarrow{X}_t^{\mathbf{x}_0} = \mathbf{y}) = \overleftarrow{P}(\mathbf{y}; \delta(\mathbf{x}_0))$, and by $\overleftarrow{Q}^{\mathbf{x}_0}$ its rate matrix. The coordinates $(\overleftarrow{X}_1^{\mathbf{x}_0}, \ldots, \overleftarrow{X}_L^{\mathbf{x}_0})$ of $\overleftarrow{X}^{\mathbf{x}_0}$ are thus assumed independent, and each described by a continuous-time Markov chain $\{\overleftarrow{x}_t^i\}_{t\in[0,1]}$ with rate matrix $\overleftarrow{Q}_t^{x_0^i} \in \mathbb{R}^{N\times N}$ for $i = 1, \ldots, L$, $t \in [0,1]$ that yields $\mathbb{P}(\overleftarrow{x}_t^i = y^i) = \overleftarrow{p}_t(y^i|x_0^i)$ for all $t \in [0,1]$ and $y^i \in S$. The hope in making this assumption is that each coordinate of $X_t^\theta \approx \overleftarrow{X}_t$ will be able to be simulated independently in parallel without introducing significant error Sun et al. (2023).

$P_t(y; \mu)$ is taken to be linear in $\mu$, so we have $P_t(\mathbf{y}; \mathbf{p}_{data}) = \sum_{\mathbf{x}\in S^L} P_t(\mathbf{y}; \delta(\mathbf{x}))\mathbf{p}_{data}(\mathbf{x})$, and hence specifying $p_t(j|i), i, j \in S$ is what ultimately what determines the form of $\overleftarrow{Q}_t$ and hence the functions needed to be approximated by $D_\theta$ in order to construct $Q^\theta$. The most common choices explored this far in the literature are the "uniform diffusion," Lou et al. (2023); Schiff et al. (2025) which sets

$$p_t(j|i) = \alpha_t \mathrm{Cat}(j; \delta(i)) + \frac{1-\alpha_t}{S} \tag{27}$$

for $\alpha : [0,1] \to [0,1]$ with $\alpha_0 = 1, \alpha_1 = 0$ and the "masked diffusion," which is out subject of focus. Note that in the Discrete Diffusion Model framework, $p_t(j|i)$ is not always defined explicitly, and is often implicitly prescribed by asserting the "forward noising" process is the independent evolution of a CTMC on $S$ with rate matrix $\hat{Q}_t \in \mathbb{R}^{N\times N}$ on each coordinate (see e.g. Equations (15) and (16) in Lou et al. (2023)). $p_t(j|i)$ is then found by solving Eq. 23 with $Q = \hat{Q}$ and $\mu = \delta(i)$.

### C.5 MASKED DIFFUSION MODEL: CONTINUOUS TIME FORMULATION

In the case of a "masked diffusion model," one extends $S$ to $\bar{S} = S \cup \{\mathbf{m}\}$ for $\mathbf{m}$ some "masked state" outside the dictionary of tokens $S$, and takes $p_t(j|i) = \alpha_t \mathrm{Cat}(j; \delta(i)) + (1-\alpha_t)\mathrm{Cat}(j; \delta(\mathbf{m}))$. From here on we will refer to $\bar{S}$ as $\mathcal{V}$ as in the body of the manuscript. This choice of forward/noising process has been seen to outperform the uniform diffusion process Schiff et al. (2025) as well as other choices of $p_t$ Austin et al. (2021) consistently among applications. It corresponds to the coordinate-wise forward matrix given by, for $i \neq j \in \mathcal{V}$:

$$\hat{Q}_t(j, i) = \begin{cases} \sigma(t) & , \quad j = \mathbf{m}, i \neq \mathbf{m} \\ 0 & , \quad \text{otherwise} \end{cases}$$

with $\sigma(t) = -\frac{d}{dt}\log(\alpha_t)$, and through Eq. 26 yields Eq. 1.

In the masked-diffusion setting, both the Discrete Flow Model and Discrete Diffusion Model framework use the rate matrices for the conditional reversed process' coordinates (Campbell et al. (2024) Appendix F.1.) for $i \neq j \in \mathcal{V}$:

$$\overleftarrow{Q}_t^{x_0^i}(j, i) = \begin{cases} -\frac{\frac{d\alpha_{1-t}}{dt}}{1-\alpha_{1-t}} & , \quad i = \mathbf{m}, j = x_0^i \\ 0 & , \quad \text{otherwise} \end{cases}.$$

The resulting conditional rate matrix generating $\overset{\leftarrow \mathbf{x}_0}{X}_t$ is then, for $\mathbf{x} \neq \mathbf{y} \in \mathcal{V}^L$:

$$\overset{\leftarrow \mathbf{x}_0}{Q}_t(\mathbf{y}, \mathbf{x}) = \begin{cases} -\frac{\frac{d\alpha_{1-t}}{dt}}{1-\alpha_{1-t}}, & d_{HAM}(\mathbf{x}, \mathbf{y}) = 1, x^i \neq y^i, x^i = \mathbf{m}, y^i = x_0^i \\ 0, & \text{otherwise} \end{cases} \tag{28}$$

with $\overset{\leftarrow \mathbf{x}_0}{Q}_t(\mathbf{x}, \mathbf{x}) = \frac{\frac{d\alpha_{1-t}}{dt}}{1-\alpha_{1-t}} \sum_{i=1}^{L} \text{Cat}(x^i; \delta(\mathbf{m}))$, and the a rate matrix generating $\overset{\leftarrow}{X}_t$ is given for $\mathbf{x} \neq \mathbf{y} \in \mathcal{V}^L$, by:

$$\overset{\leftarrow}{Q}(\mathbf{y}, \mathbf{x}) = \begin{cases} -\frac{\frac{d\alpha_{1-t}}{dt}}{1-\alpha_{1-t}} p_{data}^i(y^i|\mathbf{x}_{\neq \mathbf{m}}), & d_{HAM}(\mathbf{x}, \mathbf{y}) = 1, x^i \neq y^i, x^i = \mathbf{m} \\ 0, & \text{otherwise} \end{cases}$$

and $\overset{\leftarrow}{Q}(\mathbf{x}, \mathbf{x}) = \frac{\frac{d\alpha_{1-t}}{dt}}{1-\alpha_{1-t}} \sum_{i=1}^{L} \text{Cat}(x^i; \delta(\mathbf{m}))$ (see e.g. Ou et al. (2024) Theorem 1 and Equation (3.1)). Here for $i \in \{1, \dots, L\}$, and $j \in \mathcal{V}$:

$$p_{data}^i(j|\mathbf{z}_{\neq \mathbf{m}}) := \mathbf{p}_{data}(\{\mathbf{x} : x^i = j\}|\mathbf{z}_{\neq \mathbf{m}}),$$

where for $\mathbf{z} \in \mathcal{V}^L$, $\mathbf{z}_{\neq \mathbf{m}}$ denotes the coordinates of $\mathbf{z}$ which are not equal to $\mathbf{m}$, and $d_{HAM}$ is Hamming distance.

Note that reversing time to $1-t$ and approximating $\frac{d\alpha_t}{dt}$ via $T[\alpha_{t+1/T} - \alpha_t]$ and $\frac{d\mathbb{P}(\overset{\leftarrow}{X}_t = \mathbf{x}|\overset{\leftarrow}{X}_s = \mathbf{y})}{dt}$ via $T[\mathbb{P}(\overset{\leftarrow}{X}_{t+1/T} = \mathbf{x}|\overset{\leftarrow}{X}_s = \mathbf{y}) - \mathbb{P}(\overset{\leftarrow}{X}_{t+1/T} = \mathbf{x}|\overset{\leftarrow}{X}_s = \mathbf{y})]$ yields the discrete-time approximation scheme Eq. 2 by way of Eq. 24. The is precisely the limit taken in Eq. 13.

One then parameterizes the approximate backwards process $X^{\theta, \text{mask}}$ via the denoiser $D_\theta$ by taking it to be the CTMC with rate matrix $\mathbf{x} \neq \mathbf{y} \in \mathcal{V}^L$:

$$Q^{\theta, \text{mask}}(\mathbf{y}, \mathbf{x}) = \begin{cases} -\frac{\frac{d\alpha_{1-t}}{dt}}{1-\alpha_{1-t}} \text{Cat}(y^i; D_\theta^i(\mathbf{x})), & d_{HAM}(\mathbf{x}, \mathbf{y}) = 1, x^i \neq y^i, x^i = \mathbf{m} \\ 0, & \text{otherwise} \end{cases} \tag{29}$$

In the same way as with $\overset{\leftarrow}{X}_t$, the discrete time approximation scheme for $X^{\theta, \text{mask}}$ is Eq. 3.

## C.6 ROLE OF THE ELBO

The training objective in general is obtained via the same methodology in both the Discrete Flow and Discrete Diffusion Model framework—in fact, this methodology can also be used for continuous diffusion models and denoising processes described by more general Markovian dynamics Benton et al. (2024).

We seek to minimize the KL divergence:

$$D_{KL}(\mathbf{p}_{data}||P_1^\theta) = \sum_{\mathbf{x} \in S^L} \mathbf{p}_{data}(\mathbf{x}) \log \left( \frac{\mathbf{p}_{data}(\mathbf{x})}{P_1^\theta(\mathbf{x})} \right)$$

$$= \sum_{\mathbf{x} \in S^L} \mathbf{p}_{data}(\mathbf{x}) \log \mathbf{p}_{data}(\mathbf{x}) - \sum_{\mathbf{x} \in S^L} \mathbf{p}_{data}(\mathbf{x}) \log(P_1^\theta(\mathbf{x}))$$

$$= -H(\mathbf{p}_{data}) - \sum_{\mathbf{x} \in S^L} \mathbf{p}_{data}(\mathbf{x}) \log(P_1^\theta(\mathbf{x})).$$

The first term - that is, the Shannon entropy of $\mathbf{p}_{data}$, $H(\mathbf{p}_{data})$ - is constant in $\theta$, and so we turn our attention to finding an "Evidence Based Lower Bound"

$$E(\mathbf{x}_0) \leq \log(P_1^\theta(\mathbf{x}_0))$$

for each fixed $\mathbf{x}_0 \in S^L$. The loss that we seek to minimize is then defined as:

$$\mathcal{L}_E^\theta := - \sum_{\mathbf{x} \in S^L} \mathbf{p}_{data}(\mathbf{x}) E(\mathbf{x}), \tag{30}$$

since $D_{KL}(\mathbf{p}_{data}||P_1^\theta) \leq -H(\mathbf{p}_{data}) + \mathcal{L}_E^\theta$.

Letting $\mathbb{P}^{\mathbf{x}_0} \in \mathcal{P}(D([0,1]; S^L))$ denote the Law (on the Skorokhod space of all cádlág paths from $[0,1]$ to $S^L$) of $\overset{\leftarrow \mathbf{x}_0}{X}$ and $\mathbb{P}^\theta \in \mathcal{P}(D([0,1]; S^L))$ the same but for $X^\theta$, we have, by the data-processing

inequality (see, e.g. Budhiraja & Dupuis (2019) Lemma 2.4 (f)):

$$\log(P_1^\theta(\mathbf{x}_0)) = -D_{KL}(\delta(\mathbf{x}_0)||P_1^\theta) \geq -D_{KL}(\mathbb{P}^{\mathbf{x}_0}||\mathbb{P}^\theta) \coloneqq E(\mathbf{x}_0), \tag{31}$$

That is, in order to make sure the approximate reverse process has the desired terminal distribution, by minimizing $\mathcal{L}_E$ we attempt to make it so that the entire path of the approximate reverse process matches that of the exact one. Eq. 31 is effectively the same as the first step in the proof of the discrete time ELBO - see B.5.

$E(\mathbf{x}_0)$ can be found via an application of Girsanov's Theorem for Markov Jump processes (see e.g. Theorem III.5.34 in Jacod & Shiryaev (2013) for a general result or Ren et al. (2024) Theorem 3.3 for the specific Markov Chain setting), and is expressed solely in terms of $\overset{\leftarrow\mathbf{x}_0}{Q}$, $D_\theta$, and $P_t(\cdot; \delta(\mathbf{x}_0))$. This yields an expression analogous to the discrete time ELBO Eq. 10 but for continuous time Markov chains -see the first line in the proof in Subsection D.2.

In the masked diffusion setting, where $Q^\theta$ is given by $Q^{\theta,\text{mask}}$ from Eq. 29 and $\overset{\leftarrow\mathbf{x}_0}{Q}$ is given by Eq. 28, this expression is given by Eq. 4 (see Sahoo et al. (2024) Equation (10)). This is exactly $\mathcal{E}_D$ from Proposition 1.

# D   MATHEMATICAL DETAILS: P2 FROM A CTMC POINT OF VIEW

In this Section we continue to use the notation established in Appendix C.

## D.1   P2 CONTINUOUS TIME FORMULATION

In order to formulate P2 we begin by modifying the jump matrix for the approximate backwards process Eq. 29, recall the planner function $G_\phi : \mathcal{V}^L \times \mathcal{V}^L \to [0,1]^L$. $G_\phi^j(\mathbf{z}, \mathbf{x})$ approximates the probability that the $j$'th token in a partially denoised sequence $\mathbf{x} \in \mathcal{V}^L$ should be (re)sampled given the conditional information about the rest of the sequence $\mathbf{x}$ and of the clean data $\mathbf{z}$ as predicted by $D_\theta$.

We define $F_{\theta,\phi} : \mathcal{V}^L \times \mathcal{V}^L \to [0,1]^L$ by

$$F_{\theta,\phi}^j(\mathbf{y}, \mathbf{x}) \coloneqq \text{Cat}(x^j; \delta(\mathbf{m}))\mathbb{E}_{Z \sim D_\theta(\mathbf{x})}[G_\phi^j(Z^{-j,y^j}, \mathbf{x})]$$
$$+ (1 - \text{Cat}(x^j; \delta(\mathbf{m})))\mathbb{E}_{Z \sim D_\theta(\mathbf{x})}[G_\phi^j(Z, \mathbf{x})]$$

where here we use the shorthand $Z \sim D_\theta(\mathbf{x})$ to mean $Z \sim \otimes_{i=1}^L D_\theta^i(\mathbf{x})$, and introduce the notation $\mathbf{z}^{-i,j}$ for $\mathbf{z} \in \mathcal{V}^L$, $i \in [L]$, and $j \in \mathcal{V}$ to mean the element of $\mathcal{V}^L$ resulting from replacing the i'th coordinate of $\mathbf{z}$ with $j$.

Via our interpretation of the role of $G_\theta$, $F_\theta^j(\mathbf{y}, \mathbf{x})$ gives the probability that the $j$'th position of $\mathbf{x}$ should be (re)sampled given the information about the rest of the sequence $\mathbf{x}$ and the data's $j$'th token via averaging out the information provided about the rest of the data's tokens from $D_\theta$.

Finally, we define

$$\hat{D}_\theta^i(\mathbf{x}) = D_\theta^i(\mathbf{x})\text{Cat}(x^i; \delta(\mathbf{m})) + \frac{D_\theta^i(x^{-i,\mathbf{m}})}{1 - \text{Cat}(x^i; D_\theta^i(\mathbf{x}^{-i,\mathbf{m}}))}(1 - \text{Cat}(x^i; \delta(\mathbf{m}))).$$

That is, when $x^i$ is masked $\text{Cat}(y^i; \hat{D}_\theta^i(\mathbf{x}))$ approximates the probability that the $i$'th token of $\mathbf{x}$ should be unmasked to $y^i$ given the conditional information about the unmasked tokens in $\mathbf{x}$, and when $x^i$ is not masked, $\hat{D}_\theta^i(\mathbf{x})$ approximates the probability that $i$'th token of $\mathbf{x}$ should be resampled to a value other than $x^i$, given the conditional information about the unmasked tokens in $\mathbf{x}$ other than $x^i$.

We now seek to modify $Q^{\theta,\text{mask}}$ from Eq. 29 in a way so that $F_{\theta,\phi}$ - by way of the planner $G_\phi$ - plays the role of selecting which position should be unmasked/resampled and $\hat{D}_\theta$ plays the role of choosing what it should be (re)sampled to.

For $x \neq y \in \mathcal{V}^L$, we thus set:

$$Q_t^{\theta,\phi}(\mathbf{y}, \mathbf{x}) \coloneqq \begin{cases} -\frac{\frac{d\alpha_{1-t}}{dt}}{1 - \alpha_{1-t}} F_{\theta,\phi}^i(\mathbf{y}, \mathbf{x})\text{Cat}(y^i; \hat{D}_\theta^i(\mathbf{x})) & , \quad d_{HAM}(\mathbf{x}, \mathbf{y}) = 1, x^i \neq y^i \\ 0 & , \text{otherwise} \end{cases}. \tag{32}$$

Note that, via the same formal discrete time approximation discussed above Eq. 29, the discrete time sampling scheme outlined in Section 3.1 approximates the distribution of the CTMC $X^{\theta,\phi}$ with rate matrix $Q^{\theta,\phi}$. This is precisely the limit taken in Eq. 14.

## D.2 PROOF OF THE ELBO PROPOSITION 1: CTMC VERSION

As per Eq. 31, it suffices to find a lower bound on $-D_{KL}(\mathbb{P}^{\mathbf{x}_0}||\mathbb{P}^{\theta,\phi})$, where $\mathbb{P}^{\mathbf{x}_0}$ is the Law of the continuous time Markov chain $\overset{\leftarrow \mathbf{x}_0}{X}$ with rate matrix $\overset{\leftarrow \mathbf{x}_0}{Q}$ given by Eq. 28, $\mathbb{P}^{\theta,\phi}$ is the Law of the continuous time Markov chain $X^{\theta,\phi}$ with rate matrix $Q^{\theta,\phi}$ given by Eq. 32, and $\overset{\leftarrow \mathbf{x}_0}{X}_0 = X_0^\theta = (\mathbf{m}, \ldots, \mathbf{m})$. Via an application of Girsanov's Theorem for CTMCs (see e.g. Theorem III.5.34 in Jacod & Shiryaev (2013) for a general result or Ren et al. (2024) Theorem 3.3 for the specific CTMC setting):

$$- D_{KL}(\mathbb{P}^{\mathbf{x}_0}||\mathbb{P}^{\theta})$$

$$= -\int_0^1 \mathbb{E}_{\mathbf{x}_t \sim \mathbf{p}_{1-t}(\cdot;\mathbf{x}_0)}\left[ -Q_t^{\theta,\phi}(\mathbf{x}_t, \mathbf{x}_t) + \overset{\leftarrow \mathbf{x}_0}{Q}(\mathbf{x}_t, \mathbf{x}_t) \right.$$

$$\left. + \sum_{\mathbf{y} \neq \mathbf{x}_t} \overset{\leftarrow \mathbf{x}_0}{Q}(\mathbf{y}, \mathbf{x}_t) \log\left( \frac{\overset{\leftarrow \mathbf{x}_0}{Q}(\mathbf{y}, \mathbf{x}_t)}{Q_t^{\theta,\phi}(\mathbf{y}, \mathbf{x}_t)} \right) \right] dt$$

$$= -\int_0^1 \frac{\frac{d\alpha_t}{dt}}{1 - \alpha_t} \mathbb{E}_{\mathbf{x}_t \sim \mathbf{p}_t(\cdot;\mathbf{x}_0)}\left[ \sum_{i=1}^L \left\{ \mathrm{Cat}(x_t^i; \delta(\mathbf{m}))(1 - \mathbb{E}_{Z \sim D_\theta(\mathbf{x}_t)}[G_\phi^i(Z, \mathbf{x}_t)]) \right. \right.$$

$$- (1 - \mathrm{Cat}(x_t^i; \delta(\mathbf{m})))\mathbb{E}_{Z \sim D_\theta(\mathbf{x}_t)}[G_\phi^i(Z, \mathbf{x}_t)]$$

$$\left. \left. + \mathrm{Cat}(x_t^i; \delta(\mathbf{m})) \log(F_{\theta,\phi}^i(\mathbf{x}_0, \mathbf{x}_t)\mathrm{Cat}(x_0^i; \hat{D}_\theta^i(\mathbf{x}_t))) \right\} \right] dt,$$

where in the third equality we have inserted the definitions of $\overset{\leftarrow \mathbf{x}_0}{Q}$ and $Q^{\theta,\phi}$ and reversed the role of the time parameter $t \mapsto 1 - t$, and $\boldsymbol{p}_t$ is as in Eq. 1.

We consider this as 4 parts:

$$E_1(\mathbf{x}_0) := -\int_0^1 \frac{\frac{d\alpha_t}{dt}}{1 - \alpha_t} \mathbb{E}_{\mathbf{x}_t \sim \mathbf{p}_t(\cdot;\mathbf{x}_0)}\left[ \sum_{i=1, x_t^i = \mathbf{m}}^L \left(1 - \mathbb{E}_{Z \sim D_\theta(\mathbf{x}_t)}[G_\phi^i(Z, \mathbf{x}_t)]\right) \right] dt$$

$$E_2(\mathbf{x}_0) := -\int_0^1 \frac{\frac{d\alpha_t}{dt}}{1 - \alpha_t} \mathbb{E}_{\mathbf{x}_t \sim \mathbf{p}_t(\cdot;\mathbf{x}_0)}\left[ \sum_{i=1, x_t^i \neq \mathbf{m}}^L \left(-\mathbb{E}_{Z \sim D_\theta(\mathbf{x}_t)}[G_\phi^i(Z, \mathbf{x}_t)]\right) \right] dt$$

$$E_3(\mathbf{x}_0) := -\int_0^1 \frac{\frac{d\alpha_t}{dt}}{1 - \alpha_t} \mathbb{E}_{\mathbf{x}_t \sim \mathbf{p}_t(\cdot;\mathbf{x}_0)}\left[ \sum_{i=1, x_t^i = \mathbf{m}}^L \log(F_{\theta,\phi}^i(\mathbf{x}_0, \mathbf{x}_t)) \right] dt$$

$$E_4(\mathbf{x}_0) := -\int_0^1 \frac{\frac{d\alpha_t}{dt}}{1 - \alpha_t} \mathbb{E}_{\mathbf{x}_t \sim \mathbf{p}_t(\cdot;\mathbf{x}_0)}\left[ \sum_{i=1, x_t^i = \mathbf{m}}^L \log(\mathrm{Cat}(x_0^i; \hat{D}_\theta^i(\mathbf{x}_t))) \right] dt$$

Recalling that $\frac{d\alpha_t}{dt} \leq 0$ for all $t \in [0, 1]$ and $G_\phi^i(Z, \mathbf{x}) \in [0, 1]$ for all $i \in \{1, \ldots, L\}$, $\mathbf{x} \in \mathcal{V}^L$, we see $E_1(\mathbf{x}_0)$ is positive for all $\mathbf{x}_0 \in \mathcal{V}^L$, and artificially attempting to ensure that the rates of the original CTMC and our modified one do not differ too much out of masked positions (see the discussion of the "Rate Forcing Term" in Appendix C.2 of Campbell et al. (2024)). Hence we simply bound it below by zero:

$$E_1(\mathbf{x}_0) \geq 0,$$

because we are only interested in $P_1^\theta$ being close to $\mathbf{p}_{data}$, not the entire trajectory of the chains $X^{\theta,\phi}$ and $\overset{\leftarrow}{X}$ being close.

For $E_3(\mathbf{x}_0)$ we note that, by definition, when $x_t^i = \mathbf{m}$, $F_{\theta,\phi}^i(\mathbf{x}_0, \mathbf{x}_t) = \mathbb{E}_{Z \sim D_\theta(\mathbf{x}_t)}[G_\phi^i(Z, \mathbf{x}_t)]$. Upon inserting this equality, we observe that, up to the time change $s = 1 - t$, these are the same 4 terms from Eq. 20 which we bound below by $\mathcal{E}(\mathbf{x}_0)$ in the time discritization version of the proof found in Section B, The rest of the proof thus proceeds identically.

## D.3 EQUIVALENCE OF MDMS WITH AOARMS

Here, for completeness, we recall the connection between masked diffusion language models and Any-Order Autoregressive Models Uria et al. (2014); Hoogeboom et al. (2022) as described in Zheng et al. (2025); Ou et al. (2024). We start by providing a simplified derivation of the equivalence of the two types of models' sampling schemes.

We begin by obtaining the diagonals for the matrix Eq. 29. Recalling $D_\theta^i(\mathbf{x}) = \delta(x^i)$ if $x^i \neq \mathbf{m}$, and $\sum_{y^i=1}^{d-1} \mathrm{Cat}(y^i; D_\theta^i(\mathbf{x})) = 1$ if $x^i = \mathbf{m}$:

$$-\sum_{y \neq x} Q_t^{\theta,\mathrm{mask}}(\mathbf{y}, \mathbf{x}) = \frac{\frac{d\alpha_{1-t}}{dt}}{1-\alpha_{1-t}} \sum_{i=1}^{L} \mathrm{Cat}(x^i; \delta(\mathbf{m})) \sum_{y^i \neq x^i} \mathrm{Cat}(y^i; D_\theta^i(\mathbf{x}))$$

$$= \frac{\frac{d\alpha_{1-t}}{dt}}{1-\alpha_{1-t}} \sum_{i=1}^{L} \mathrm{Cat}(x^i; \delta(\mathbf{m})) \sum_{y^i=1}^{d-1} \mathrm{Cat}(y^i; D_\theta^i(\mathbf{x}))$$

$$= \frac{\frac{d\alpha_{1-t}}{dt}}{1-\alpha_{1-t}} \sum_{i=1}^{L} \mathrm{Cat}(x^i; \delta(\mathbf{m})).$$

Then, if one considers the effective jump chain's transition probabilities as described in Eq. 25, we have, for $\mathbf{x} \neq \mathbf{y}$:

$$\mathbb{P}(X_{\tau_{k+1}}^{\theta,\mathrm{mask}} = \mathbf{y} | X_{\tau_k}^{\theta,\mathrm{mask}} = \mathbf{x}, \tau_{k+1} = t) = \mathbb{P}(X_{\tau_{k+1}}^{\theta,\mathrm{mask}} = \mathbf{y} | X_{\tau_k}^{\theta,\mathrm{mask}} = \mathbf{x})$$

$$= \frac{\mathrm{Cat}(x^i; \delta(\mathbf{m}))\mathrm{Cat}(y^i; D_\theta^i(\mathbf{x}))}{\sum_{i=1}^{L} \mathrm{Cat}(x^i; \delta(\mathbf{m}))},$$

when $d_{HAM}(\mathbf{x}, \mathbf{y}) = 1$ and $x^i \neq y^i$, and 0 when $d_{HAM}(\mathbf{x}, \mathbf{y}) \neq 1$.

Then, for any $j \in \{1, \ldots, L\}$:

$$\mathbb{P}([X_{\tau_{k+1}}^{\theta,\mathrm{mask}}]^j \neq [X_{\tau_k}^{\theta,\mathrm{mask}}]^j | X_{\tau_k}^{\theta,\mathrm{mask}} = \mathbf{x}, \tau_{k+1} = t) = \sum_{y^j \neq x^j} \mathbb{P}([X_{\tau_{k+1}}^{\theta,\mathrm{mask}}]^j = y^j | X_{\tau_k}^{\theta,\mathrm{mask}} = \mathbf{x})$$

$$= \sum_{y^j \neq x^j} \frac{\mathrm{Cat}(x^j; \delta(\mathbf{m})\mathrm{Cat}(y^j; D_\theta^j(\mathbf{x}))}{\sum_{i=1}^{L} \mathrm{Cat}(x^i; \delta(\mathbf{m}))}$$

$$= \frac{\mathrm{Cat}(x^j; \delta(\mathbf{m})) \sum_{y^j=1}^{d-1} \mathrm{Cat}(y^j; D_\theta^j(\mathbf{x}))}{\sum_{i=1}^{L} \mathrm{Cat}(x^i; \delta(\mathbf{m}))}$$

$$= \frac{\mathrm{Cat}(x^j; \delta(\mathbf{m}))}{\sum_{i=1}^{L} \mathrm{Cat}(x^i; \delta(\mathbf{m}))}$$

and, for $\mathbf{x}$ such that $x^j = \mathbf{m}$:

$$\mathbb{P}([X_{\tau_{k+1}}^{\theta,\mathrm{mask}}]^j = y^j | X_{\tau_k}^{\theta,\mathrm{mask}} = \mathbf{x}, \tau_{k+1} = t, [X_{\tau_{k+1}}^{\theta,\mathrm{mask}}]^j \neq [X_{\tau_k}^{\theta,\mathrm{mask}}]^j)$$

$$= \frac{\mathbb{P}([X_{\tau_{k+1}}^{\theta,\mathrm{mask}}]^j = y^j, [X_{\tau_{k+1}}^{\theta,\mathrm{mask}}]^j \neq [X_{\tau_k}^{\theta,\mathrm{mask}}]^j | X_{\tau_k}^{\theta,\mathrm{mask}} = \mathbf{x}, \tau_{k+1} = t)}{\mathbb{P}([X_{\tau_{k+1}}^{\theta,\mathrm{mask}}]^j \neq [X_{\tau_k}^{\theta,\mathrm{mask}}]^j | X_{\tau_k}^{\theta,\mathrm{mask}} = \mathbf{x}, \tau_{k+1} = t)}$$

$$= \frac{\sum_{i=1}^{L} \mathrm{Cat}(x^i; \delta(\mathbf{m})}{\mathrm{Cat}(x^j; \delta(\mathbf{m})} \sum_{\mathbf{y}' \in \mathcal{V}^L : [y']^j = y^j \neq x^j} \mathbb{P}(X_{\tau_{k+1}}^{\theta,\mathrm{mask}} = \mathbf{y}' | X_{\tau_k}^{\theta,\mathrm{mask}} = \mathbf{x})$$

$$= \mathrm{Cat}(x^j; \delta(\mathbf{m}))\mathrm{Cat}(y^j; D_\theta^j(\mathbf{x}))$$

$$= \mathrm{Cat}(y^j; D_\theta^j(\mathbf{x})).$$

Defining for $\mathbf{x} \in \mathcal{V}^L$, $M(\mathbf{x}) \coloneqq \{j \in \{1, \ldots, L\} : x^j = \mathbf{m}\}$, the corresponding Gillespie sampling scheme Gillespie (1977; 1976) for a standard masked diffusion model is thus as follows:

---

**Algorithm 3** Gillespie Sampler for masked diffusion language models

---

1: **Initialize:** $x_0 \leftarrow (\mathbf{m}, \mathbf{m}, \ldots, \mathbf{m})$, denoiser $D_\theta$
2: **for** $t = 1 : L$ **do**
3:  **Choose Random Coordinate for Unmasking:**
4:  Sample dimension $i \sim \mathrm{Unif}(M(x_t))$
5:  **Denoise:**
6:  Sample $z^i \sim D_\theta^i(x_t)$
7:  $x_{t+1}^i \leftarrow z^i$
8: **end for**
9: **return** $x_L$

---

Letting $\mathbb{S}_L$ be the set of all permutations of $\{1, \ldots, L\}$, we then have:

$$\mathbb{P}(X_1^{\theta,\text{mask}} = \mathbf{x}) = \frac{1}{L!} \sum_{\sigma \in \mathbb{S}_L} \prod_{i=1}^{L} \text{Cat}(x^{\sigma(i)}; D_\theta^{\sigma(i)}(\mathbf{x}^{-\sigma(\geq i), \mathbf{m}}))$$

$$= \mathbb{E}_{\sigma \sim \text{Unif}(\mathbb{S}_L)} \left[ \mathbb{P}(X_1^{\theta,\text{mask}} = \mathbf{x}|\sigma) \right]$$

where $\mathbf{x}^{-\sigma(\geq i), \mathbf{m}} \in \mathcal{V}^L$ is $\mathbf{x}$ but with $x^{\sigma(j)} = \mathbf{m}, \forall j \geq i$. Here $\sigma(i)$ represents the coordinate which is unmasked at time $\tau_i$. From this it is clear that with each unmasking, $D_\theta$ is gaining additional conditional information about the sequence it is denoising, and could potentially benefit from backtracking and remasking previously unmasked tokens.

Moreover, in Ou et al. (2024), it is proved that the loss that $D_\theta$ is trained on (see Eq. 30 and Eq. 4) is equivalent to:

$$\mathcal{L}_{\text{mask}}(\theta) = -\mathbb{E}_{\mathbf{x} \sim \mathbf{p}_{data}} \left[ \mathbb{E}_{\sigma \sim \text{Unif}(\mathbb{S}_L)} \left[ \log \left( \mathbb{P}(X_1^{\theta,\text{mask}} = \mathbf{x}|\sigma) \right) \right] \right]$$

$$= \mathbb{E}_{\sigma \sim \text{Unif}(\mathbb{S}_L)} \left[ D_{KL}(\mathbf{p}_{data}||\mathbb{P}(X_1^{\theta,\text{mask}} = \cdot|\sigma)) \right] + H(\mathbf{p}_{data}),$$

where $H$ is the Shannon Entropy of $\mathbf{p}_{data}$. This is minimized with value $H(\mathbf{p}_{data})$ if and only if $\mathbb{P}(X_1^{\theta,\text{mask}} = \cdot|\sigma) = \mathbf{p}_{data}, \forall \sigma \in \mathbb{S}_L$; that is, if every choice of unmasking order exactly recovers the data distribution.

It becomes clear that if the training objective used for a Masked Diffusion Model was made uniformly 0, every choice of unmasking order would exactly recover the data distribution (the KL divergence is 0 if and only if the distributions are equal - see e.g. Budhiraja & Dupuis (2019) Lemma 2.1). In practice, however, $D_\theta$ is far from perfect (and even if it were, it is trained using samples form $\mathbf{p}_{data}$, so would just recover those samples). As such, not all such orders will be created equal - that is there will be denoising orders $\sigma, \hat{\sigma} \in \mathbb{S}_L$ such that

$$D_{KL}(\mathbf{p}_{data}||\mathbb{P}(X_1^{\theta,\text{mask}} = \cdot|\sigma)) >> D_{KL}(\mathbf{p}_{data}||\mathbb{P}(X_1^{\theta,\text{mask}} = \cdot|\hat{\sigma})).$$

This was observed empirically in Ou et al. (2024) Appendix J.4, Shih et al. (2022), and Li et al. (2021) Section 6.

## D.4 COMPARISON WITH OTHER SAMPLING METHODS

Here we discuss how existing sampling methods fall under the P2 framework as outlined in Table 1.

Ancestral sampling disables the remasking by setting the Unmasked Planner ($G_U$) to always output 1, i.e., the probability that an unmask token should be kept is always 1, and the mask planner $G_M$ functions as a uniform sampler as it randomly selects mask positions. Greedy ancestral sampling improves open this by using the denoiser $\text{Cat}(z^j; D_\theta^j(\mathbf{x}))$ as the mask planner $G_M^j(\mathbf{z}, \mathbf{x})$. DFM sampling randomly selects positions, and enables remasking by introducing a tunable stochasticity strength $\eta$. RDM functions identically to our self-planning by using the denoiser for both mask and unmask planning but it omits the stochasticity control with the default stochasticity strength $\eta = 1$. DDPD introduces external planners and purely relies on the planner for both mask and unmask position planning with default stochasticity strength $\eta = 1$. Crucially, it disallows for the possibility of mask-informed planning and the decomposition of $G_\phi$ into $G_U$ and $G_M$. As it is the most similar work to ours in the existing literature, here we provide a thorough comparison with DDPD Liu et al. (2024).

Given that our objective is to plan a denoising order assuming access to a Masked Diffusion Model for our denoiser (as with DDPD-MaskD) and not to train a uniform diffusion-based denoiser from scratch (as with DDPD-DFM-Uni), we focus on their framework in the former setting.

Even with DDPD-MaskD, the framework uses a "uniform discrete diffusion" Eq. 27 as the starting-point for their token-wise forward noising process, as opposed to the "masked diffusion" forward noising process Eq. 1 used in our work. They modify the state space $S^L = \{1, \ldots, d-1\}^L$ to $\tilde{S}^L$, where $\tilde{S} = S \times \{N, D\}$. For $(\mathbf{y}, \mathbf{z}) \in \tilde{S}^L$, $(y^i, z^i)$ denotes the pair describing the state $y^i \in S$ in of $i$'th token and $z^i \in \{N, D\}$ denotes whether that token is noise ($N$) or data ($D$). They then modify the forward noising process to:

$$p_t((j, \zeta)|i) = \alpha_t \text{Cat}((j, \zeta); \delta(i, D)) + \frac{1 - \alpha_t}{d - 1} \text{Cat}(\zeta; \delta(N)), \quad i, j \in S, \quad \zeta \in \{N, D\},$$

see Equation (17) therein.

Thus, their reference distribution $\pi \in \Delta^{(d+1)^L}$ is given by $\pi = \text{Unif}(S^L) \otimes \delta_{N^L}$, where $N^L \in \{N, D\}^L$ consists of all $N$'s, and the corresponding backwards processes' $S^L$ marginal is initialized at the reference distribution $\text{Unif}(S^L)$ as opposed to $[\delta_{\mathbf{m}}]^L$ as in our setting.

They approximate a resulting true backward process on $S^L$'s rate matrix $\overleftarrow{Q}_t$ (given by Proposition 3.1 therein) with $Q_t^{\theta,\phi,\text{DDPD}}$ given by, for $\mathbf{x} \neq \mathbf{y}$:

$$Q_t^{\theta,\phi,\text{DDPD}}(\mathbf{y},\mathbf{x}) = \begin{cases} -\frac{\frac{d\alpha_{1-t}}{dt}}{1-\alpha_{1-t}} \sum_{i=1}^{L} \Big\{ \text{Cat}(N; G_{\phi,\text{DDPD}}^i(\mathbf{x})) \\ \qquad \times \mathbb{E}_{Z \sim G_\phi(\mathbf{x})}[\text{Cat}(y^i; D_\theta^i(\mathbf{x}^{Z,-i,\mathbf{m}}))] \Big\} & , d_{HAM}(\mathbf{x},\mathbf{y}) = 1, x^i \neq y^i \\ 0 & , \text{otherwise} \end{cases}$$

where $D_\theta : \mathcal{V}^L \to (\Delta^d)^L$ is a denoiser for a masked diffusion model trained via the ELBO Eq. 4. Here for $\mathbf{x} \in S^L$, $\mathbf{z} \in \{N, D\}^L$, $\mathbf{x}^{\mathbf{z},-i,\mathbf{m}} \in \mathcal{V}^L$ is obtained from $\mathbf{x}$ via:

$$[\mathbf{x}^{\mathbf{z},-i,\mathbf{m}}]^j = \begin{cases} \mathbf{m}, & z^j = N \\ x^j, & z^j = D, j \neq i \\ \mathbf{m}, & j = i \end{cases}.$$

$G_{\phi,\text{DDPD}} : S^L \to (\Delta^2)^L$ is another neural network with $\text{Cat}(N; G_{\phi,\text{DDPD}}^i(\mathbf{x}))$ approximating the probability that the $i$'th coordinate of $\mathbf{x} \in S^L$ is noise, and is trained via Eq. 30 with $E(\mathbf{x}_0) = E^{\text{DDPD}}(\mathbf{x}_0)$ given by:

$$E^{\text{DDPD}}(\mathbf{x}_0) = E_P^{\text{DDPD}}(\mathbf{x}_0) + E_D^{\text{DDPD}}(\mathbf{x}_0)$$

$$E_P^{\text{DDPD}}(\mathbf{x}_0) = -\int_0^1 \frac{\frac{d\alpha_t}{dt}}{1-\alpha_t} \mathbb{E}_{(\tilde{X}_t, Z_t) \sim P_t^{\text{DDPD}}(\cdot|\delta((\mathbf{x}_0, D^L)))} \left[ \sum_{i=1}^{L} \log \text{Cat}(Z_t^i; G_{\phi,\text{DDPD}}^i(\tilde{X}_t)) \right] dt$$

$$E_D^{\text{DDPD}}(\mathbf{x}_0) = -\int_0^1 \frac{\frac{d\alpha_t}{dt}}{1-\alpha_t} \mathbb{E}_{(\tilde{X}_t, Z_t) \sim P_t^{\text{DDPD}}(\cdot|\delta((\mathbf{x}_0, D^L)))} \Bigg[$$

$$\sum_{i=1, Z_t^i = N}^{L} \mathbb{E}_{\hat{Z} \sim G_{\phi,\text{DDPD}}(\tilde{X}_t)} \left[ \log \text{Cat}(\mathbf{x}_0^i; D_\theta^i(\tilde{X}_t^{\hat{Z},-i,\mathbf{m}})) \right] \Bigg] dt,$$

where for $\mathbf{y} \in S^L$, $\mathbf{z} \in \{N, D\}^L$:

$$P_t((\mathbf{y},\mathbf{z})|\delta((\mathbf{x}_0, D^L))) := \alpha_t \prod_{i=1}^{L} \text{Cat}((y^i, z^i); \delta((x_0^i, D))) + \frac{(1-\alpha_t)}{(d-1)^L} \prod_{i=1}^{L} \text{Cat}(z^i; \delta(N)).$$

Note that in the above ELBO, $E_D^{\text{DDPD}}$ is slightly modified from what which they present in Theorem 4.1. As written, they would take the expected value with respect to $G_{\phi,\text{DDPD}}$ inside the second log, which requires $2^{L-1}$ function evaluations of $D_\theta$. When the denoiser $D_\theta$ is given by that of a masked diffusion, one should instead use the above, which can be readily arrived at the same proof with an extra application of Jensen's inequality.

Comparing this with our Proposition Eq. 1, the comparison between DDPD and P2 becomes evident: $E_P^{\text{DDPD}}(\mathbf{x}_0)$ is playing the role of $E_{UP}(\mathbf{x}_0) + E_{MP}(\mathbf{x}_0)$ (that is, it yields the training objective for the Planner) and $E_D^{\text{DDPD}}(\mathbf{x}_0)$ is playing the role of $E_D(\mathbf{x}_0)$ (that is, it yields the training objective for the denoiser). However, we note the following key distinguishing factors:

1. In P2, $\mathcal{E}_D$ is the same as the ELBO originally used to train the denoiser $D_\theta$: that is, $D_\theta$ has already be trained to maximize $\mathbb{E}_{x_0 \sim \mathbf{p}_{\text{data}}}[\mathcal{E}_D(\mathbf{x}_0)]$. Meanwhile, $E_D^{\text{DDPD}}$ depends on the output of $G_{\phi,\text{DDPD}}$, increasing the importance of the role of planner in the quality of the generations output. For this reason, DDPD must train an external Planner whose model size is comparable to that of the denoiser - they are essentially asking the planner to play a role akin to the denoiser in a uniform diffusion model. Meanwhile, due to the "flipped" importance of the roles of the planner and denoiser in P2, we show that we can use lightweight BERT models or even the denoiser itself as an effective Planner. See Table S5, where we confirm DDPD's inability to make use of such lightweight models.

2. In P2, we separate the Planner's training objective into two components. This is natural because our planner may use information both from the partially masked data $X_t$ and the output of the denoiser. Meanwhile, in DDPD, the Planner only has access to $\tilde{X}_t$-unmasked data perturbed by random flips of its tokens. Because DDPD's generation process is grounded in a uniform diffusion process, there is no ability to separate the Planner into unmasked and masked components as we do in Section Eq. 3.2. In particular, their framework does not allow for a general enough planner to introduce our stochasticity strength parameter $\eta$ and design an algorithm analogous to the P2 Sampler Algorithm 1.

The practical differences between DDPD and P2 are further elucidated by comparing their Gillespie sampling strategy (Algorithm 1 therein) with ours (see Algorithm 5). For convenience, we reproduce it here.

Letting $\hat{G}_{\phi,\text{DDPD}} : S^L \to \Delta^L$ be given by $\hat{G}_{\phi,\text{DDPD}}^j(\mathbf{x}) = \frac{\text{Cat}(N; G_{\phi,\text{DDPD}}^j(\mathbf{x}))}{\sum_{j=1}^{L} \text{Cat}(N; G_{\phi,\text{DDPD}}^j(\mathbf{x}))}$, DDPD's Gillespie sampling algorithm is given by Algorithm 4.

---

**Algorithm 4** DDPD Sampler

---

1: **init** $i \leftarrow 0$, $\mathbf{x}_0 \sim \text{Unif}(S^L)$, planner $G_{\phi,\text{DDPD}}$, denoiser $D_\theta$, maximum steps $T$
2: **for** $t = 1 : T$ **do**
3:      **Plan** Sample dimension $i \sim \hat{G}_{\phi,\text{DDPD}}(\mathbf{x}_t)$
4:      **Denoise** Sample $\mathbf{z} \sim G_{\phi,\text{DDPD}}$
5:      Sample $y^i \sim D_\theta^i(\mathbf{x}_t^{\mathbf{z},-i,\mathbf{m}})$
6:      Update: $\mathbf{x}_{t+1}^i \leftarrow y^i$
7: **end for**
8: **return** $\mathbf{x}_T$

---

As is clear from Algorithm 4, in DDPD, the input to the Planner only depends on some unmasked, randomly flipped sequence of tokens, and does not depend on the output of the denoiser, and the input to the denoiser is entirely dependent on the output of the planner. Meanwhile, in P2, the Planner may use the both the information about the partially unmasked sequence (whose unmasked tokens all result from samples from the denoiser) and the output of the denoiser, and the input to the denoiser only depends on the output of the planner insofar as it may choose to remask a single token. We note that difficulty of the precise task performed by DDPD's planner was recently shown to be the reason for MDMs performance over uniform diffusion models in Amin et al. (2025). In Proposition 5.1 they essentially show that if one conditions on whether each position in a sequence of unmasked tokens is clean or noise, uniform diffusion models reverse to masked diffusion. So the ability of the model to make this distinction is the bottleneck preventing uniform diffusion models from performing comparably to MDMs, and likely the same reason for P2-BERT, P2-train, and P2-self's superior performance to DDPD as evidenced in Table S5. For ease of comparison, we derive the corresponding Gillespie sampling scheme for P2 in the forthcoming §D.5.

## D.5 DERIVING THE P2 GILLESPIE SCHEME ALGORITHM 5

Here, for ease of comparison with the works discussed in Subsection D.4 and to motivate the connection between the sampling scheme described in Eq. 5 and the practical top-k sampling scheme Alg. 1, we derive the Gillespie sampler for the continuous time limit of P2.

Let $\{\tau_k\}_{k \in \mathbb{N}}$ be the jump times for the CTMC $X^{\theta,\phi}$ with rate matrix $Q^{\theta,\phi}$ as described in Equation Eq. 32 (see Appendix C.2). To derive a Gillespie sampling scheme, we need to find the transition probabilities for the effective jump chain as described in Eq. 25. We first need to obtain the diagonal entries for the jump matrix $Q^{\theta,\phi}$. We have for $\mathbf{x} \in \mathcal{V}^L$:

$$- \sum_{\mathbf{y} \neq \mathbf{x}} Q_t^{\theta,\phi}(\mathbf{y}, \mathbf{x})$$

$$= \frac{\frac{d\alpha_{1-t}}{dt}}{1 - \alpha_{1-t}} \sum_{i=1}^{L} \sum_{y^i = 1, y^i \neq x^i}^{d-1} F_{\theta,\phi}^i(\mathbf{x}^{-i,y^i}, \mathbf{x}) \text{Cat}(y^i; \hat{D}_\theta^i(\mathbf{x}))$$

$$= \frac{\frac{d\alpha_{1-t}}{dt}}{1 - \alpha_{1-t}} \sum_{i=1}^{L} \Bigg[ \text{Cat}(x^i; \delta(\mathbf{m})) \sum_{y^i = 1, y^i \neq x^i}^{d-1} \mathbb{E}_{Z \sim D_\theta(\mathbf{x})}[G_\phi^i(Z^{-i,y^i})] \text{Cat}(y^i; D_\theta^i(\mathbf{x}))$$

$$+ \frac{(1 - \text{Cat}(x^i; \delta(\mathbf{m})))}{1 - \text{Cat}(x^i; D_\theta^i(\mathbf{x}^{-i,\mathbf{m}}))} \sum_{y^i = 1, y^i \neq x^i}^{d-1} \mathbb{E}_{Z \sim D_\theta(\mathbf{x})}[G_\phi^i(Z, x)] \text{Cat}(y^i; D_\theta^i(\mathbf{x}^{-i,\mathbf{m}})) \Bigg]$$

$$= \frac{\frac{d\alpha_{1-t}}{dt}}{1 - \alpha_{1-t}} \sum_{i=1}^{L} \Bigg\{ \text{Cat}(x^i; \delta(\mathbf{m})) \mathbb{E}_{Z \sim D_\theta(\mathbf{x})}[G_\phi^i(Z, \mathbf{x})] + (1 - \text{Cat}(x^i; \delta(\mathbf{m}))) \mathbb{E}_{Z \sim D_\theta(\mathbf{x})}[G_\phi^i(Z, \mathbf{x})] \Bigg\}$$

$$= \frac{\frac{d\alpha_{1-t}}{dt}}{1 - \alpha_{1-t}} \sum_{i=1}^{L} \mathbb{E}_{Z \sim D_\theta(\mathbf{x})}[G_\phi^i(Z, \mathbf{x})]$$

$$= Q_t^{\theta,\phi}(\mathbf{x}, \mathbf{x}).$$

Then for $\mathbf{x} \neq \mathbf{y} \in \mathcal{V}^L$, $k \in \mathbb{N}$, and $t \in [0, 1]$:

$$\mathbb{P}(X_{\tau_{k+1}}^{\theta,\phi} = \mathbf{y} | X_{\tau_k}^{\theta,\phi} = \mathbf{x}, \tau_{k+1} = t) = \frac{F_{\theta,\phi}^i(\mathbf{y}, \mathbf{x}) \text{Cat}(y^i; \hat{D}_\theta^i(\mathbf{x}))}{\sum_{i=1}^{L} \mathbb{E}_{Z \sim D_\theta(\mathbf{x})}[G_\phi^i(Z, x)]},$$

when $d_{HAM}(\mathbf{x}, \mathbf{y}) = 1$ and $x^i \neq y^i$ and 0 when the Hamming distance $d_{HAM}(\mathbf{x}, \mathbf{y}) \neq 1$. We note that this is and independent of $t$ and $k$.

Then, for $j \in [L] = \{1, \ldots, L\}$ and $\mathbf{x}, \mathbf{y}, k, t$ as before:

$$\mathbb{P}([X^{\theta,\phi}_{\tau_{k+1}}]^j \neq [X^{\theta,\phi}_{\tau_k}]^j | X^{\theta,\phi}_{\tau_k} = \mathbf{x}, \tau_{k+1} = t)$$

$$= \sum_{\mathbf{y} \in \mathcal{V}^L : y^j \neq x^j} \mathbb{P}(X^{\theta,\phi}_{\tau_{k+1}} = \mathbf{y} | X^{\theta,\phi}_{\tau_k} = \mathbf{x})$$

$$= \sum_{y^j=1, y^j \neq x^j}^{d-1} F^j_{\theta,\phi}(\mathbf{x}^{-j,y^j}, \mathbf{x}) \mathrm{Cat}(y^j; \hat{D}^j_\theta(\mathbf{x})) / \left( \sum_{i=1}^L \mathbb{E}_{Z \sim D_\theta(\mathbf{x})}[G^i_\phi(Z, x)] \right)$$

$$= \frac{\mathbb{E}_{Z \sim D_\theta(\mathbf{x})}[G^j_\phi(Z, \mathbf{x})]}{\sum_{i=1}^L \mathbb{E}_{Z \sim D_\theta(\mathbf{x})}[G^i_\phi(Z, \mathbf{x})]}$$

$$=: P(j, \mathbf{x})$$

and for $y^j \in \mathcal{V}$ with $y^j \neq x^j$:

$$\mathbb{P}([X^{\theta,\phi}_{\tau_{k+1}}]^j = y^j | X^{\theta,\phi}_{\tau_k} = \mathbf{x}, \tau_{k+1} = t, [X^{\theta,\phi}_{\tau_{k+1}}]^j \neq [X^{\theta,\phi}_{\tau_k}]^j)$$

$$= \frac{\mathbb{P}([X^{\theta,\phi}_{\tau_{k+1}}]^j = y^j, [X^{\theta,\phi}_{\tau_{k+1}}]^j \neq [X^{\theta,\phi}_{\tau_k}]^j | X^{\theta,\phi}_{\tau_k} = \mathbf{x}, \tau_{k+1} = t)}{\mathbb{P}([X^{\theta,\phi}_{\tau_{k+1}}]^j \neq [X^{\theta,\phi}_{\tau_k}]^j | X^{\theta,\phi}_{\tau_k} = \mathbf{x}, \tau_{k+1} = t)}$$

$$= \sum_{\mathbf{y}' \in \mathcal{V}^L : [y']^j = y^j \neq x^j} \frac{\mathbb{P}(X^{\theta,\phi}_{\tau_{k+1}} = \mathbf{y}' | X^{\theta,\phi}_{\tau_k} = \mathbf{x})}{\mathbb{P}([X^{\theta,\phi}_{\tau_{k+1}}]^j \neq [X^{\theta,\phi}_{\tau_k}]^j | X^{\theta,\phi}_{\tau_k} = \mathbf{x}, \tau_{k+1} = t)}$$

$$= \frac{F^j_{\theta,\phi}(\mathbf{x}^{-j,y^j}, \mathbf{x}) \mathrm{Cat}(y^j; \hat{D}^j_\theta(\mathbf{x}))}{\mathbb{E}_{Z \sim D_\theta(\mathbf{x})}[G^j_\phi(Z, x)]}$$

$$= \left( \mathrm{Cat}(x^j; \delta(\mathbf{m})) \mathbb{E}_{Z \sim D_\theta(\mathbf{x})}[G^j_\phi(Z^{-j,y^j}, x)] \mathrm{Cat}(y^j; D^j_\theta(\mathbf{x})) \right.$$

$$\left. + (1 - \mathrm{Cat}(x^j; \delta(\mathbf{m}))) \mathbb{E}_{Z \sim D_\theta(\mathbf{x})}[G^j_\phi(Z, x)] \frac{\mathrm{Cat}(y^j; D^j_\theta(\mathbf{x}^{-i,\mathbf{m}}))}{1 - \mathrm{Cat}(x^j; D^j_\theta(\mathbf{x}^{-j,\mathbf{m}}))} \right)$$

$$/ \mathbb{E}_{Z \sim D_\theta(\mathbf{x})}[G^j_\phi(Z, x)]$$

$$= \mathrm{Cat}(x^j; \delta(\mathbf{m})) \frac{\mathbb{E}_{Z \sim D_\theta(\mathbf{x})}[G^j_\phi(Z^{-j,y^j}, x)]}{\mathbb{E}_{Z \sim D_\theta(\mathbf{x})}[G^j_\phi(Z, x)]} \mathrm{Cat}(y^j; D^j_\theta(\mathbf{x}))$$

$$+ (1 - \mathrm{Cat}(x^j; \delta(\mathbf{m}))) \frac{\mathrm{Cat}(y^j; D^j_\theta(\mathbf{x}^{-j,\mathbf{m}}))}{1 - \mathrm{Cat}(x^j; D^j_\theta(\mathbf{x}^{-j,\mathbf{m}}))}$$

$$=: \tilde{P}(j, x, y^j).$$

Thus, an exact Gillespie sampling scheme would be given by Gillespie (1977; 1976):

When the chain is in state $x \in \mathcal{V}^L$, sample a dimension $i \sim \hat{P}(\cdot, x)$ to change, then sample a value $y^j \sim \tilde{P}(i, x, \cdot)$ to change it to.

In practice it is impractical to approximate these expected values with respect to $Z \sim D_\theta(\mathbf{x})$, as this would require many function evaluations of the denoiser. However, assuming that the token space is large, conditioning on the value of one coordinate should have little impact on the expected output of the Planner over the entire sequence (see e.g. the discussion under Proposition 3.5. and Appendix E.4 in Liu et al. (2024)). Given that Algorithm 5 is provided for the purpose of exposition and in practice we make use of Algorithm 1 in sampling, we use this intuition to formally approximate:

$$\tilde{P}(j, \mathbf{x}, y^j) \approx \mathrm{Cat}(x^j; \delta(\mathbf{m})) \mathrm{Cat}(y^j; D^j_\theta(\mathbf{x})) + (1 - \mathrm{Cat}(x^j; \delta(\mathbf{m}))) \frac{\mathrm{Cat}(y^j; D^j_\theta(\mathbf{x}^{-j,\mathbf{m}}))}{1 - \mathrm{Cat}(x^j; D^j_\theta(\mathbf{x}^{-j,\mathbf{m}}))}$$

and

$$P(j, \mathbf{x}) \approx \frac{\mathbb{E}_{Z \sim D_\theta(\mathbf{x})}[G^j_\phi(Z, \mathbf{x})]}{\sum_{i=1}^L \mathbb{E}_{Z \sim D_\theta(\mathbf{x})}[G^i_\phi(Z, \mathbf{x})]} \approx \mathbb{E}_{Z \sim D_\theta(\mathbf{x})}[\hat{G}^j(Z, \mathbf{x})],$$

where $: \hat{G}_\phi : \mathcal{V}^L \times \mathcal{V}^L \to \Delta^L$ is given by:

$$\hat{G}^j_\phi(\mathbf{z}, \mathbf{x}) := \frac{G^j_\phi(\mathbf{z}, \mathbf{x})}{\sum_{j=1}^L G^j_\phi(\mathbf{z}, \mathbf{x})}.$$

We then arrive at Algorithm 5.

---

**Algorithm 5** Our Gillespie Sampler

---
1: **Initialize:** $t \leftarrow 0, \mathbf{x}_0 \leftarrow (\mathbf{m}, \ldots, \mathbf{m})$, planner $G_\phi$, denoiser $D_\theta$, maximum steps $T$
2: **for** $t = 1 : T$ **do**
3:  **Plan** Sample $\mathbf{z} \sim D_\theta(\mathbf{x}_t)$
4:  Sample dimension $i \sim \hat{G}_\phi(\mathbf{z}, \mathbf{x}_t)$
5:  **Denoise**
6:  **if** $x_t^i \neq \mathbf{m}$ **then**
7:    $x_t^i \leftarrow \mathbf{m}$
8:    Resample $z^i \sim D_\theta^i(\mathbf{x}_t)$
9:    $x_{t+1}^i \leftarrow z^i$
10:  **else**
11:    $x_{t+1}^i \leftarrow z^i$
12:  **end if**
13: **end for**
14: **return** $\mathbf{x}_T$

---

Observe that Algorithm 1 is simply the result of modifying Algorithm 5 so that $\hat{G}$ is replaced by $\tilde{G}_\eta$ (allowing for $\eta \neq 1$), dropping the requirement that a token is denoised immediately after remasking, and replacing faithful sampling from $\tilde{G}_\eta$ with top-k sampling.

# E  IMPLEMENTATION DETAILS

In Listing 1, we provide a self-contained PyTorch implementation of our *Path-Planning Sampling* procedure. The code consists of three core components, each addressing a distinct step in the sampling process:

**1) `topk_lowest_masking`:** Given a matrix of scalar scores, this function returns a boolean mask that flags the "lowest-scoring" positions per row. The user can specify how many positions should be re-masked by providing a `cutoff_len` tensor. Internally, the function sorts the score matrix and determines the threshold score for each row before comparing every score to this cutoff.

**2) `stochastic_sample_from_categorical`:** This function draws samples from a categorical distribution using Gumbel noise. It first applies Gumbel noise to the input logits (if a non-zero temperature is specified), then computes the log-softmax to obtain token probabilities. The sampled tokens and their corresponding log probabilities are returned.

**3) `path_planning_sampling`:** Positions initially set to the `mask_token_id` are iteratively predicted and updated. At each iteration, we:

1. Compute model logits and identify positions that remain masked.

2. Sample from the model outputs via `stochastic_sample_from_categorical`.

3. Integrate a `planner` (if provided) to re-score predictions for currently unmasked positions, giving users the flexibility to incorporate any additional guidance or constraints.

4. Construct a `score` and re-mask positions with the lowest scores. Fixed positions are ignored by assigning them infinite scores so that they cannot be re-masked.

5. Scale the scores of unmasked positions by the factor $\eta$, which adjusts how aggressively new tokens are updated.

The function continues for `num_steps`, revealing high-confidence predictions and re-masking uncertain positions. Finally, any remaining masks are replaced with the last sampled tokens. The key parameters are:

- `xt`: The initial token matrix of shape $[B, L]$, containing masked tokens.

- `model`: A callable mapping tokens to logits.

- `tokenizer`: Provides the special `mask_token_id`.

- `num_steps`: Number of refinement iterations.

- `tau`: Temperature for controlling sampling noise.

- `kappa_fn`: A schedule function in $[0, 1]$ that dictates how many positions remain masked vs. unmasked over time.

- `eta`: A multiplier for scores in unmasked positions.

- `planner`: An optional model for additional re-scoring.

- `score_type`: Either `'confidence'` (uses log probabilities) or `'random'` (random re-masking).

Listing 1: Path-Planning Sampling procedure in PyTorch

```python
import torch

def topk_lowest_masking(scores, cutoff_len):
    sorted_scores, _ = scores.sort(dim=-1)
    threshold = sorted_scores.gather(dim=-1, index=cutoff_len)
    return scores < threshold

def stochastic_sample_from_categorical(logits, temperature=1.0,
     noise_scale=1.0):
    logits = logits.double()
    if temperature != 0.0:
        gumbel = -torch.log(-torch.log(torch.rand_like(logits) + 1e-8) + 1e
            -8)
        logits = logits / temperature + noise_scale * gumbel
    scores, tokens = logits.log_softmax(dim=-1).max(dim=-1)
    return tokens, scores

@torch.inference_mode()
@torch.cuda.amp.autocast()
def path_planning_sampling(
    xt,
    model,
    tokenizer,
    num_steps,
    tau=1.0,
    kappa_fn=lambda t: t,
    eta=1.0,
    planner=None,
    score_type='confidence'
):
    fix_mask = (xt != tokenizer.mask_token_id)
    dt = 1.0 / num_steps

    for step in range(1, num_steps + 1):
        t = step * dt
        kappa_t = kappa_fn(t)
        logits = model(xt).double()

        last_mask = (xt == tokenizer.mask_token_id)
        unmask_candidates = ~last_mask & ~fix_mask

        x0, logp = stochastic_sample_from_categorical(logits, temperature=
            tau)

        if planner is not None:
            planner_logits = planner(x0).double()
            planner_logp = planner_logits.log_softmax(dim=-1).gather(-1, x0.
                unsqueeze(-1)).squeeze(-1)
            logits[unmask_candidates] = planner_logits[unmask_candidates]
            logp[unmask_candidates] = planner_logp[unmask_candidates]

        if score_type == 'confidence':
            score = logp
        elif score_type == 'random':
            score = torch.rand_like(logp).log()
        else:
            raise ValueError("Invalid score_type.")

        score = score.masked_fill(fix_mask, float('inf'))
        score[unmask_candidates] *= eta

        num_to_mask = ((~fix_mask).sum(dim=1, keepdim=True).float() * (1 -
            kappa_t)).long()
        mask = topk_lowest_masking(score, num_to_mask)
```

```
        xt[mask] = tokenizer.mask_token_id

        mask_to_x0 = last_mask & ~mask
        xt[mask_to_x0] = x0[mask_to_x0]

    remaining_mask = (xt == tokenizer.mask_token_id)
    xt[remaining_mask] = x0[remaining_mask]

    return xt
```

# F  EXPERIMENTAL DETAILS

## F.1  PROTEIN GENERATION EVALUATION DETAILS

**Setup**  We compare our method with state-of-the-art protein sequence generation models, including three discrete diffusion models—DPLM (Wang et al., 2024), EvoDiff (Alamdari et al., 2024), and ESM3 (Hayes et al., 2025)—and an autoregressive model, ProGen2 (Nijkamp et al., 2022), across three model sizes: small, medium, and large. Additionally, we benchmark masked language models, ESM2 (Lin et al., 2023), at three scales: 150M, 650M, and 3B parameters.

For our path-planning algorithm (P2), we vary the stochasticity strength from 1.0 to 2.0 in increments of 0.1 and report optimal results. Baselines are evaluated with default sampling strategies. Since ESM2 lacks a masked diffusion loss, it uses ancestral sampling. Each model generates 100 sequences for sequence lengths in $[200, 300, \ldots, 800]$. DPLM employs a sequence length matching the number of sampling steps and a temperature of 0.9, with rejection-resampling disabled for fairness. ESM3 is sampled with a temperature of 1, a cosine schedule, top-$p = 1$, and 500 steps. Special tokens are removed to ensure valid amino acid sequences.

**Evaluation.**  Protein sequence generation quality is evaluated via protein folding models, using ESMFold (Lin et al., 2023) as a proxy for structural stability. We extract three folding metrics:

- **pLDDT** (predicted Local Distance Difference Test): Measures local structural accuracy.
- **pTM** (predicted Template Modeling): Assesses global structural plausibility.
- **pAE** (predicted Alignment Error): Evaluates overall compactness.

A sequence can achieve high pLDDT while exhibiting poor global compactness (high pAE). To ensure robust evaluation, we define *foldability* as the proportion of sequences satisfying pLDDT $> 80$, pTM $> 0.7$, and pAE $< 10$. This metric effectively identifies low-quality sequences, such as repetitive patterns (e.g., "ABABABAB"), which tend to have high pAE.

Beyond folding scores, we compute:

- **Token entropy**, excluding tokens not present in generated sequences.
- **Sequence diversity**, defined as $1-$ pairwise sequence identity within a batch. Since all sequences in a batch share equal length, no sequence alignment is needed.

These metrics detect mode collapse, where models generate highly repetitive sequences.

### F.1.1  TRAINING DETAILS OF THE 150M MDM.

We train a 150M mask diffusion model on protein sequences for the ablation of self-planning. The 150M MDM is trained using the open-sourced DPLM code[4]. We use the same transformer architecture as DPLM-150M as well as ESM2-150M. We train our MDM from scratch for 500k steps with a total of 320K tokens in each iteration, which is achieved by multi-GPU and multi-node training with gradient accumulation. The training data is Uniref50, consisting of around 40M protein sequences with 50% sequence-identity cutoff, namely, the sequences in uniref50 are at least higher than 50% dissimilar. Uniref50 is widely used for training protein language models.

### F.1.2  TRAINING DETAILS FOR P2 TRAIN

For results on P2 train, we fine-tune $T_\phi^i(\mathbf{z}, \mathbf{x})$ where $T_\phi^i(\mathbf{z}, \mathbf{x}) = \mathrm{Cat}(z^i; B^i(\mathbf{z}))$ for $B$ given by ESM-8M for 100k steps using $G_U = G_M = T_\phi$ in Alg. 2 with the same data and hyperparameter setup as for the 150M MDM. During sampling for P2 train, we take $G_U^i(\mathbf{z}, \mathbf{x}) = T_\phi^i(\mathbf{z}, \mathbf{x})$ and $G_M^i(\mathbf{z}, \mathbf{x}) = \mathrm{Cat}(z^i; D_\theta^i(\mathbf{x}))$ in Alg. 1.

---

[4]https://github.com/bytedance/dplm

### F.1.3 Computing the ELBO

The Evidence Lower Bound (ELBO) serves as the training objective of mask diffusion models and can be used to assess how well the model fits the data. The ELBO experiments are conducted on protein sequence generation tasks. We compute the negative ELBO for five planners, namely ESM-8M, ESM-35M, ESM-150M, ESM-650M, and ESM-3B, alongside the self-planning ELBO, using a weighted cross-entropy loss function to quantify reconstruction accuracy.

**Dataset Preparation.** We utilize sequences from the UniRef50 dataset, filtering to include only test sequences with lengths shorter than 300 residues to align with the experiments in Figure S3 and mitigate memory constraints. The dataset is loaded into a PyTorch DataLoader using a sequence length of 1022 tokens and a maximum token budget of 60,000. For consistent evaluation, we run the ELBO calculation over 20 independent simulations and report the average across these runs.

**Masking Strategy.** For each sequence, we randomly generate a mask ratio uniformly sampled from the range $[1/500, 1 - 1/500]$. Positions are masked based on this ratio, but masking is constrained to avoid altering non-maskable tokens (e.g., special symbols). The masked tokens are replaced with a designated mask token provided by the denoiser model.

**Loss Calculation.** To compute the ELBO, the denoiser and planner models predict the original tokens for both masked and unmasked positions. The cross-entropy loss is calculated separately for these categories. Both masked and unmasked loss values are weighted inversely by the mask ratio to ensure probabilistic consistency in the evaluation. Each model is evaluated across 20 independent simulations, and the average ELBO is reported to capture the robustness of the planners under stochastic settings.

### F.2 Language Generation Evaluation Details

**Tasks and Metrics.**

- **TriviaQA** (Joshi et al., 2017): reading comprehension (exact match).
- **LAMBADA** (Paperno et al., 2016): last-token prediction (accuracy).
- **GSM8K** (Cobbe et al., 2021): math reasoning (accuracy).
- **ROCStories** (Mostafazadeh et al., 2016): story infilling, evaluated by ROUGE-1/2/L (Lin, 2004).
- **HumanEval** (Bavarian et al., 2022): code completion, measured by pass@1.

**Example of Language generation Task** We provide Table S1 consisting of examples for the five language generation tasks.

**Setup.** We follow SMDM (Gong et al., 2025) and DiffuLLaMA (Nie et al., 2025) protocols. MDM (1.1B) and DiffuLLaMA (7B) are used as base models. We apply P2 with $\eta \in [0, 2.0]$ and report best-performing settings. Decoding follows standard ancestral sampling unless otherwise noted. For AR baselines lacking native infilling support, we use oracle length truncation. Evaluation is done using the LM Harness (Biderman et al., 2024).

**Baselines.** We report published results from GPT2-S/M, DiffuGPT, SEDD (Lou et al., 2023), Plaid1B (Gulrajani & Hashimoto, 2023), and LLaMA2 (Touvron et al., 2023). TinyLlama is also included as an open-source AR baseline.

**Implementation Notes.** For P2, stochasticity is critical to quality. For each model-task pair, we tune $\eta$ using a grid sweep and hold evaluation set fixed. We do not use instruction tuning or CoT prompting.

### F.3 RNA Generation Details

### F.4 RNA Evaluation Details

**Training.** We train a 150M-parameter MDM on 27M RNA sequences from RNACentral (Petrov, 2021) using a batch size of 320K tokens for 100K steps. The tokenizer and vocabulary follow RiNALMo (Penić et al., 2024).

**Evaluation.** We generate 100 RNA sequences of 100 base pairs. Predicted structures are obtained using the RNA folding model from Shen et al. (2024). Evaluation metrics include:

- **pLDDT** ($\uparrow$): predicted local structure confidence.
- **MFE** ($\downarrow$): minimum free energy of folded structure.
- **Entropy** ($\uparrow$): mean token entropy across positions.
- **GC Content** ($\uparrow$): proportion of guanine-cytosine nucleotides.

**Baselines.** We compare against RiNALMo-150M and RiNALMo-650M (Penić et al., 2024), two masked language models pretrained on RNA. We also include a reference set of 100 natural RNA sequences of matching length. For P2, we use BERT-Planning derived from RiNALMo-150M, sweeping $\eta \in [0, 2]$ with step size 0.02 and reporting the best-performing configuration.

Table S1: Examples from language understanding benchmarks.

| Metric | Question | Answer |
|---|---|---|
| LAMBADA | "Again, he left that up to you. However, he was adamant in his desire that it remain a private ceremony. He asked me to make sure, for instance, that no information be given to the newspaper regarding his death, not even an obituary. I got the sense that he didn't want anyone, aside from the three of us, to know that he'd even ___." | died |
| GSM8K | Weng earns $12 an hour for babysitting. Yesterday, she just did 50 minutes of babysitting. How much did she earn? | 10 |
| TriQA | The Dodecanese Campaign of WWII that was an attempt by the Allied forces to capture islands in the Aegean Sea was the inspiration for which acclaimed 1961 commando film? | The Guns of Navarone |
| ROCStories | Morgan and her family lived in Florida. They heard a hurricane was coming. **(Story infills here...)** They arrived and learned from the news that it was a terrible storm. They felt lucky they had evacuated when they did. | They decided to evacuate to a relative's house. |
| Code | ```python
from typing import List

def has_close_elements(numbers: List[float], threshold:
    float) -> bool:
    """
    Check if in given list of numbers, are any two numbers
        closer
    to each other than given threshold.

    >>> has_close_elements([1.0, 2.0, 3.0], 0.5)
    False
    >>> has_close_elements([1.0, 2.8, 3.0, 4.0, 5.0, 2.0],
        0.3)
    True
    """
    # Infill Code
    if distance < threshold:
        return True
    return False
``` | ```python
for idx, elem in enumerate(numbers):
    for idx2, elem2 in enumerate(numbers):

        if idx != idx2:
            distance = abs(elem - elem2)
``` |

Table S2: Results on breaking the reverse curse: Performance comparison of models on Description-ToName and NameToDescription tasks. Metrics include accuracy (Acc.) and BLEU scores (BLEU) for both same and reverse directions.

| | DescriptionToName | | NameToDescription | | | |
|---|---|---|---|---|---|---|
| | Same direction Acc. ↑ | Reverse direction Acc. ↑ | Same direction Acc. ↑ | BLEU ↑ | Reverse direction Acc. ↑ | BLEU ↑ |
| GPT3 (175B) | 97 | 0 | 50 | - | 0 | - |
| Llama-2 (13B) | 99 | 0 | - | 74 | - | 19 |
| T5 (3B) | **100** | 0 | 47 | **87** | 0 | 20 |
| MDM (1.1B) | 97 | 92 | **49** | 76 | **37** | 67 |
| MDM (1.1B) + **Path Planning (P2)** | 96 | **93** | 48 | 78 | 36 | **68** |

**Findings.** P2 improves MDM's structural quality beyond native baselines and pretrained models, while keeping sequence diversity nearly unchanged. Structure visualizations are provided in Appendix G.3.2.

**RNA MDM Training Implementation.** The RNA MDM follows the same discrete diffusion described in (Zheng et al., 2023). The MDM was trained using a machine mounted with 4 A100 GPUs, each with 40GB memory. The training implementation is otherwise identical to the second-stage fine-tuning described in (Wang et al., 2024), where we continued from a RiNALMo (Penić et al., 2024) checkpoint instead of ESM-2 (Lin et al., 2023).

# G ADDITIONAL RESULTS

## G.1 LANGUAGE GENERATION

### G.1.1 BREAKING THE REVERSE CURSE

**Benchmark.** Berglund et al. (2023) introduced the concept of the reverse curse, which refers to the difficulty of ARMs in generalizing bidirectional relationships. Specifically, this occurs when a model is trained on information in the form "A is B" but fails to infer the reverse relationship "B is A." For example, a model trained on the fact "Valentina Tereshkova was the first woman to travel to space" may not correctly answer the reverse question "Who was the first woman to travel to space?" This limitation raises concerns about whether large language models genuinely possess logical reasoning capabilities.

**Baselines.** We compare with the leading AR models including GPT3 (175B), Llama-2 (13B), and the T5 consisting of both bidirectional encoder and unidirectional decoder, finetuned on the reverse curse dataset. For

the MDM baseline, We use the existing MDM (1.1B) from Gong et al. (2025) with its default greedy ancestral sampling strategy.

**Setup.** It is observed in SMDM(Gong et al., 2025) that MDMs easily break the reverse curse, displaying near-perfect reverse accuracy where ARs achieve 0 accuracy. We follow SMDM(Gong et al., 2025) and evaluate MDMs on the same reverse curse dataset used by Berglund et al. (2023), which consists of fictitious statements in the format "⟨name⟩ is ⟨description⟩" and the reversals. We use the pretrained MDMs and baseline results from SMDM (Gong et al., 2025) which on these statements and assess their performance using questions not seen during training. Following the same protocol as (Berglund et al., 2023), we generate responses and report the exact match accuracy and use the BLEU metric (Papineni et al., 2002) to evaluate the quality of name-to-description generation (Lv et al., 2023).

**Results.** As shown in Table S2, both the T5 model and ARMs achieve zero accuracy and low BLEU scores with reverse queries. Equipping with P2, we successfully improve the accuracy of MDMs in Reverse direction of task Description To Name and the BLEU metric of Name To Description in both directions.

### G.1.2 ADDITIONAL COMPARISON AMONG SAMPLING METHODS

We provide an expanded ablation over sampling strategies for diffusion code generation in Table **??**, complementing the results in Table 6. We evaluate on two families of benchmarks. HumanEval and MBPP measure standard left-to-right code completion, while HumanEval+ and MBPP+ use strengthened unit tests to probe functional correctness under more adversarial cases. To stress the non-causal advantages of diffusion models, we additionally report infilling performance on HumanEval-Infill and SantaCoder-FIM, where models must complete missing spans given both left and right context. All methods share the same Open-dCoder base model and inference budget; only the sampling rule for choosing which masked positions to update differs. The compared samplers span common baselines and recent state-of-the-art. Vanilla Ancestral follows the standard stochastic reverse diffusion procedure, unmasking positions uniformly at each step. Greedy Ancestral replaces stochastic updates with always taking the argmax token, which typically improves short-horizon correctness but can cause premature commitment. Entropy-based Confidence prioritizes positions with low predictive entropy (high confidence), akin to confidence-ordered decoding used in MaskGIT-style samplers (Chang et al., 2022b). TopK-Margin (Kim et al., 2025) selects positions by the logit margin between the top two candidates, a stronger confidence proxy that has shown gains in recent dLLM work. Finally, P2-self-plan is our planner that jointly decides unmasking and selective remasking based on a lookahead objective, explicitly optimizing the global denoising path instead of applying a local heuristic. Across all six tasks and both Pass@1 and Pass@10, P2-self-plan consistently achieves the best performance. Relative to Vanilla Ancestral, P2 yields large gains on standard completion (e.g., +17.5 Pass@1 on HumanEval and +14.9 on MBPP) and also improves infilling and FIM, indicating that its path-level planning benefits both causal and non-causal settings. Greedy Ancestral and Entropy-based Confidence provide clear improvements over Vanilla, confirming that informed, confidence-driven ordering is important for dLLM sampling; however, their gains saturate because they remain purely myopic and cannot revise earlier low-quality decisions. TopK-Margin is competitive among heuristic baselines but still trails P2, suggesting that better confidence estimates alone are insufficient without explicit planning over future denoising dynamics. Overall, these results reinforce that the sampling algorithm is a first-order determinant of dLLM performance, and that P2 offers a robust, consistently superior default for both code completion and infilling.

| Method | HumanEval | | HumanEval+ | | MBPP | | MBPP+ | | HumanEval Infill | SantaCoder |
| --- | --- | --- | --- | --- | --- | --- | --- | --- | --- | --- |
| | P@1 | P@10 | P@1 | P@10 | P@1 | P@10 | P@1 | P@10 | P@1 | P@1 |
| **P2-self-plan** | **20.8** | **38.4** | **17.6** | **35.2** | **16.7** | **38.4** | **23.9** | **53.6** | **77.4** | **56.4** |
| Vanilla Ancestral | 3.3 | 18.3 | 3.2 | 15.2 | 1.8 | 13.2 | 2.9 | 21.8 | 72.7 | 53.8 |
| Greedy Ancestral | 9.3 | 31.1 | 8.1 | 28.7 | 5.3 | 29.0 | 8.7 | 41.5 | 75.1 | 53.7 |
| Entropy-based Confidence | 12.6 | 35.4 | 10.9 | 29.9 | 9.2 | 36.8 | 15.2 | 50.7 | 75.1 | 53.2 |
| TopK-Margin | 7.6 | 27.4 | 6.5 | 26.2 | 3.9 | 24.0 | 6.2 | 33.5 | 75.0 | 54.4 |

Table S3: Performance comparison across coding benchmarks for different sampling methods.

### G.2 PROTEIN GENERATION

#### G.2.1 PERFORMANCE ACROSS LENGTH CATEGORIES.

We analyze the performance of protein generation models across various sequence lengths, ranging from 200 to 800 base pairs. Certain models, such as ProGen, do not generate proteins of fixed lengths; therefore, we group results into length categories to facilitate meaningful comparisons. As shown in Figure S1, the performance of these models varies with length, highlighting their capabilities and limitations across diverse length categories.

#### G.2.2 ABLATION OVER MODEL SCALE FOR PROGEN2 AND DPLM.

#### G.2.3 ABLATION OF PATH PLANNING

**The Design Space of Path Planning**. Our Path Planning (P2) framework generalizes existing sampling strategies, including vanilla ancestral sampling, greedy ancestral sampling, RDM sampling, and DFM sampling,

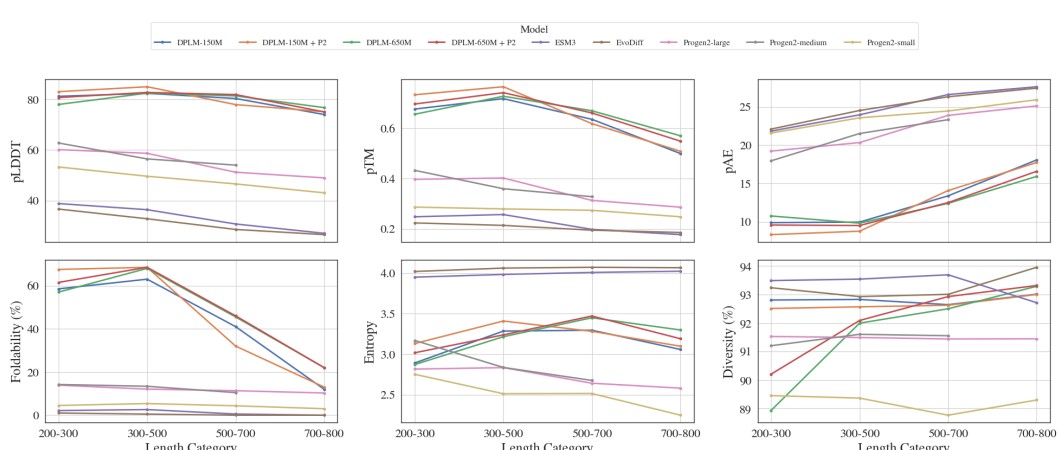

Figure S1: Protein Sequence Generation Benchmark: Performance across length categories (200–800).

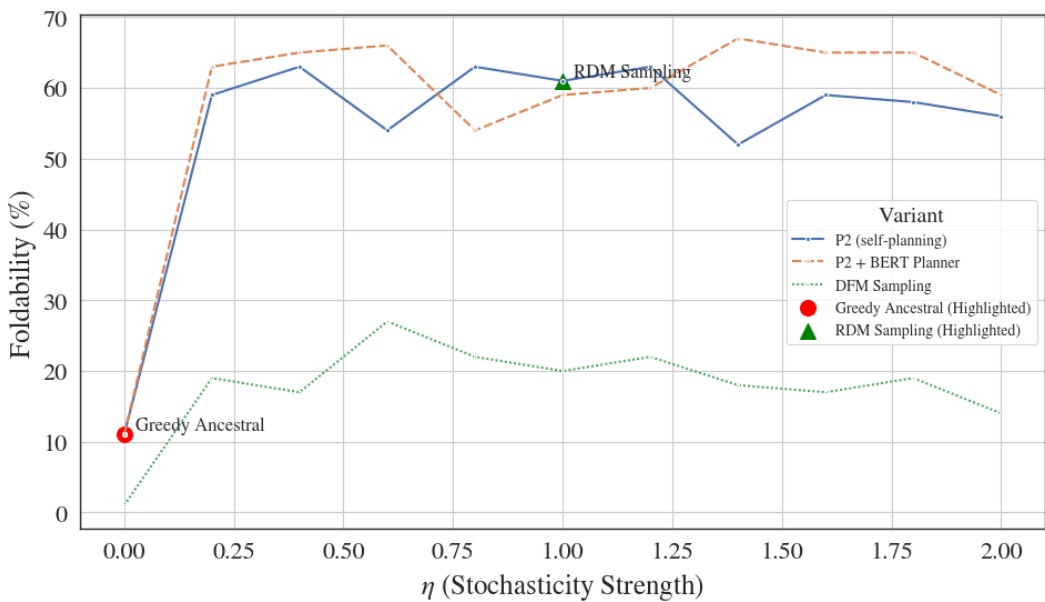

Figure S2: The Design Space of P2 (See Figure S5 for more). P2 Generalizes existing sampling algorithms with specific stochasticity strength and planner choice.

Table S4: Ablation on model scale for ProGen2 and DPLM. P2 (Trained Planner, 8M) consistently improves DPLM variants. Scaling alone does not ensure better performance.

| Model Variant | pLDDT↑ | pTM↑ | pAE↓ | Foldability (%)↑ | Entropy↑ | Diversity (%)↑ |
|---|---|---|---|---|---|---|
| ProGen2-small | 49.38 | 0.28 | 23.38 | 4.48 | 2.55 | 89.31 |
| ProGen2-medium | 57.94 | 0.38 | 20.81 | 12.75 | 2.91 | 91.45 |
| ProGen2-large | 55.07 | 0.35 | 22.00 | 11.87 | 2.73 | 91.48 |
| DPLM-150M | 80.23 | 0.65 | 12.07 | 48.14 | 3.14 | 92.80 |
| **+ P2-Train** | **83.45** | **0.72** | **10.15** | **58.86** | **3.35** | **92.69** |
| DPLM-650M | 79.53 | 0.66 | 11.85 | 49.14 | 3.18 | 92.22 |
| **+ P2-Train** | **81.69** | **0.69** | **11.05** | **54.08** | **3.25** | **91.25** |

by incorporating specific parameterizations. In Figure S2, we instantiate these sampling algorithms and evaluate their performance on protein sequence generation, focusing on foldability (additional metric results are provided in Figure S5).

Vanilla and greedy ancestral sampling employ a stochasticity strength of 0, effectively disabling remasking, which results in poor performance. DFM sampling introduces tunable stochasticity, leading to improved performance over ancestral sampling; however, it lacks trajectory planning, which limits its effectiveness. RDM sampling, by contrast, enables remasking with a default stochasticity strength of 1 and utilizes the denoiser's confidence for self-planning, yielding better sampling quality.

P2 combines the advantages of these existing algorithms, offering both controllable stochasticity strength and planning guidance. By tuning stochasticity strength, P2 can enhance RDM sampling and optionally leverage an external BERT planner to further steer the sampling trajectory toward generating high-quality sequences.

Table S5: Ablation of Sampling Strategies. Path planning (P2) outperforms existing sampling strategies, including DDPD. The arrows indicate whether higher (↑) or lower (↓) values are better.

| Sampling Algorithm | pLDDT (↑) | pTM (↑) | pAE (↓) | Foldability (%) (↑) | Entropy (↑) | Diversity (%) (↑) |
|---|---|---|---|---|---|---|
| Vanilla Ancestral | 44.08 | 0.34 | 20.61 | 2.00 | **4.03** | **93.63** |
| RDM Sampling | 74.67 | 0.71 | 10.33 | 43.00 | 3.85 | 93.12 |
| **P2 + 8M BERT Planner** | **78.24** | **0.74** | **9.11** | **44.50** | 3.80 | 92.77 |
| DDPD + 8M BERT Planner | 46.51 | 0.24 | 23.20 | 0.25 | 0.31 | 51.69 |
| Ancestral | 52.67 | 0.46 | 17.64 | 7.75 | 3.98 | 93.42 |

In this section, we utilize the protein sequence generation task as an ablation benchmark to analyze the implications of our Path Planning (P2) design choices. We experiment with the ESM2 (Lin et al., 2023) family of protein language models, including versions with 8M, 35M, 150M, 650M, and 3B parameters, for variants incorporating a BERT planner. For the denoiser, we train a 150M MDM from scratch, using the same architecture as ESM2-150M and DPLM-150M, for 500k steps with approximately 320k tokens per step. Training details are provided in Appendix F.1.1.

**Results.** Table S5 demonstrates that our P2 approach consistently outperforms existing sampling strategies across all folding metrics, while maintaining strong token entropy and sequence diversity. Notably, results are further enhanced when an external BERT planner is utilized. To provide a comparative perspective, we perform an apple-to-orange evaluation against a planner-based sampling algorithm, DDPD, equipped with the same BERT planner. DDPD is prone to generating low-entropy, repetitive sequences with poor foldability, as it relies exclusively on the planner to dictate both unmasking and remasking. In contrast, P2 separates these responsibilities: remasking is delegated to the BERT planner, while unmasking is guided by the denoiser itself. This decomposition mitigates the planner's bias and leverages the denoiser's planning capabilities effectively.

In Figure S3, we ablate the size of the planner and evaluate foldability under varying temperatures (entropy). Additional metric results are shown in Figure S6. Our findings reveal that an 8M BERT planner is sufficient to guide a 150M MDM, achieving competitive performance relative to its 3B counterpart across a broad range of entropy values. Furthermore, the BERT planner demonstrates superior scalability compared to the self-planning variant, preserving foldability under extreme high and low temperature conditions.

**Self-Planning Analysis.** In our self-planning approach, we leverage the predicted probabilities from unmasked positions to guide unmasking decisions. This raises a key question: Are the predicted probabilities from unmasked tokens meaningful? We conducted an ablation study where we replaced predicted probabilities for unmasked tokens with uniformly random values and performed the experiments on two MDM variants: one trained from scratch and another fine-tuned from a BERT-based model (DPLM-150M (Wang et al., 2024)).

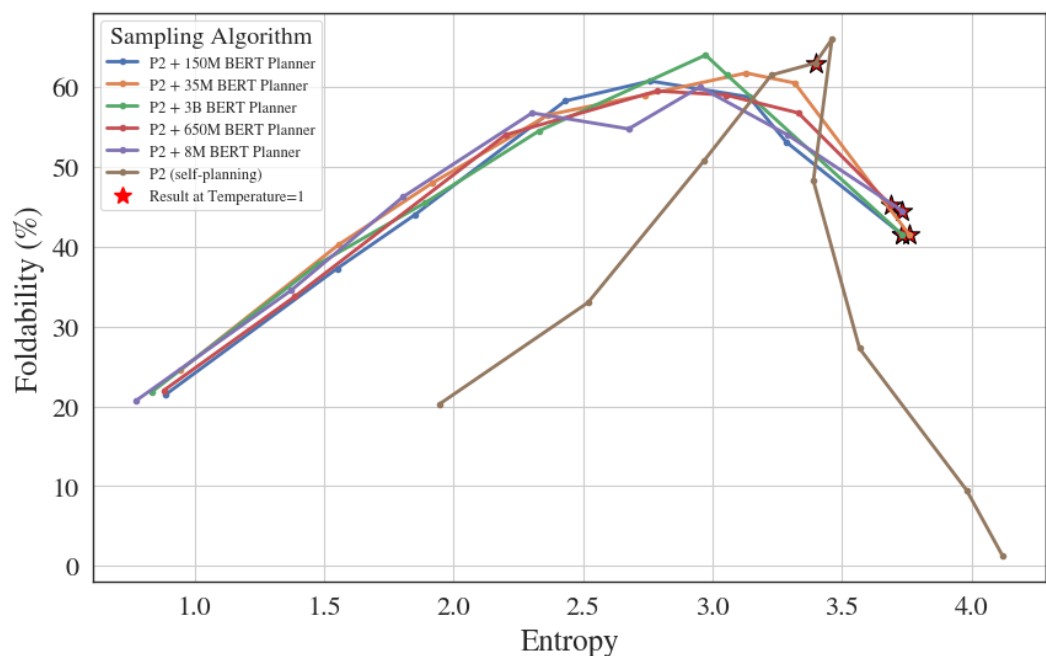

Figure S3: Ablation of the Planner Size: an 8M BERT planner functions similarly to a 3B BERT. Self-planning performs better in a default temperature of 1. We sweep the temperature from 0.1 to 2.0 and plot the scaling between the resultant sequence entropy and the foldability. For more see Figure S6.

Table S6: Comparison of negative ELBOs for Path Planning Planners and self-planning, averaged on 20 runs. Lower values ($\downarrow$) indicate better ELBO. The ELBO is computed at default temperature 1, corresponding to the star-annotation results in Figure S3.

| Method | Unmasked pos.-ELBO ($\downarrow$) | Masked pos.-ELBO ($\downarrow$) |
|---|---|---|
| P2 + Planner ESM2-8M | 22.5 | 13.4 |
| P2 + Planner ESM2-35M | 22.0 | 13.4 |
| P2 + Planner ESM2-150M | 21.8 | 13.4 |
| P2 + Planner ESM2-650M | 21.7 | 13.4 |
| P2 + Planner ESM2-3B | 21.6 | 13.4 |
| P2 (self-planning) | 15.7 | 13.4 |

The DPLM-150M was fine-tuned from ESM2, which was pretrained to predict both masked and randomly mutated tokens, making it more likely to inherit meaningful logits for unmasked positions. As shown in Table S7, randomizing unmasked token probabilities leads to a substantial decline in performance across both variants. This finding confirms that unmasked token logits are informative, despite the lack of direct supervision. It is also evidenced by the ELBO from Proposition 1 in Table S6 where self-planning displays an even better ELBO compared with BERT planners, further validating its effectiveness.

### G.2.4 SAMPLING EFFICIENCY

Increasing the number of sampling steps generally enhances generative quality, albeit with increased computational time. To evaluate the scaling efficiency, we benchmark three sampling algorithms—ancestral sampling, P2 (self-planning), and P2 augmented with an 8M BERT planner—on the task of protein sequence generation. We measure the foldability across increasing sampling steps in terms of elapsed time (benchmarked on NVIDIA A100 GPUs). In Figure S4 top, P2 achieves superior foldability compared to ancestral sampling, while the inclusion of the external BERT planner demonstrates exceptional scalability, particularly at higher sampling steps. In Figure S4 bottom, we further analyze inference efficiency by examining elapsed time and speed (tokens per second) as a function of sequence length. P2 with self-planning maintains the same inference cost as ancestral sampling, as it does not rely on an external model. Conversely, P2 with the BERT planner doubles the number of

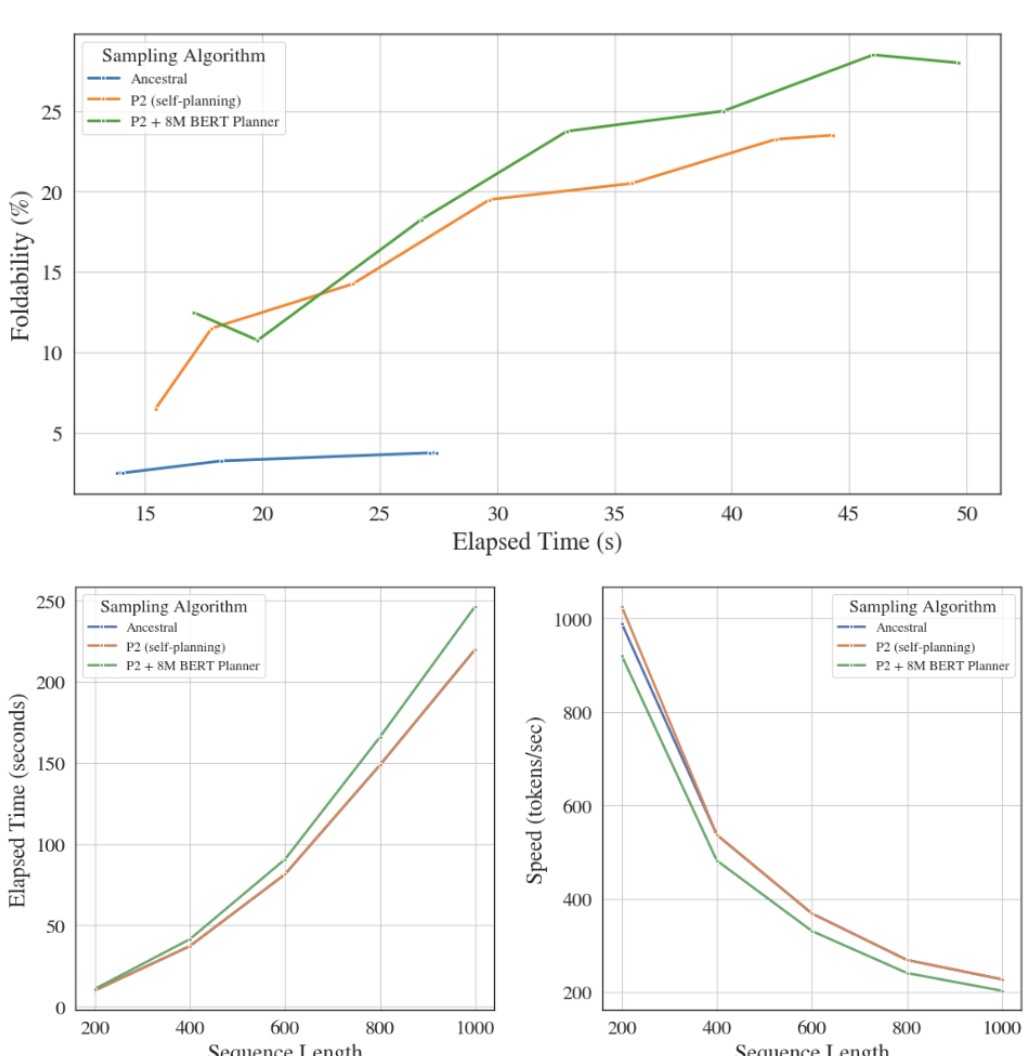

Figure S4: Top: Performance vs. Sampling Time (steps). Bottom: Running Time (left) and Speed (right) vs. Sequence Length.

Table S7: Ablation study of self-planning. We compare self-planning using denoiser-predicted probabilities with a uniformly sampled probability baseline. finetuned MDM refers to MDM finetuned from BERT (DPLM-150M (Wang et al., 2024)), while tfs-MDM refers to MDM trained from scratch.

| Configuration | pLDDT (↑) | pTM (↑) | pAE (↓) | Foldability (↑) | Entropy (↑) | Diversity (↑) |
|---|---|---|---|---|---|---|
| finetuned MDM | 82.62 | 0.72 | 9.15 | 63.00 | 3.40 | 93.05 |
| finetuned MDM + Uniform | 72.61 | 0.66 | 11.82 | 39.00 | 4.01 | 93.62 |
| tfs-MDM | 74.67 | 0.71 | 10.33 | 43.00 | 3.85 | 93.12 |
| tfs-MDM + Uniform | 59.88 | 0.52 | 15.57 | 20.00 | 4.00 | 93.57 |

sampling steps due to one additional BERT evaluation. However, since the planner is a lightweight 8M model compared to the 150M MDM, the overhead is negligible. This is evident in the figure, where the performance gap between P2 (self-planning) and P2 with the 8M BERT planner becomes indistinguishable at higher sampling scales.

### G.2.5 DESIGN SPACE OF P2.

We explore the design space of our proposed P2 framework using key metrics, including pLDDT, pAE, pTM, entropy, and diversity. As illustrated in Figure S5, P2 demonstrates a strong ability to balance structural accuracy and diversity, underscoring its versatility and robustness in protein generation tasks.

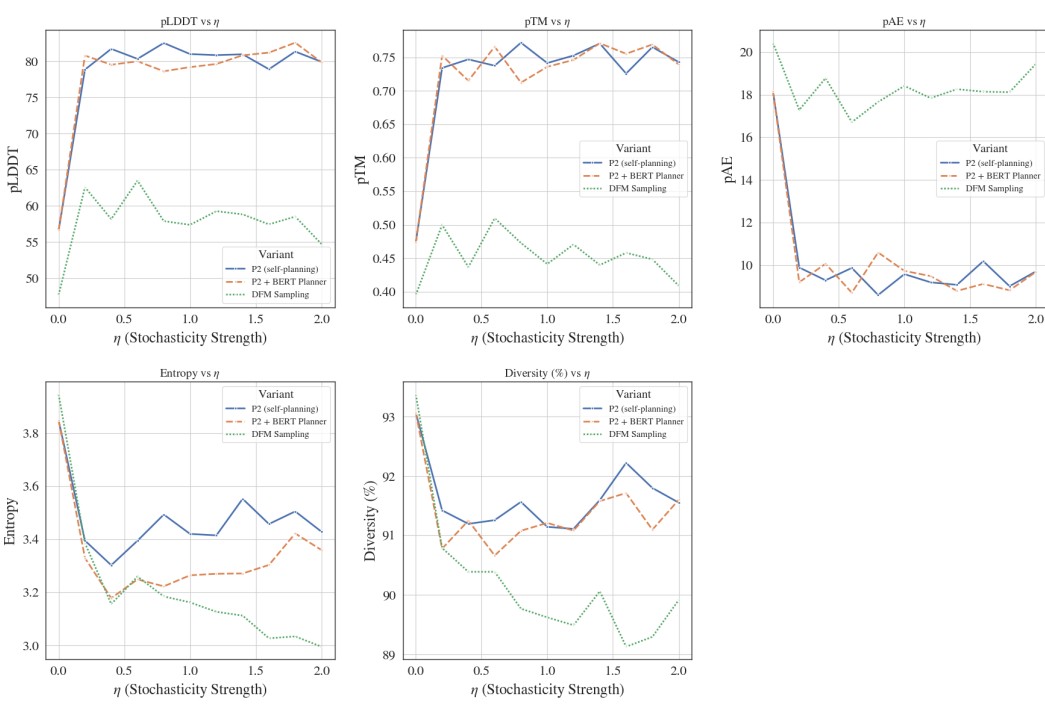

Figure S5: Design space of P2, characterized by pLDDT, pAE, pTM, entropy, and diversity metrics.

### G.2.6 ABLATION STUDY ON THE PLANNER.

We investigate the impact of planner size on model performance through an ablation study. Figure S6 shows how varying the planner size affects key metrics such as pLDDT and diversity. These results emphasize the importance of planner size in optimizing the quality and consistency of generated sequences.

### G.2.7 INFERENCE-TIME SCALING: PERFORMANCE VS. SAMPLING TIME.

To evaluate the trade-off between inference time and performance, we investigate how sampling time scales with model performance. These results will be detailed in future work, but they highlight the scalability of our approach for efficient protein generation.

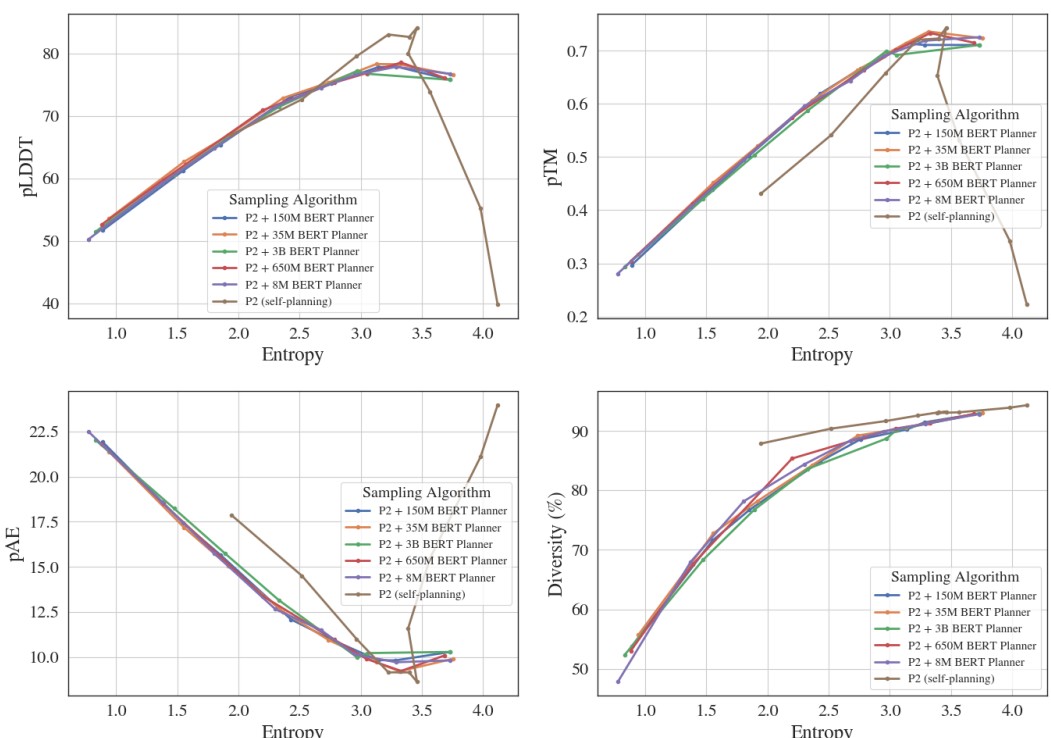

Figure S6: Ablation study of planner size and its impact on protein generation performance.

### G.2.8 PERFORMANCE ON SHORT PROTEIN SEQUENCES (<200 RESIDUES).

While our main results focus on proteins of length 200–800, we also examined performance on shorter proteins. As shown in Table S8, P2 provides substantial improvements even for shorter sequences (64–200 residues). The gains are less pronounced than for longer sequences due to two factors: (1) shorter sequences are underrepresented in the UniRef50 training corpus, limiting model learning; and (2) ESMFold, used for evaluation, is less accurate on shorter sequences. Nevertheless, P2 consistently improves pLDDT and pTM while reducing pAE.

Table S8: Performance on short proteins of different lengths. P2 substantially improves generation quality.

| Length | pLDDT$_{Anc}$ | pLDDT$_{P2}$ | pTM$_{Anc}$ | pTM$_{P2}$ | pAE$_{Anc}$ | pAE$_{P2}$ | Entropy$_{Anc}$ | Entropy$_{P2}$ |
|---|---|---|---|---|---|---|---|---|
| 64 | 49.62 | 70.67 | 0.26 | 0.48 | 16.03 | 10.13 | 3.74 | 2.27 |
| 100 | 43.92 | 70.08 | 0.25 | 0.48 | 18.50 | 11.91 | 3.80 | 2.17 |
| 150 | 46.32 | 71.17 | 0.29 | 0.54 | 19.24 | 11.94 | 3.79 | 2.41 |
| 200 | 56.94 | 80.11 | 0.38 | 0.68 | 17.96 | 9.43 | 3.67 | 2.80 |

### G.2.9 COMPARISON WITH ADDITIONAL BASELINES (ESM2).

We further tested P2 in combination with alternative protein language models to assess generalizability. Although ESM2 is not designed as a generative model, adding P2 yields measurable improvements. For reference, we also include ESM3, ProGen2, and EvoDiff baselines. As shown in Table S9, P2 enhances generation quality across models, with particularly large gains when combined with DPLM.

### G.2.10 COMPARISON WITH TOP-K MARGINAL.

Recent work by Kim et al. (2025) introduced the *Top-K Marginal* method for masked diffusion models. We directly compare Top-K Marginal with our P2 framework in the protein generation setting. As shown in Table S10, P2 substantially outperforms Top-K Marginal, achieving large improvements in all structural quality metrics (pLDDT, pTM, pAE) as well as entropy. These results demonstrate that P2 not only subsumes Top-K Marginal as a special case, but also provides a significant empirical advantage.

Table S9: Comparison with additional protein language model baselines.

| Model | pLDDT↑ | pTM↑ | pAE↓ | Entropy↑ |
|---|---|---|---|---|
| EvoDiff | 31.84 | 0.21 | 24.76 | 4.05 |
| ESM3 | 34.13 | 0.23 | 24.65 | 3.99 |
| ProGen2 | 49.38 | 0.28 | 23.38 | 2.55 |
| DPLM | 80.23 | 0.65 | 12.07 | 3.14 |
| DPLM + P2 | **83.45** | **0.72** | **10.15** | **3.35** |
| ESM2-150M + P2 | 40.99 | 0.16 | 27.08 | 1.51 |

Table S10: Comparison with Top-K Marginal (Kim et al., 2025).

| Model | pLDDT↑ | pTM↑ | pAE↓ | Entropy↑ |
|---|---|---|---|---|
| DPLM | 80.23 | 0.65 | 12.07 | 3.14 |
| DPLM + Top-K Marginal | 53.89 | 0.31 | 22.49 | 2.03 |
| DPLM + P2 (ours) | **83.45** | **0.72** | **10.15** | **3.35** |

### G.2.11 VARIANCE ANALYSIS OF P2.

To assess robustness, we computed variance statistics over 20 independent runs of DPLM+P2. As shown in Table S11, while variance is non-negligible—particularly for pLDDT due to local fluctuations in poorly generated residues—P2 consistently maintains strong mean performance.

Table S11: Variance of DPLM+P2 performance over 20 runs.

| Metric | pLDDT↑ | pTM↑ | pAE↓ | Entropy↑ |
|---|---|---|---|---|
| Mean | 77.39 | 0.62 | 11.62 | 2.91 |
| Std. Dev. | 18.52 | 0.27 | 7.89 | 1.20 |

### G.2.12 GENERATED PROTEIN SEQUENCES AND THEIR PREDICTED STRUCTURES.

We fold the protein sequences generated by our model using ESMFold and visualize their predicted structures in Figures S7–S10. For each length category—200, 300, 400, 500, 600, 700, and 800—we display 15 representative proteins. These visualizations highlight the structural diversity and consistency of the generated sequences, providing evidence of the model's ability to predict biologically plausible structures across diverse lengths.

### G.3 RNA GENERATION

### G.3.1 RNA MDM TRAINING IMPLEMENTATION.

The RNA MDM follows the same discrete diffusion described in (Zheng et al., 2023). The MDM was trained using a machine mounted with 4 A100 GPUs, each with 40GB memory. The training implementation is otherwise identical to the second-stage fine-tuning described in (Wang et al., 2024), where we continued from a RiNALMo (Penić et al., 2024) checkpoint instead of ESM-2 (Lin et al., 2023).

### G.3.2 VISUALIZING THE PREDICTED STRUCTURES OF GENERATED RNA SEQUENCES.

We extend our analysis to RNA sequence generation by folding RNA sequences of 200 base pairs using AlphaFold3 (Abramson et al., 2024). The predicted folding structures, visualized in Figures S11 and S12, highlight the diversity and consistency of the RNA structures generated by the model. Particularly, predicted structures exhibit greater diversity as sequence length increases, as is observed in nature, while their pLDDT's mirroring those computed for natural sequences. We also include the predicted secondary structures of generated RNAs in Figure S13. These results demonstrate the model's ability to generate biologically plausible RNA sequences suitable for downstream applications.

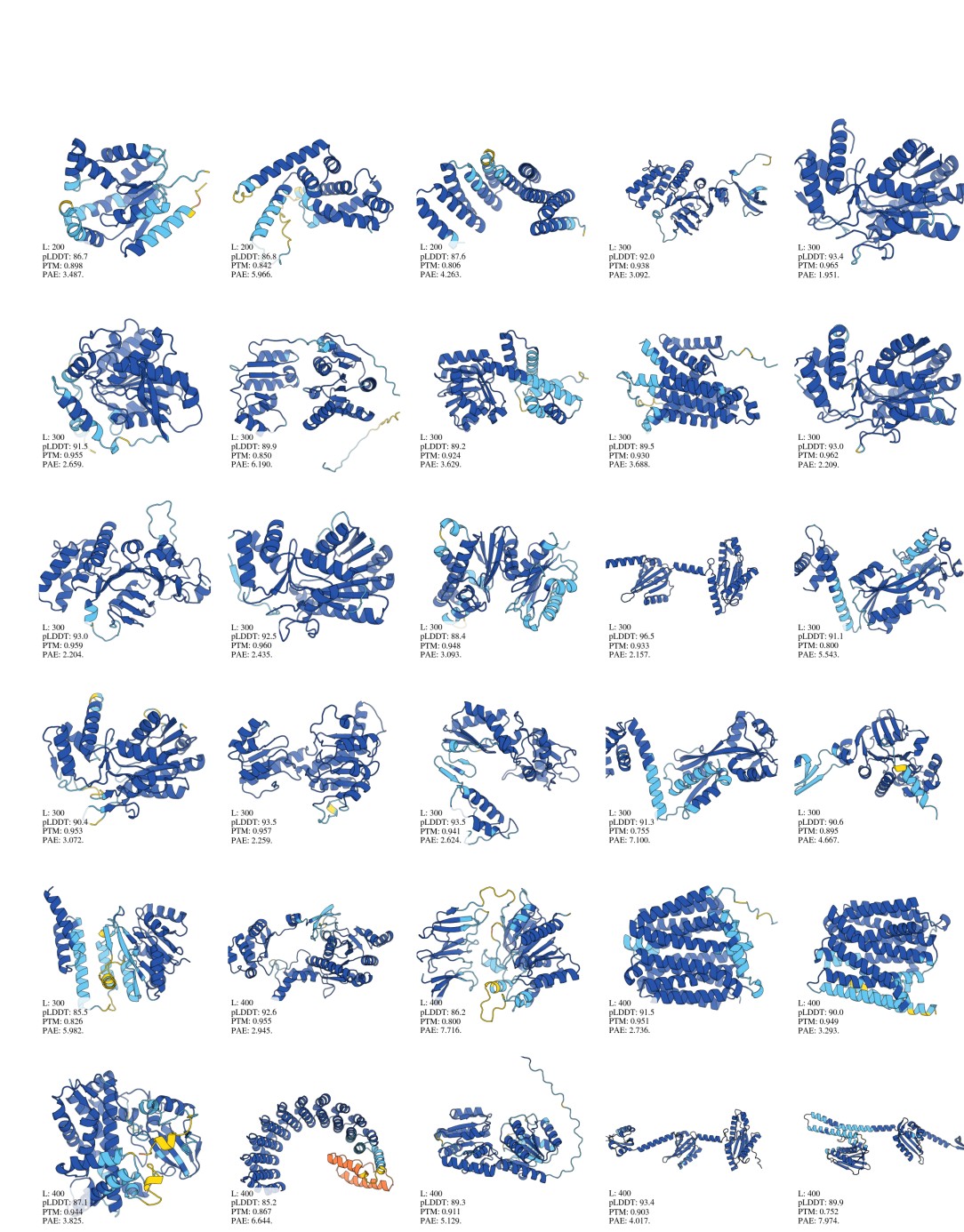

Figure S7: Predicted structures of generated protein sequences (Group 1). Each panel represents structures generated for specific length categories.

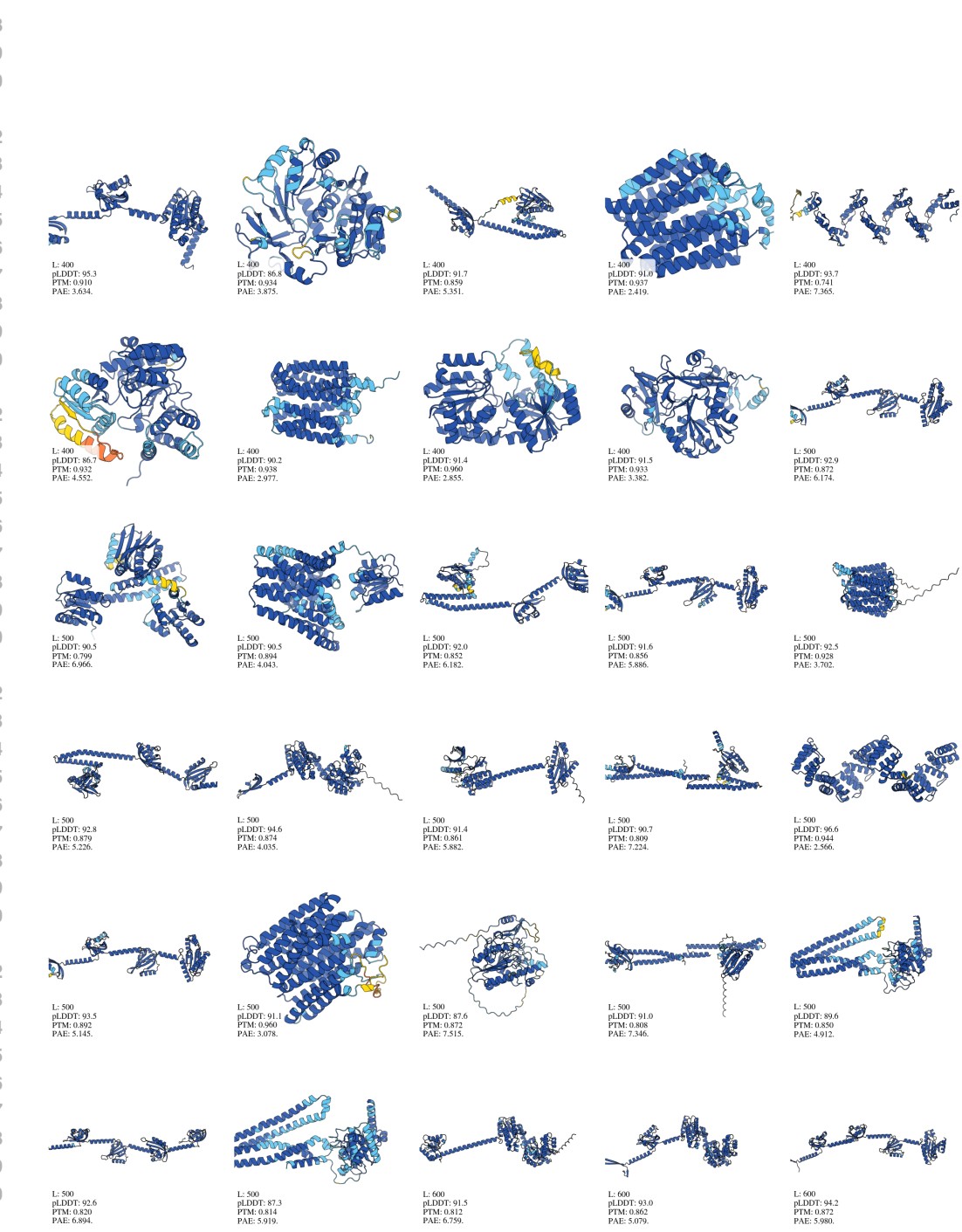

Figure S8: Predicted structures of generated protein sequences (Group 2). Each panel corresponds to different length categories.

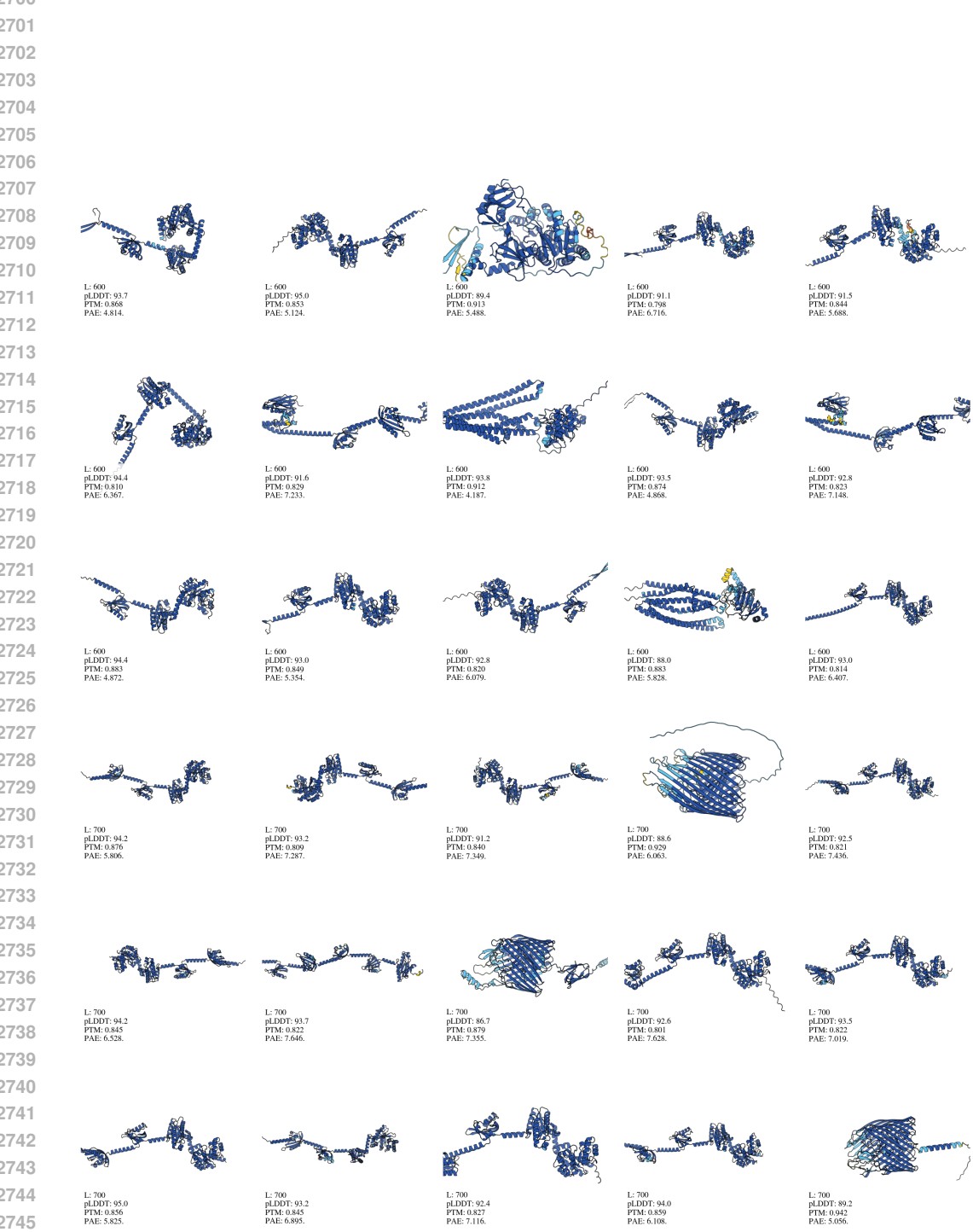

Figure S9: Predicted structures of generated protein sequences (Group 3). These structures illustrate the diversity and robustness of the generation process.

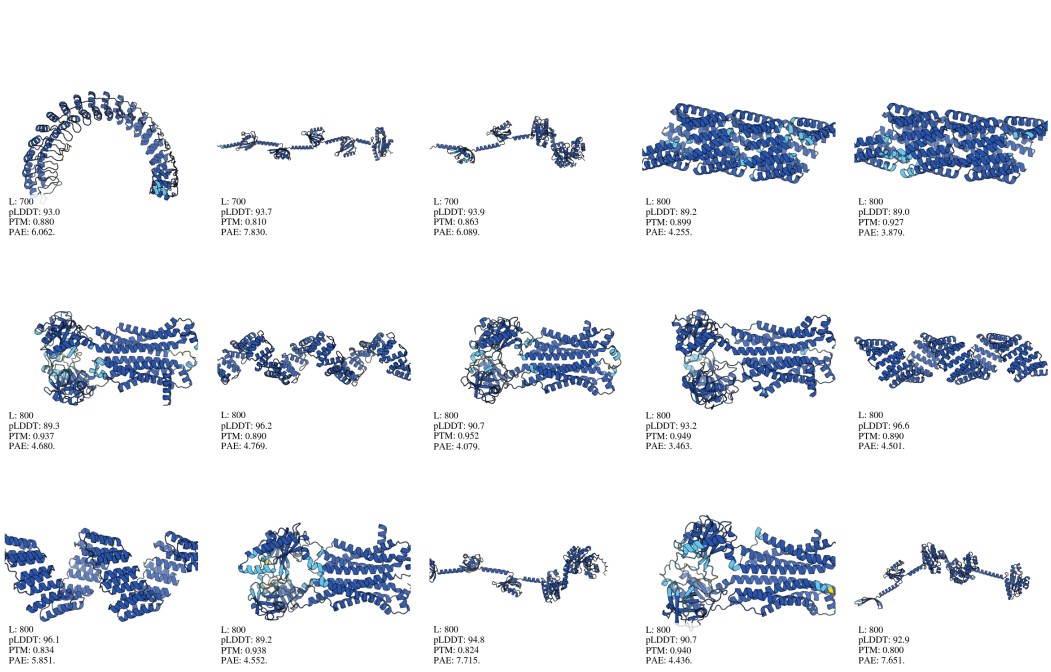

Figure S10: Predicted structures of generated protein sequences (Group 4). This group emphasizes structures for the longest generated sequences.

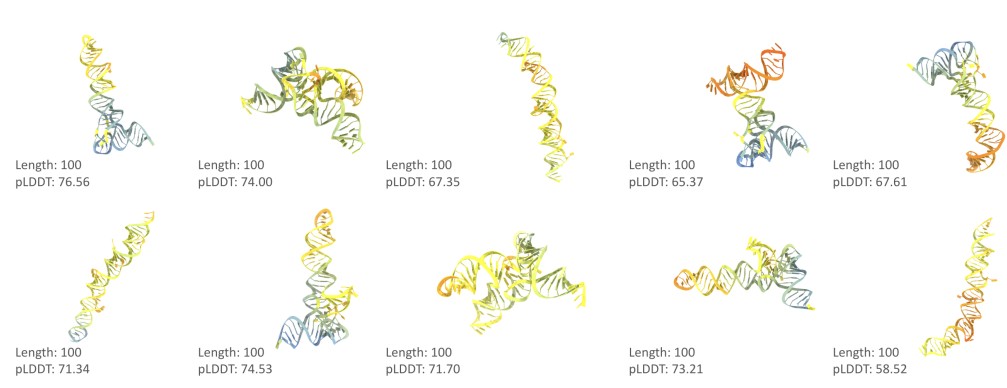

Figure S11: Predicted structures of additional generated RNA sequences (100 bps).

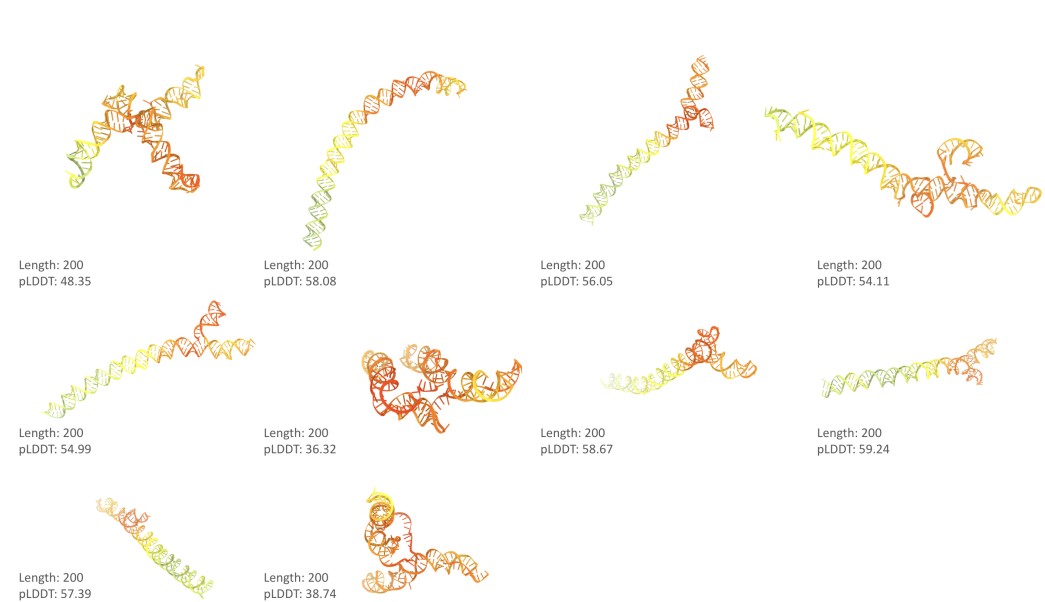

Figure S12: Predicted structures of generated RNA sequences (200 bps). This figure showcases the structural diversity of RNA sequences generated by the model as sequence length increases, which is observed in nature.

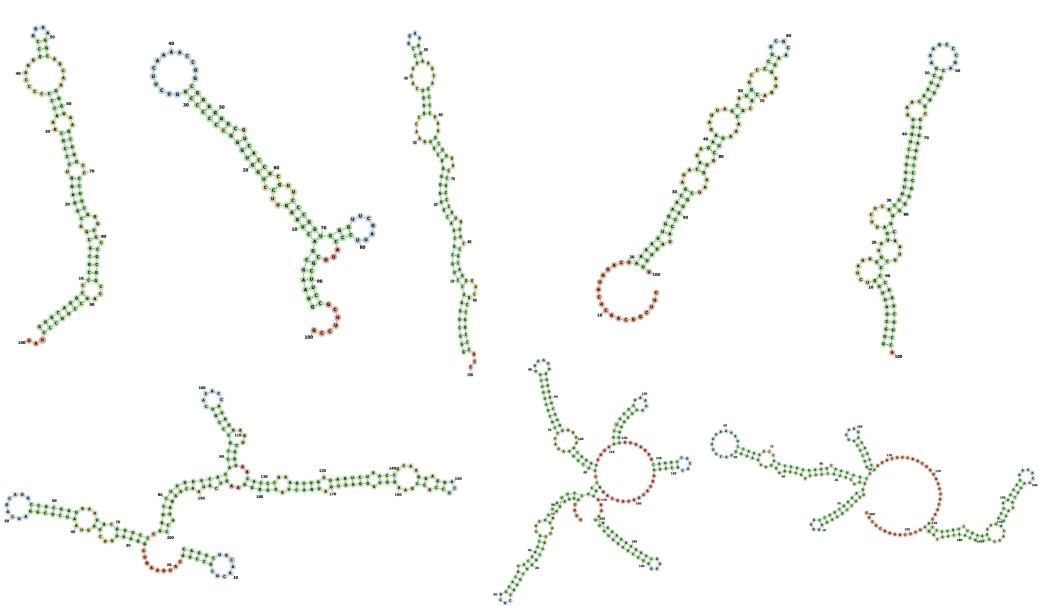

Figure S13: Predicted secondary structures of generated RNA sequences of length 100 (top) and 200 bp (bottom). Predictions were made using ViennaRNA (Lorenz et al., 2011) and visualized with forna (Kerpedjiev et al., 2015).