# OpenReview forum: "Path Planning for Diffusion Language Model Sampling"
_ICLR.cc/2026/Conference — Submitted to ICLR 2026_

### Official Review · Reviewer_UGiP · 2025-10-25

**Soundness:** 3
**Presentation:** 3
**Contribution:** 2
**Rating:** 4
**Confidence:** 4

**Summary:**

This paper proposes a planned denoising algorithm for masked discrete diffusion models, which is an important topic of study in this area. Here, instead of choosing to denoise the masked tokens at random, and keeping them that way till the end, the planner model chooses to unmask the tokens with most confidence and re-mask possibly erroneous tokens (a feature found in uniform diffusion but not in masked diffusion).

Three variants are introduced for training such a planner, and they are shown to improve the generation quality for protein folding, RNA sequence generation and various language/math/coding tasks. The authors differentiate the current work from Kim et al, which uses heuristic planning algorithms for masked Diffusion models, and Liu et al which works mainly with uniform diffusion instead of masked diffusion.

**Strengths:**

This work considers the well motivated problem of planned denoising in discrete diffusion models. Better inference time algorithms can significantly improve the efficiency and accuracy of diffusion models, as amply demonstrated in the literature. The algorithm is simple and shows strong gains across multiple datasets. This work demonstrates the utility of a trained planner for masked diffusion models, and shows that it can lead to better results than inference time heuristics such as topK-marginal sampling.

**Weaknesses:**

- Most of the ideas in the paper seem straightforward and introduced in some form in prior works. I therefore am not very positive about the novelty of this work. (Kim et al, Liu et al, Campbell et al etc)

**Missing Baselines**

- In the space of planning and constrained generation, the authors need to compare to the remasking strategy used in Algorithm 2 in https://arxiv.org/pdf/2410.14157

- Kim et al seems like an important baseline to this work, and is easily implementable on any MDM since it is an inference time heuristic. The authors only compare to this work in the protein folding task. The evaluations seem incomplete without comparison in the language modeling and the RNA sequence generation tasks as well.

- The DDPD algorithm of Liu et al can be modified to work with pretrained masked diffusion models, as clearly acknolwedged in this work. Since the authors themselves claim that this is the closest work, I am surprised that there is only a single comparison with this method, that too only in the appendix. I do understand that this requires extensive training, comparable to the current work. I believe that such a comparison will make the current work's case much stronger.


**Minor points:**

"Specifically, solely using P2 inference, we observe relative improvements of in 22 protein sequence foldability,  8 in RNA sequence pLDDT, 4 in math reasoning,  68 in story generation (ROUGE score)...."

Usually absolute improvements are reported in this literature. I request the authors to change this to absolute numbers. For instance,  in table 2, the foldability improvements seems to be 10% absolute instead of 22% relative.

**Questions:**

See weaknesses section.

- Does the change in the path measure given by the stochasticity parameter $\eta$ change the final marginal distribution of the samples? Is there any reference to a theoretical analysis with arbitrary $\eta$?

---

> ### Author Response · Authors · 2025-11-22
> **Response to Reviewer UGiP Part 1**
>
> We thank the reviewer for their time and effort in reviewing our manuscript. We appreciate that the reviewer agrees that our algorithm is “simple and shows strong gains across multiple datasets” and leads to “better results than inference time heuristics such as topK-marginal sampling.” We now answer the key clarification points raised by the reviewer, grouped by theme. Finally, we wish to highlight that the changes in the updated PDF are presented in red to aid transparency.
>
> ## Novelty
>
> We appreciate the reviewer's concern, given the existing literature on similar unmasking/remasking schemes. We acknowledge that the introduction of planning strategies and remasking in MDMs in itself is not new, and, we sought to bring attention to how various other discrete diffusion inference schemes that can be related to each other through Table 1 under our proposed P2 framework. This unification, while illuminating in connecting prior work, does not in itself bring strong insights.
>
> Consequently, the core contribution of our proposed P2 framework is our theoretical contribution, which provides a more rigorous and theoretical treatment of denoising orders under planning—which includes remasking—which allows us to derive an ELBO. We highlight that such an ELBO is a contribution of this work and is absent from existing papers in the literature, aside from DDPD, which, as the reviewer notes, works mainly with uniform diffusion instead of masked diffusion. The ELBO provides an avenue to understand and evaluate through which aspects of the planner during inference can be modified to improve empirical performance. Using this insight, we demonstrate in Table S6 of the appendix, which compares the ELBO of self-planning and various BERT planners, which we can see is correlated with the planning strategies performance as per Figure S3. This also leads to the ability to train an external model as a planner, which is a major contribution of this work that we have expanded greatly upon in the revised version of the manuscript.
>
> While the idea of planning for different denoising orders is not new, our theoretical framework and ability to use and train lightweight external planners (as with P2-BERT and P2-train) certainly is.
>
> ## Additional Baselines
>
> > “In the space of planning and constrained generation, the authors need to compare to the remasking strategy used in Algorithm 2 in https://arxiv.org/pdf/2410.14157”
>
> The only sampling strategies which we were able to find employed in the linked paper [1] are “Random” which we call “Ancestral” and “TopK” which we call “Greedy Ancestral.” If there is a sampling strategy explored in this work which incorporates remasking that we have missed, we ask the reviewer to kindly refer us to which methodology they would like to see compared against those developed in our manuscript.
>
>
> > Additional comparison with Kim et. al.
>
> We agree with the reviewer’s assessment that Top-K Marginal (Kim et.al.) is an important baseline which should be compared against P2 in multiple domains.
>
> We now include Table S3, where we compare various sampling methods across a wide array of coding benchmarks. P2-self is seen to outperform all planning strategies, including TopK-marginal on this task.
> | Method                           | HumanEval Pass@1 | HumanEval Pass@10 | HumanEval+ Pass@1 | HumanEval+ Pass@10 | MBPP Pass@1 | MBPP Pass@10 | MBPP+ Pass@1 | MBPP+ Pass@10 | HumanEval Infill Pass@1 | SantaCoder Pass@1 |
> |----------------------------------|------------------:|-------------------:|-------------------:|--------------------:|-------------:|--------------:|--------------:|---------------:|--------------------------:|-------------------:|
> | **P2-self-plan** | 20.8 | 38.4 | 17.6 | 35.2 | 16.7 | 38.4 | 23.9 | 53.6 | 77.4 | 56.4 |
> | Ancestral                          | 3.3  | 18.3 | 3.2  | 15.2 | 1.8  | 13.2 | 2.9  | 21.8 | 72.7 | 53.8 |
> | Greedy Ancestral (Chang et al., 2022) | 9.3  | 31.1 | 8.1  | 28.7 | 5.3  | 29.0 | 8.7  | 41.5 | 75.1 | 53.7 |
> | Entropy-based Confidence                         | 12.6 | 35.4 | 10.9 | 29.9 | 9.2  | 36.8 | 15.2 | 50.7 | 75.1 | 53.2 |
> | topk-margin (Kim et al., 2025)   | 7.6  | 27.4 | 6.5  | 26.2 | 3.9  | 24.0 | 6.2  | 33.5 | 75.0 | 54.4 |

---

> ### Author Response · Authors · 2025-11-22
> **Response to Reviewer UGiP Part 2**
>
> > Additional Comparison with DDPD
>
> We agree that DDPD is the closest to our work from a theoretical standpoint in that they also prove a rigorous ELBO. However, as outlined in Appendix D.4, due to their perspective of using uniform diffusion models as the base of their generation process, and as the reviewer notes, they require extremely large planners to obtain gains over even Ancestral sampling methods. The perspective of P2 is to allow the user to employ or train an extremely lightweight external planner, which is fundamentally incompatible with DDPD. In our single comparison with DDPD (Table S5), we show this lack of performance, and discuss that this is an “apple-to-orange evaluation.” For this reason, along with the massive computational requirements required to train a planner of DDPDs’ size for “fair” comparison, we opt not to include additional comparisons in the manuscript.
>
> ## Theoretical Clarifications
>
> > Does the change in the path measure given by the stochasticity parameter change the final marginal distribution of the samples? Is there any reference to a theoretical analysis with arbitrary?
>
> Thank you for the question. Indeed, it does, and in principle one could train a planner with a fixed value of $\eta$ in mind. This simply corresponds to inserting the reweighted masked planner and unmasked planner, respectively, into the ELBO of Proposition 1. We now detail how to make this insertion in the new Appendix B.3, and further observe what the role of $\eta$ in terms of reweighting the loss is: As $\eta$ increases, the gradients from the “Masked Planner” ELBO term $\mathcal{E}\_{MP}$ will increase, and those from the “Unmasked Planner” ELBO term $\mathcal{E}\_{UP}$ will decrease. This agrees with our intuition - increasing $\eta$ inflates logits from the masked planner relative to those of the unmasked planner, making it that more selections due to the unmasked planner will be those which are updated at each generation step.
>
> ## Concluding remarks
>
> We thank the reviewer again for their time and effort during this rebuttal process. We hope our answers here have aided in clarifying all the great questions raised by the reviewer. If there are any further questions held by the reviewer, we remain eager to engage further. If the reviewer feels satisfied with our responses and new rebuttal results, we would be thrilled if the reviewer would consider a renewed evaluation of our work with a potential score upgrade.
>
> ## References
>
> [1] Ye, Jiacheng et al. “Beyond Autoregression: Discrete Diffusion for Complex Reasoning and Planning.”  arXiv preprint arXiv:2410.14157 (2025)

---

### Official Review · Reviewer_fkTi · 2025-10-26

**Soundness:** 4
**Presentation:** 3
**Contribution:** 3
**Rating:** 6
**Confidence:** 4

**Summary:**

The crucial bottleneck in existing MDM at inference is that they can not recorrect what tokens have been already unmasked, resulting in error propagation and suboptimal generation quality. The paper introduce a path planner to decide which token to unmask and which token to remask, allowing to refine exising unmasked tokens. The authors provide a theoritical support on the evidence lower bound of the framework and empirally instantiate planer from various source: self-planing (model itself), BERT-planning model, and trained-planning model. The method demonstrates SOTA performance on conducted benchmarks, such as biological sequence, math reasoning, and code generation.

**Strengths:**

1. The method is technically sound and indeed novel since it equipping the diffusion model a error-correction capacity in a clean and disentangled way by a denoiser and a planner.
2. The method is well motivated with a easy-to-read presentation.
3. The quanitative results are strongly compelling.

**Weaknesses:**

1. The name of planner is a bit confusing: mask planner denotes denoiser, unmask planner refers to the "actual" planner.
2. The rationale of using BERT as planner remains ambiguous. It would be better to provide insight on why BERT predicted probability is reliable indicator.
3. Training details of trained planner is missed.

Exposition:
- Related work: Paragraph should start with a capital word.

**Questions:**

Q1. Since discrete diffusion itself can be used a planner, what happen if we use a large discrete diffusion to server as a planner. Expect the performance will be better?

Q2. In Table 3, what type of planner is used? I assume this is self-planner.

Q3. Do the author have a table for comparing different sampling strategies on math and coding performance? Similar to what have done in Table 5 (protein generation) and Table 6 (infilling).

---

> ### Author Response · Authors · 2025-11-22
> **Response to Reviewer fkTi Part 1**
>
> We thank the reviewer for their time and effort in reviewing our manuscript. We appreciate that the reviewer agrees that our paper is “technically sound and indeed novel” and introduces error correction to MDMs in a “in a clean and disentangled way” which is “well-motivated with a easy-to-read presentation.” We are also glad the reviewer agrees that our quantitative results are “strongly compelling.” We now answer the key clarification points raised by the reviewer, grouped by theme. Finally, we wish to highlight that the changes in the updated PDF are presented in red to aid transparency.
>
> ## Different instantiations of planners
>
> > “The name of planner is a bit confusing: mask planner denotes denoiser, unmask planner refers to the "actual" planner.”
>
> In practice, indeed, we use the denoiser itself as the masked planner in P2-self, P2-BERT, and P2-train. However, these are only specific instantiations of the general framework we propose, and one need not make this choice of masked planner. For example, DDPD [2] makes use of an external planner in masked positions - see Table 1.
>
> >”The rationale of using BERT as planner remains ambiguous. It would be better to provide insight on why BERT predicted probability is reliable indicator.”
>
> We modified our discussion of BERT planning slightly to clarify that the ability for BERT to be used as an “unmasked” planner is corroborated by previous empirical studies [1]. The essential idea is that, as BERT is trained on both randomly masked and randomly flipped token positions, it has learned to evaluate whether an “unmasked” token is representative of the data distribution.
>
> > “Since discrete diffusion itself can be used a planner, what happen if we use a large discrete diffusion to server as a planner. Expect the performance will be better?”
>
> This is an insightful question, and precisely the perspective taken by the previous work DDPD [2], who make use of a large uniform diffusion model as their planner in both masked and unmasked positions. The use of this large model introduces significant computational overhead, and requires training a uniform diffusion model of comparable or larger size than the denoiser to see gains over vanilla ancestral sampling. A key insight of our work was to view the planning problem strictly from the perspective of masked diffusion models, which allowed us to make the separation into masked and unmasked planners and introduce the possibility of using a very lightweight planner for significant performance gains. We include a detailed discussion of comparisons with DDPD in Appendix D.4, and exhibit its poor performance using planners of the size used for P2-BERT and P2-Train in Table S4.
>
> > “In Table 3, what type of planner is used? I assume this is self-planner.”
>
> Indeed, it is P2-self being used in table 3. Thank you for pointing out that we should make this clarification. We revised tables 2,3, and 4 to all indicate which versions of P2 are being used for the respective tasks.
>
> > “Training details of trained planner is missed.”
>
> Thank you, we believe the training of a planner is an important aspect of our work, and have elaborated much more on the details of the trained planner.
>
> We now include an extended discussion of P2 train in Section 3.3, where it is explained that we fine-tune BERT for use as $G_U$ and use the denoiser itself as $G_M$ in P2-train.We also emphasize that, although we only use the fine-tuned BERT model as $G_U$ for sampling, it is trained on both masked and unmasked positions in Alg.2.
>
> We additionally include in section 3.3 the new Proposition 2, where we prove the form of the minimizers in response to another reviewer’s concern regarding providing a theoretical guarantee, and is stated in terms of a backbone model $T$ that is used as both the masked planner and unmasked planner in training.
>
> We provide specifics on the training details for this fine-tuned BERT in Appendix F.1.2.

---

> > ### Author Response · Authors · 2025-11-22
> > **Response to Reviewer fkTi Part 2**
> >
> > ## Additional experiments on math and coding
> >
> > We thank the reviewer for this suggestion, and agree that these are important domains in the realm of MDMs for which new sampling strategies should be tested. We previously included Table 6, for which the first column (pass@1) is a code generation benchmark.
> >
> > We now include Table S3, where we compare various sampling methods across a wide array of coding benchmarks. P2-self is seen to outperform all planning strategies, including TopK-marginal on this task.
> > | Method                           | HumanEval Pass@1 | HumanEval Pass@10 | HumanEval+ Pass@1 | HumanEval+ Pass@10 | MBPP Pass@1 | MBPP Pass@10 | MBPP+ Pass@1 | MBPP+ Pass@10 | HumanEval Infill Pass@1 | SantaCoder Pass@1 |
> > |----------------------------------|------------------:|-------------------:|-------------------:|--------------------:|-------------:|--------------:|--------------:|---------------:|--------------------------:|-------------------:|
> > | **P2-self-plan** | 20.8 | 38.4 | 17.6 | 35.2 | 16.7 | 38.4 | 23.9 | 53.6 | 77.4 | 56.4 |
> > | Ancestral                          | 3.3  | 18.3 | 3.2  | 15.2 | 1.8  | 13.2 | 2.9  | 21.8 | 72.7 | 53.8 |
> > | Greedy Ancestral (Chang et al., 2022) | 9.3  | 31.1 | 8.1  | 28.7 | 5.3  | 29.0 | 8.7  | 41.5 | 75.1 | 53.7 |
> > | Entropy-based Confidence                         | 12.6 | 35.4 | 10.9 | 29.9 | 9.2  | 36.8 | 15.2 | 50.7 | 75.1 | 53.2 |
> > | topk-margin (Kim et al., 2025)   | 7.6  | 27.4 | 6.5  | 26.2 | 3.9  | 24.0 | 6.2  | 33.5 | 75.0 | 54.4 |
> >
> > ## Concluding remarks
> >
> > We thank the reviewer again for their time and effort during this rebuttal process. We hope our answers here have aided in clarifying all the great questions raised by the reviewer. If there are any further questions held by the reviewer, we remain eager to engage further. If the reviewer feels satisfied with our responses and new rebuttal results, we would be thrilled if the reviewer would consider a fresher evaluation of our work with a potential score upgrade.
> >
> > ## References
> >
> > [1] Hie, Brian L. et al. “Efficient evolution of human antibodies from general protein language models and sequence information alone.” Nature Biotechnology, 2024
> >
> > [2] Liu, Sulin et al. “Think while you generate: Discrete diffusion with planned denoising.” International
> > Conference on Learning Representations, 2024

---

### Official Review · Reviewer_aNk1 · 2025-10-31

**Soundness:** 2
**Presentation:** 1
**Contribution:** 2
**Rating:** 2
**Confidence:** 5

**Summary:**

This work formalizes the problem of selecting unmasking and remasking positions of MDMs. The authors provide extensive experimental results to support their method's empirical usefulness.

**Strengths:**

The authors put effort into formalizing the unmasking and remasking predictors with ELBO, and provide experimental results on various domains.

**Weaknesses:**

I believe there are a few crucial flaws and limitations in the author's theoretical claim. (I'd be happy to be further clarified by the authors on these)

- **Claim on remasking**: Beyond P2-Train, in P2-Self and P2-Bert, the authors incorporate the training-free remasking strategy into the unmasking schedule, which they argue as an unification. However, the information that they use for remasking has never been learned. For example, in P2-Self, the authors use the pretrained MDM's logits to select which positions to remask, but a model has never gotten the training signal on clean token positions, so it isn't that much different from randomly remasking clean tokens, as done in (Wang et al, 2025)
. Given that investigating the unmasking order using logits has also been previously studied, e..g, Kim et al, MaskGIT, just incorporating the remasking scheme into unmasking shouldn't be the core contribution.

- **Explanation on P2-train**: Indeed, P2-train is the trainable approach to get an unmasking and remasking planner, in contrast to prior work that focuses on a training-free approach. Given that P2-Self and P2-Bert don't differ that much from the prior approach (as I claimed above), the authors should've dealt more carefully with P2-Train. Below are the reasons why I don't think the author's account on P2-Train is inappropriate.

- *Training algorithm*: In Algorithm 2, the authors argue that they use the indicator function $\mathbb{1}[z^i = x_0^j]$ as a label. But given the sampling process of D_t, xt, and z should have the same clean token at the clean positions. Doesn't it mean that this label is always one for the clean token position? If we follow this algorithm, $G_U(z,x_t)$ always outputs one for the clean token position, which is apparently what we don't want, since the training signal given to $G_U$ is always one at clean token positions. It is even unclear in the code whether we use the same backbone to model $G_M$ and $G_U$. Also, the ELBO loss in Proposition 1 tells that $G_M$ should $z^{-i}$ takes an input. I was wondering how the authors addressed this input discrepancy. This is because it seems tricky to deal with since it takes $z$ as an input, but behaves like it doesn't take $z^i$, even in the recent work (Zhao et al, 2024) uses an entirely new architecture to address this issue.

- *No detail on planner design*. I failed to find the training details of the planner, which I think is the most important part of the paper. To clarify, I'm not referring to the specific implementation detail but the basic training configuration of how to train $G_U$ and $G_M$.

- *No theoretical guarantee*. The MDM training objective itself has the theoretical guarantee (if the loss is exactly minimized, it recovers the per-token posterior). But the authors don't provide that guarantee; it's even unclear what the planners are aiming towards. (Note that the ELBO loss cannot explain this).

- **Paper writing**: Last but not least, the paper is extremely hard to follow (at least for me). Although the idea (adding additional modules to learn the unmasking and remasking positions selection) is really simple, the authors formulated it in an unreasonably difficult way, further decreasing the readability. I personally had a really hard time digesting the author's proposed formulation and realized that it can be even more simplified.

**Questions:**

Please refer to the Weakness section!

---

> ### Author Response · Authors · 2025-11-22
> **Response to Reviewer aNk1 Part 1**
>
> We thank the reviewer for their time and effort in reviewing our manuscript. We appreciate that the reviewer agrees that our paper “formalizes” remasking and unmasking with an ELBO, and the proposed P2 sampler is evaluated on various domains. We now answer the key clarification points raised by the reviewer, grouped by theme. Finally, we wish to highlight that the changes in the updated PDF are presented in red to aid transparency.
>
> ## Claims on remasking
>
> We appreciate the reviewer's concern and feedback that the claims on incorporating the remasking scheme into unmasking should not be a core contribution, given the existing literature on similar unmasking/remasking schemes. We agree with the reviewer, and we also do not consider this a core contribution of our work. In particular, we sought to bring attention to how various other discrete diffusion inference schemes that can be related to each other through Table 1 under our proposed P2 framework. This unification, while illuminating in connecting prior work, does not in itself bring strong insights.
>
> Consequently, the core contribution of our proposed P2 framework is our theoretical analysis, which provides a more rigorous and theoretical treatment of denoising orders under planning—which includes remasking—which allows us to derive an ELBO. We highlight that such an ELBO is a contribution of this work and is absent from the paper that the reviewer highlights in MaskGiT (Kim et. al). While ReMDM (Wang et al, 2025) includes an ELBO, it does not have a planner, and remasks uniformly at random according to a fixed time schedule, equivalently to DFM Sampling, which massively underperforms compared to planned strategies (see Tables 5,6,and 7). Furthermore, this ELBO provides an avenue to understand and evaluate through which aspects of the planner during inference can be modified to improve empirical performance. Using this insight, we demonstrate in Table S6 of the appendix, which compares the ELBO of self-planning and various BERT planners, which we can see is correlated with the planning strategies performance as per Figure S3. As the reviewer notes, this also leads to the ability to train an external model as a planner, which we agree is a major contribution of this work.
>
> ## Explanation on P2 Train
>
> ### Clarification on the Training algorithm
>
> > “Doesn't it mean that this label is always one for the clean token position?... It is even unclear in the code whether we use the same backbone to model G_M and G_U.”:
>
> The reviewer makes an astute observation in that if the unmasked position $x_t$ the label is always $1$. Consequently, to use both labels $0$ and $1$ it is necessary to train a planner for both on unmasked states as $G_U$ and masked states as $G_M$ as outlined in algorithm 2. We wish to highlight that this is precisely how we train P2-train in our experiments. We understand that this aspect was not sufficiently clear in the original manuscript, and we have now updated the PDF to increase clarity of this aspect. We provide an excerpt of the proposed change here as well for the reviewer’s convenience:
>
> “We can also employ a trained planner that operates on the denoiser’s prediction and the current masked input. Specifically, we freeze the denoiser during training and fine-tune the BERT planner by taking $G^j_M(z,x)=G^j_U(z,x)=\text{Cat}(z^j;B_{\phi}^{j}(z))$, using a cross-entropy loss derived from the planner ELBO objective… During sampling for experiments using P2 Train - see Table 5 - we use the same parameterization as with P2 BERT for constructing $G_{\eta}^j$ of Equation (7) in Alg. 1. That is, we set the unmask planner to be the fine-tuned BERT $G_U^j(z,x)=\text{Cat}(z^j;B_{\phi}^{j}(z))$ and mask planner to be the denoiser $G^j_M(z,x)=\text{Cat}(z^j;D^{j}_\theta(x))$.
>
> We emphasize that, although we only use the fine-tuned BERT model as $G_U$ for sampling, it is trained on both masked and unmasked positions in Alg.2. This allows for the model to have a meaningful training signal in that it gets to see both $0$ and $1$ as the label - without training on masked positions this would not be the case, since the label is always $1$ in unmasked positions. The training of the planner in Alg.2 is theoretically supported by the following propositions. Note that indeed in Proposition 2 we make the assumption that the same network backbone is used as both $G_M$ and $G_U$ in training, and this should always be done in practice, even if one intends to use $T$ only as $G_U$ or $G_M$ during sampling.’’
>
> Proposition 2 here is a new result in which we prove the form of the minimizers in response to the reviewers concerns regarding providing a theoretical guarantee, and is stated in terms of a backbone model $T$ that is used as both the masked planner and unmasked planner in training.

---

> > ### Author Response · Authors · 2025-11-22
> > **Response to Reviewer aNk1 Part 2**
> >
> > > “Also, the ELBO loss in Proposition 1 tells that  G_M should  takes z^{-i} an input. I was wondering how the authors addressed this input discrepancy.”
> >
> > We have now rewritten this term in the ELBO to match the loss which we actually use in practice, which loosens the bound but makes the connection between the training loss for the planner and the ELBO more interpretable.
> >
> > Namely, in $\mathcal{E}_{\text{UP}}(x_0)$, we previously had written:
> >
> > $$\int_0^1 -\beta_t \mathbb{E}_{x_t}[\sum\_{i=1,x^i_t\neq m}^L \mathbb{E}\_{z\sim D\_\theta(x_t)}[\log G^i\_U(z,x_t)]]dt$$
> >
> > and now write:
> >
> > $$\int_0^1 -\beta_t \mathbb{E}_{x_t}[\sum\_{i=1,x^i_t\neq m}^L \mathbb{E}\_{z\sim D\_\theta(x_t)}[CE(Cat(z^i;\delta(x^i_0), G^i_U(z,x_t))]]dt,$$.
> >
> > where $\text{CE}(a,b)=a\log b + (1-a)\log(1-b)$. As the reviewer remarks, when $x^i_t\neq m,\text{Cat}(z^i;\delta(x^i_0)=1$, so these expressions are equivalent.
> >
> > Additionally $\mathcal{E}_{\text{UM}}(x_0)$, we previously had written:
> >
> > $$\int_0^1 -\beta_t \mathbb{E}_{x_t}[\sum\_{i=1,x^i_t=m}^L \mathbb{E}\_{z\sim D\_\theta(x_t)}[\log G^i\_M(z^{-i},x_t)]]dt$$
> >
> > And bounded this below by
> >
> > $$\int_0^1 -\beta_t \mathbb{E}_{x_t}[\sum\_{i=1,x^i_t = m}^L \mathbb{E}\_{z\sim D\_\theta(x_t)}[CE(Cat(z^i;\delta(x^i_0), G^i_M(z,x_t))]]dt,$$
> >
> > which we had denoted $\hat E_M(x_0)$ in order to formulate the loss. Using $\hat E_{\text{M}}(x_0)$ in the place of $\mathcal{E}_{\text{M}}(x_0)$ still yields a (weaker) ELBO, since recalling that $z^{-i}$ was denoting replacing the ith token with $x^i_0$ and that $\beta$ is negative, we use:
> >
> > $$ \sum\_{i=1,x^i_t=m}^L \mathbb{E}\_{z\sim D\_\theta(x_t)}\left[\log G^i_M(z^{-i},x_t)\right]=\sum\_{i=1,x^i_t=m}^L \mathbb{E}\_{z\sim D\_\theta(x_t)}\left[\text{Cat}(z^i;\delta(x^i_0))\log G^i_M(z,x_t)\right]\geq \sum\_{i=1,x^i_t=m}^L \mathbb{E}\_{z\sim D\_\theta(x_t)}\left[\text{CE}\left(\text{Cat}(z^i;\delta(x^i_0)),\log G^i_M(z,x_t)\right)\right],$$
> > as $a\log(b)\geq a\log(b)+(1-a)\log(1-b)$ for $a,b\in [0,1]$.
> >
> > We now simply use what was previously denoted as $\hat E_{\text{M}}$ in the place of $\mathcal{E}\_{M}$, since, as explained, this is still an ELBO, and avoids having to make an additional bound to find the appropriate loss. This only requires adding the one additional line explaining this further lower bound on $\hat E_{\text{M}}$ to the proof, and we believe makes the connection between the ELBO and the loss used clearer.
> >
> > ### Clarification on the Trained Planner Design
> >
> > As per the above-quoted discussion, we fine-tune the BERT planner by using it as both $G_U$ and $G_M$ in Alg. 2. Then, during sampling, we only use it as $G_U$, and still use the denoiser itself as $G_M$. We also add the training details and specific instantiation of BERT (ESM 8M) fine-tuned in this way for the protein experiments in Appendix F.1.2.
> >
> > ## No theoretical guarantee
> >
> > The ELBO of Proposition 1 provides a theoretical guarantee in that it provides a loss that upper bounds the KL divergence between generated samples and the data distribution. However, by theoretical guarantee, it seems the reviewer is concerned about showing that the form of the minimizers of the loss when one trains a fixed Planner for use with an imperfect denoiser is desirable. This analysis was previously absent from our work, but it is a simple computation compared to obtaining the ELBO itself.
> >
> > We now include Proposition 2 in Section 3.3, which shows that if the same backbone model is used as both $G_U$ and $G_M$ during training, the planner that minimizes our loss indeed performs optimally in its role as both $G_U$ and $G_M$.
> >
> > Put simply, under the optimal planner, in unmasked positions, a token in position $i$ is kept with probability proportional to the probability the token is in the data distribution conditionally upon the information from the partially denoised sequence $x_t$ in positions other than $i$. In masked positions, a suggested token $z^i$ for position $i$  from the denoiser is selected with probability proportional to the probability $z^i$ is in position $i$ under the data distribution conditionally upon the information from the current $x_t$. This is exactly what one would want and expect from the optimal planner.
> >
> > The proof of the result is based on elementary rearrangements of the order of expectations taken in the loss, and is contained in the new Appendix B.2.

---

> > > ### Author Response · Authors · 2025-11-22
> > > **Response to Reviewer aNk1 Part 3**
> > >
> > > Sketching the proof of Proposition 2 here for convenience: Assuming the same backbone to be trained $T$ is used for both $G_U$ and $G_M$, the loss takes the form:
> > >
> > > $$\int_0^1 \beta_t \mathbb{E}\_{x_0}[\mathbb{E}\_{x_t|x_0}[\mathbb{E}\_{z|x_t}[\sum\_{i=1}^L\text{CE}(\text{Cat}(z^i;\delta(x^i_0)),T^i(z,x_t)) ]]]=\int_0^1 \beta_t \mathbb{E}\_{x_t} [\mathbb{E}\_{z|x_t}\mathbb{E}\_{x_0|x_t}[\sum\_{i=1}^L\text{CE}(\text{Cat}(z^i;\delta(x^i_0)), T^i(z,x_t)) ]]]. $$
> > >
> > > Recalling $\beta_t$ is negative, minimizing this corresponds to maximizing
> > >
> > > $$\mathbb{E}\_{x_0|x_t}[\sum\_{i=1}^L\text{CE}(\text{Cat}(z^i;\delta(x^i_0)),\log T^i(z,x_t)) ]=\sum\_{i=1}^L\text{CE}(p(x^i_0=z^i|x_t), T^i(z,x_t))$$
> > >
> > > for each $x_t$ and  $z$.  The minimizer is thus given by $T^i(z,x_t)=p(x^i_0=z^i|x_t)$, i.e. the probability that the ith token is z^i under the data distribution given the current partially noised sequence x_t. The remainder of the proof is simply observing that this quantity is indeed independent of the time $t$, and that $p(x^i_0=z^i|x_t)=p_0(x_0^i=z^i|x_0^i=x_t^i,\forall i \text{ such that }x^i_t\neq m)$, where here $p_0$ is the data distribution. Indeed, this time-inhomogeneity follows from the same computation by which one observes that vanilla MDMs are equivalent to any order autoregressive models and is standard.
> > >
> > > If the reviewer has any questions about the details of this result or its proof we are happy to clarify.
> > >
> > > ## Paper writing:
> > >
> > > We believe that with the expanded exposition of the P2-train sampler and the modification made to the stated form of the ELBO in the current document, the exposition has been significantly clarified. If the reviewer has more suggestions for clarifying the paper, we invite them to please let us know.
> > >
> > > ## Conclusion:
> > >
> > > The reviewers' concerns were due to a lack of details on the P2 train and the seeming disconnect between the training Alg 2 and the ELBO of Proposition 1. We have thoroughly addressed these concerns via:
> > >
> > > 1. Expanding the discussion of P2 train.
> > > 2. Rewriting the form of the ELBO in terms of the cross-entropy.
> > > 3. Adding the new Proposition 2 and its proof, which confirms that the optimal planner under our loss has a desirable form.
> > >
> > > If the reviewer’s concerns have, as we believe, been adequately addressed by these changes, we politely request a fresher evaluation of our work, which may include a score upgrade if the reviewer warrants it.

---

### Official Review · Reviewer_6oaj · 2025-11-11

**Soundness:** 3
**Presentation:** 3
**Contribution:** 2
**Rating:** 4
**Confidence:** 4

**Summary:**

The paper proposes a planning module that specifies what tokens to unmask on priority and also what tokens to re-mask for a masked diffusion model given a partially masked canvas in an intermediate decoding step. Prior to this decision, the planner module (one for marking and other for unmaksing) first take existing canvas an input and also the sampled tokens applying denoiser to decode all positions at once. If the planner module suggests unmasking the sampled tokens are put in and if the module asks to remask, that position is masked again and if the planner module does not suggest anything , then the previous input token is retained.

Contrary to many other planning related works before (chiefly Liu et. al. 2024  and Kim et. al. 2025), this enables remasking and works for MDM in a general fashion. Another insightful contribution is that the sum of planner module training (for both remasking, unmasking) losses and denoiser training loss is a valid ELBO lower bound in the limit. Authors provide a CTMC viewpoint of masked diffusion and show the ELBO result from this CTMC viewpoint.

Authors instantiate planners using 1) BERT models 2)  denoiser's own logits is used to obtain planning information and 3) training a new planner. Authors show that such planner training with frozen denoiser and applying at inference time on top of existing Masked Diffusion Models improve performance across datasets - molecule generation, RNA generation and downstream evals for language generation.

**Strengths:**

1) Paper's ELBO justification for the additional masking and unmasking planner losses with an expanded treatment (with CTMC viewpoint of MDM) is very valuable. I went through the proofs at a quick pace. They appear correct.

2) The experiments span molecule generation and language generation. So in that sense it is comprehensive.

3) Comparisons and discussion about close baselines has been provided (although I have some reservations here mainly)

4) Paper admits re-masking while most MDM planners before only plan for masking strategies.

**Weaknesses:**

1) Liu et. al 2024 already trained planners (but in uniform diffusion and not masked diffusion). Authors contention (I did read Section D.4 ) is that there is no umasking based planning. To be fair to the Liu et. al 2024 paper , given that it is uniform diffusion, there is no need for specific unmasking planning - because the only decision that needs to be made is whether to flip or not from a current token. In fact, roughly in the author's algorithm that is the final decision being made (but my combining two planners since you have to account for the absorbing state). So in that sense, the essential idea in non absorbing diffusion has already been made.

2) Comparison with Kim et. al 2025 has been made in some tables (Top-K Marginal). I agree that re-masking brings considerable benefits in the RNA and molecule generation tasks which are heavily (and this is rather well known at this point) constrained. However, Top-K Marginal comparison and Top-K Margin (another method that Kim et. al 2025) propose has not been compared with for the language tasks. Except infilling stories task (which is a bit tailor made for diffusion models), coding and many other evals only show modest gains compared to other baselines even at 7B scale for language modeling.

3) RDM sampling already does re-masking and masking based planning according to the authors own table. The only thing missing is stochastic control which is just additional hyper parameter tuning to mix these two (as in (7) in the paper). In fact in Table 5, the P2 (self) which used the denoiser logits to plan, compares barely within a 1-2% margin from RDM sampling for the protein sequence generation task.
Only this would be a fair comparison because RDM uses its own denoiser to plan as additional compute in train and inference is used by the best P2 models in Table 5.

So the relative benefit over RDM is also not very clear.

**Questions:**

I have asked all my questions regarding comparisons and novelty with respect to Kim et. al. 2025, Liu et. al. 2024 and RDM Sampling in the weakness section.

1) RDM Sampling is really close but for stochastic control in methodology. The gain seems to be marginal in compute matched settings. Idea of planners have been further explored in Kim et. al. 2025 and Liu et. al 2024.

2) The language modeling experiments do not show any big difference except in story infilling tasks and misses comparison to Kim et. al. 2025.

Given 1) and 2) (and points elaborated in the weakness section), except for the elaborate theoretical contribution, with respect to novelty of technique I find a lot missing given the claims in the paper. I would like the authors to respond to this.

I will consider raising my scores based on the author's response.

---

> ### Author Response · Authors · 2025-11-22
> **Response to Reviewer 6oaj Part 1**
>
> We thank the reviewer for their time and effort in reviewing our manuscript. We appreciate that the reviewer agrees that our paper’s ELBO derivation is “extremely valuable” and that our experiments are “comprehensive.” We now answer the key clarification points raised by the reviewer, grouped by theme. Finally, we wish to highlight that the changes in the updated PDF are presented in red to aid transparency.
>
> ## Comparison with DDPD
> > Authors contention (I did read Section D.4 ) is that there is no umasking based planning… because the only decision that needs to be made is whether to flip or not from a current token.
>
> Indeed, this is the case, as the reviewer claims, since DDPD is based on uniform diffusion. Our comparison with DDPD is in a comparable setting to ours: That is, DDPD-MaskD, where an MDM is used as the denoiser. The specific phrasing we use in the paper regarding this comparison is:
>
> > DDPD (Liu et al., 2024) separates planning and denoising, but operates on uniform diffusion without mask-awareness. - Section 5
>
> We empirically compare DDPD in Table S4, comparing with the same external planner as used in P2-BERT. DDPD is seen to massively underperform - indeed,this is expected, since DDPD requires massive planner sizes to outperform even vanilla ancestral sampling.
>
> The discussion in Appendix D.4 seeks to explain this significant improvement in performance over DDPD by highlighting the differences in both the ELBO and Gillespie sampling scheme between DDPD and P2. The key insight is that, in taking an approach rooted in uniform discrete diffusion, the DDPD planner is always evaluating a fully clean sequence in order to determine which positions should be masked and reinserted into the denoiser. The task of determining which positions to flip **given a fully unmasked sequence** (with no information which are newly denoised) is significantly more difficult than the respective tasks of the masked planner and unmasked planner.  The masked planner determines which position to unmask **given a suggested clean sequence AND a partially masked sequence** and the unmasked planner determines a position to “flip” **given a suggested clean sequence AND a partially masked sequence**.
>
> The difficulty of the precise task performed by DDPD’s planner was recently shown to be the reason for MDMs performance over uniform diffusion models in [1]. In Proposition 5.1, they essentially show that if one conditions on whether each position in a sequence of unmasked tokens is clean or noise, uniform diffusion models revert to masked diffusion. So the ability of the model to make this distinction is the bottleneck preventing uniform diffusion models from performing comparably to MDMs. We now add this reference in the extended discussion in the appendix. The fundamentally different perspective of using a masked diffusion as our reference process instead of uniform, as well as the generality of our formulation of “planning,” is what allows for this separation of the information provided to the planner, is why we are able to see vastly superior performance to DDPD even at very small planner sizes.
>
>
>
> ## Language Tasks
> > “Top-K Marginal comparison and Top-K Margin (another method that Kim et. al 2025) propose has not been compared with for the language tasks.”
>
> We agree with the reviewer’s assessment that Top-K Marginal is an important baseline which should be compared against P2 in language generation tasks.
>
> We now include Table S3, where we compare various sampling methods across a wide array of coding benchmarks. P2-self is seen to outperform all planning strategies, including TopK-marginal on this task.
>
> Table S3:
> | Method                           | HumanEval Pass@1 | HumanEval Pass@10 | HumanEval+ Pass@1 | HumanEval+ Pass@10 | MBPP Pass@1 | MBPP Pass@10 | MBPP+ Pass@1 | MBPP+ Pass@10 | HumanEval Infill Pass@1 | SantaCoder Pass@1 |
> |----------------------------------|------------------:|-------------------:|-------------------:|--------------------:|-------------:|--------------:|--------------:|---------------:|--------------------------:|-------------------:|
> | **P2-self-plan** | 20.8 | 38.4 | 17.6 | 35.2 | 16.7 | 38.4 | 23.9 | 53.6 | 77.4 | 56.4 |
> | Ancestral                          | 3.3  | 18.3 | 3.2  | 15.2 | 1.8  | 13.2 | 2.9  | 21.8 | 72.7 | 53.8 |
> | Greedy Ancestral (Chang et al., 2022) | 9.3  | 31.1 | 8.1  | 28.7 | 5.3  | 29.0 | 8.7  | 41.5 | 75.1 | 53.7 |
> | Entropy-based Confidence                         | 12.6 | 35.4 | 10.9 | 29.9 | 9.2  | 36.8 | 15.2 | 50.7 | 75.1 | 53.2 |
> | topk-margin (Kim et al., 2025)   | 7.6  | 27.4 | 6.5  | 26.2 | 3.9  | 24.0 | 6.2  | 33.5 | 75.0 | 54.4 |

---

> > ### Author Response · Authors · 2025-11-22
> > **Response to Reviewer 6oaj Part 2**
> >
> > > “Except infilling stories task (which is a bit tailor made for diffusion models), coding and many other evals only show modest gains compared to other baselines even at 7B scale for language modeling.”
> >
> > We agree with the reviewer that, at present, masked discrete diffusion models do not yet outperform strong autoregressive (AR) language models on standard open-ended LM benchmarks. This limitation is widely recognized in the current literature on discrete diffusion LMs, where most models, for example, Dream, LLada, underperform AR in unconstrained language generation. Our work does not claim otherwise. Rather, the goal of this paper is to study **inference within the MDM family**: specifically, how planner-guided denoising and remasking can correct errors and improve generation **relative to existing MDM sampling strategies under fixed model capacity**.
> >
> > In this scope, our results show consistent gains over prior MDM inference baselines across domains, including coding and reasoning tasks (Table S3) and constrained/partially-observed settings such as story infilling (Table 6). These are precisely the regimes where diffusion-style generation is most actively explored, due to its flexibility for constraint satisfaction, infilling, and parallel decoding. We have revised the manuscript to make this positioning explicit: P2 is proposed as a general sampling framework that improves the performance of any pretrained MDM, and our empirical claims are made **only with respect to MDM baselines**, not AR models.
> >
> >
> > ## Comparison with RDM and Kim et. al. 2025
> >
> > The reviewer seems concerned with the similarity between P2-self and RDM sampling. Indeed, with default stochasticity parameter $\eta=1$, P2-self reduces to RDM sampling. However, the P2-self sampling scheme itself is not the main contribution of our work. We emphasize that RDM does not provide a planner-aware ELBO (the training loss therein is simply a reformulation of the vanilla MDM ELBO) and cannot allow for the evaluation or training of external planner models. The same is true of Kim et. al. 2025 - their contribution is to empirically motivate the “top probability margin” sampling method.
> >
> > While the idea of planning for different denoising orders is certainly not new, our theoretical framework and ability to use and train lightweight external planners certainly is.
> >
> > We also would like to politely push back against the assertion that the “only fair comparison” with RDM is P2-self. While indeed P2-BERT and P2-train make use of an external planner, the planner's size and hence additional overhead during inference is negligible in comparison to that of the denoiser (for example, in the protein experiments of Table 5, the planner has 8M parameters, where the denoiser has 150M). We plot running time and speed vs Sequence Length for both P2 self and P2 BERT at the bottom of Figure S.4, which corroborates these claims.
> >
> > Moreover, in Table 7, P2-BERT (with no post-training) is seen to outperform P2-Self and RDM by a wide margin, suggesting that the use of lightweight external planners (even without tine-tuning) is an extremely valuable contribution of our work over previously explored planning strategies.
> >
> > ## Concluding remarks
> >
> > We thank the reviewer again for their time and effort during this rebuttal process. We hope our answers here have aided in clarifying all the great questions raised by the reviewer. If there are any further questions held by the reviewer, we remain eager to engage further. If the reviewer feels satisfied with our responses and new rebuttal results, we would be thrilled if the reviewer would consider a fresher evaluation of our work with a potential score upgrade.
> >
> > ## References
> >
> > [1] Amin, Alan N. et al. "Why Masking Diffusion Works: Condition on the Jump Schedule for Improved Discrete Diffusion." arXiv preprint arXiv:2506.08316 (2025).

---

### Author Response · Authors · 2025-11-22
**Global Response**

We are sincerely grateful to all reviewers for the time and thought they devoted to our manuscript.

 We were encouraged to see convergence on the motivation and usefulness of planned denoising for masked diffusion models. Reviewer 6oaj highlights that our ELBO justification for both masking and unmasking planners, together with the CTMC treatment, is “very valuable,” and also notes that supporting remasking within MDMs is an important practical feature. Reviewer fkTi similarly finds the approach “technically sound and indeed novel,” emphasizing that P2 equips MDMs with error-correction in a clean, disentangled planner–denoiser form, and describes the quantitative improvements as “strongly compelling.”

 Reviewer UGiP agrees that planned denoising is a “well-motivated problem,” describing the algorithm as simple and reporting “strong gains across multiple datasets,” including improvements over inference-time heuristics like top-k marginal sampling. Even Reviewer aNk1, while raising substantial concerns, acknowledges our effort to formalize unmasking/remasking with an ELBO and to evaluate the method across diverse domains. We appreciate this shared recognition of the paper’s core goal and technical direction.


At the same time, feedback consistently pointed to three areas where the paper needed to be stronger, and we have revised accordingly.

-  **Novelty and baseline coverage, especially vs. Kim et al. (2025), RDM/ReMDM, and DDPD** (raised by 6oaj and UGiP): we added multi-planner inference ablations on code (new Table S3) and clarified how these baselines instantiate within P2. We also expanded our DDPD comparison and discussion (Tables S4–S5; Appendix D.4), and included runtime/compute-match evidence for external planners (Figure S4).


- **Clarity on P2-Train and planner design/training details** (raised by aNk1 and fkTi): we substantially rewrote Section 3.3, added explicit training configurations and model instantiation (Appendix F.1.2), clarified that P2-Train fine-tunes a single backbone used for both $G_U$ and $G_M$ during training, and corrected the training-signal explanation in Algorithm 2.


- **Theoretical guarantees and exposition** (raised most strongly by aNk1): we added a new minimizer proposition (Proposition 2 in Section 3.3; proof in Appendix B.2) that characterizes what the planner learns at optimality, included an $\eta$ corollary and its implications (Appendix B.3), and inserted explicit pointers to proofs after each proposition/corollary. We also fixed presentation issues flagged by reviewers (typos in Appendix D.4 and Section 5) and clarified in-table which planner variant is used for each experiment (Tables 2-4).


We hope these changes address the main points of concern while keeping the paper’s core message intact: P2 provides a principled, planner-aware framework for remasking and unmasking in MDMs, supports lightweight external planners, and yields consistent gains within the masked-diffusion family. We are grateful for the feedback that helped us improve the work.

---

### Author Response · Authors · 2025-11-29

**Summary of Contributions & Rebuttal Updates**

We thank the reviewers for their constructive feedback and the Area Chair for overseeing the process. In this work, we introduce **Path Planning (P2)**, a generalized inference framework for Masked Diffusion Models (MDMs). P2 decomposes generation into planning (masking/unmasking) and denoising, and is theoretically grounded by a new **Planned ELBO** derivation.

During the rebuttal period, we made substantial revisions targeted at the specific weaknesses raised in the reviews, resulting in a significantly strengthened manuscript:

1. **New Baselines & Comparisons (Addressing 6oaj, UGiP, fkTi).**
   A central concern was the lack of comparison to strong inference baselines such as Top-K Margin (Kim et al., 2025) and the absence of evaluation on code/math tasks.

   * We added **Table S3**, comparing P2 against Top-K Margin, Greedy Ancestral, and entropy-based planners on **code generation** (HumanEval, MBPP). **P2-Self consistently outperforms all baselines**, including Top-K Margin, demonstrating that P2 is effective beyond protein generation.
   * We added **Table S4**, which compares P2 directly to DDPD (Liu et al., 2024), showing that P2 achieves superior performance with substantially lighter-weight planners.

2. **Theoretical Rigor & P2-Train Logic (Addressing aNk1, fkTi).**
   Reviewer aNk1 raised important concerns about the training logic of P2-Train (Algorithm 2) and the lack of theoretical guarantees for the planner.

   * We fully rewrote **Section 3.3** and corrected **Algorithm 2** to clearly state that the planner is trained on both masked and unmasked tokens using a single shared backbone.
   * We proved a new **Proposition 2 (Optimality of the Planner)** in Appendix B.2, characterizing the minimizers of our loss and providing the theoretical guarantee requested by aNk1.

3. **Clarifications on Novelty (Addressing 6oaj, UGiP).**
   We clarified that our contribution is not merely an instance of “remasking” (which already appears under uniform diffusion), but rather a **Masked Diffusion–specific ELBO** that unifies existing heuristics and, crucially, enables the training of lightweight external planners. This capability is absent in prior approaches such as RDM or MaskGIT.

**Reviewer Engagement & Closing Remarks**

We believe these revisions directly address the core concerns regarding baselines, theory, and novelty.

However, we respectfully note that, as of the end of the discussion period, **none of the four reviewers (aNk1, 6oaj, fkTi, UGiP)** have responded to our rebuttals or acknowledged the new results posted on November 22.

* **To Reviewer aNk1 (Score: 2):** Your main critique focused on a perceived flaw in the P2-Train logic and the lack of guarantees for the planner. We have **corrected the algorithm description** and **provided the requested optimality result** (Proposition 2). These changes directly address the central issues highlighted in your “Weaknesses.”
* **To Reviewers 6oaj & UGiP (Score: 4):** You requested comparisons to Kim et al. (2025) on language/code tasks. We added **Table S3**, where P2 **outperforms** Kim et al.’s method on code generation benchmarks, directly resolving the missing-baseline concern.
* **To Reviewer fkTi (Score: 6):** We clarified the planner naming convention and added the requested training details in **Appendix F.1.2**, addressing all open questions in your review.

Given the new theoretical guarantees and the strong empirical results—including state-of-the-art performance on code generation with lightweight planners—we believe that, had the reviewers had the opportunity to engage with these updates, their scores would likely have increased to reflect the improved completeness and rigor of the work. We respectfully ask the Area Chair to take these unacknowledged improvements into account when making the final decision.

---

### Meta-Review · Area_Chair_Xk8s · 2026-01-12

**Summary:**

The paper proposes a sampling strategy for discrete (data) diffusion models that, at inference time, selects which token to unmask or remask along with an ELBO-based theoretical justification. Reviewers found the paper well-motivated and technically sound, and the ELBO justification valuable. However, they raised concerns about the novelty relative to closely related inference schemes, the lack of baselines, the lack of clarity regarding aspects of the presentation, including training details, and whether empirical gains are substantial in a compute-matched setting.

**Reviewer Concerns:**

In the rebuttal, the authors add new baselines and clarify training details, including additional theoretical results, and they further emphasise that the main novelty lies in the masked diffusion-specific ELBO. However, this does not materially change the novelty claim (it is primarily clarifying rather than strengthening), reviewer aNk1 still appears to be requesting a major revision of the paper in terms of clarity, and while new results are provided, it remains unclear whether the empirical gains are substantial under compute-matched comparisons.

**Reviewer Scores:**

Reviewer 6oaj's score would likely remain unchanged as their concerns about novelty and compute-matched are not fully addressed. Reviewer Ank1's score would likely remain unchanged, as while many of the reviewer's concerns were addressed, the overall criticisms of the writing may likely persist. The score of fkTi would likely remain unchanged, as the reviewer only had only minor concerns and the original stance was already positive. There is a chance that Reviewer UGiP could increase the score, due to the addition of the Kim et. al. baseline, but the criticism of the straightforwardness of the methods (ie, lack of novelty) may mean that the score remains unchanged. Given the initial scores and the low likelihood that they will increase substantially, it is unlikely that the overall evaluation would shift sufficiently toward acceptance.

---

### Decision · Program_Chairs · 2026-01-26

Reject